# FEDERATED OPTIMIZATION ALGORITHMS WITH RANDOM RESHUFFLING AND GRADIENT COMPRESSION

## ABSTRACT

Gradient compression is a popular technique for improving communication complexity of stochastic first-order methods in distributed training of machine learning models. However, the existing works consider only with-replacement sampling of stochastic gradients. In contrast, it is well-known in practice and recently confirmed in theory that stochastic methods based on without-replacement sampling, e.g., Random Reshuffling (RR) method, perform better than ones that sample the gradients with-replacement. In this work, we close this gap in the literature and provide the first analysis of methods with gradient compression and without-replacement sampling. We first develop a distributed variant of random reshuffling with gradient compression (Q-RR), and show how to reduce the variance coming from gradient quantization through the use of control iterates. Next, to have a better fit to Federated Learning applications, we incorporate local computation and propose a variant of Q-RR called Q-NASTYA. Q-NASTYA uses local gradient steps and different local and global stepsizes. Next, we show how to reduce compression variance in this setting as well. Finally, we prove the convergence results for the proposed methods and outline several settings in which they improve upon existing algorithms.

## 1 INTRODUCTION

Federated learning (FL) (Konečný et al., 2016; McMahan et al., 2017) is a framework for distributed learning and optimization where multiple nodes connected over a network try to collaboratively carry out a learning task. Each node has its own dataset and cannot share its data with other nodes or with a central server, so algorithms for federated learning often have to rely on local computation and cannot access the entire dataset of training examples. Federated learning has applications in language modelling for mobile keyboards (Liu et al., 2021), healthcare (Antunes et al., 2022), wireless communications (Yang et al., 2022), and continues to find applications in many other areas (Kairouz et al., 2019).

Federated learning tasks are often solved through *empirical-risk minimization* (ERM), where the $m$-th devices contributes an empirical loss function $f_m(x)$ representing the average loss of model $x$ on its local dataset, and our goal is to then minimize the average loss over all the nodes:

$$\min_{x \in \mathbb{R}^d} \left[ f(x) \stackrel{\text{def}}{=} \frac{1}{M} \sum_{m=1}^{M} f_m(x) \right], \tag{1}$$

where the function $f$ represents the average loss. Every $f_m$ is an average of sample loss functions $f_m^i$ each representing the loss of model $x$ on the $i$-th datapoint on the $m$th clients' dataset: that is for each $m \in \{1, 2, \dots, M\}$ we have

$$f_m(x) \stackrel{\text{def}}{=} \frac{1}{n_m} \sum_{i=1}^{n_m} f_m^i(x).$$

For simplicity we shall assume that the datasets on all clients are of equal size: $n_1 = n_2 = \dots = n_M$, though this assumption is only for convenience and our results easily extend to the case when clients

---

**Algorithm 1** The generalized FedAvg framework for methods with local steps

---

**Input:** $x_0$ – starting point, $\gamma > 0$ – local stepsize, $\eta > 0$ – global stepsize, $H$ - number of local steps

1: **for** communication rounds $t = 0, 1, \ldots, T - 1$ **do**
2:    **for** clients $m \in [M]$ in parallel **do**
3:       Receive $x_t$ from the server and set $x_{t,m}^0 = x_t$
4:       **for** local steps $i = 0, 1, \ldots, H$ **do**
5:          Set $x_{t,m}^{i+1} = \text{ClientUpdate}(x_{t,m}^i, \gamma, m, i)$
6:       **end for**
7:       Send $x_{t,m}^H$ to the server, or alternatively send the update $\Delta_{t,m} = x_{t,m}^H - x_t$ to the server
8:    **end for**
9:    Compute $x_{t+1} = \frac{1}{M} \sum_{m=1}^M x_{t,m}^H$ (or $x_{t+1} = x_t + \frac{1}{M} \sum_{m=1}^M \Delta_{t,m}$ if clients sent updates)
10:   Broadcast $x_{t+1}$ to the clients
11: **end for**
**Output:** $x_T$

---

have datasets of unequal sizes. Thus our optimization problem is

$$\min_{x \in \mathbb{R}^d} \left[ f(x) = \frac{1}{nM} \sum_{m=1}^M \sum_{i=1}^n f_m^i(x) \right]. \tag{2}$$

Because $d$ is often very large in practice, the dominant paradigm for solving (2) relies on first-order (gradient) information. Federated learning algorithms have access to two key primitives: (a) local computation, where for a given model $x \in \mathbb{R}^d$ we can compute stochastic gradients $\nabla f_m^i(x)$ locally on client $m$, and (b) communication, where the different clients can exchange their gradients or models with a central server.

## 1.1 COMMUNICATION BOTTLENECK: FROM ONE TO MULTIPLE LOCAL STEPS

In practice, communication is more expensive than local computation (Kairouz et al., 2019), and as such one of the chief concerns of algorithms for federated learning is communication efficiency. Algorithms for federated learning have thus focused on achieving communication efficiency, with one common ingredient being the use of multiple *local steps* (Wang et al., 2021; Malinovskiy et al., 2020), where each node uses multiple gradients locally for several descent steps between communication steps. In general, algorithms using local steps fit the following pattern of generalized FedAvg (due to (Wang et al., 2021)); see Algorithm 1.

When the client update method in Algorithm 1 is stochastic gradient descent, we get the FedAvg algorithm (also known as Local SGD). While FedAvg is popular in practice, recent theoretical progress has given tight analysis of the algorithm and shown that it can be definitively slower than its non-local counterparts (Khaled et al., 2020; Woodworth et al., 2020a; Glasgow et al., 2022). However, by using bias-reduction techniques one can use local steps and still maintain convergence rates at least as fast as non-local methods (Karimireddy et al., 2020), or in some cases even faster (Mishchenko et al., 2022). Thus local steps continue to be a useful algorithmic ingredient in both theory and practice for achieving communication efficiency.

## 1.2 COMMUNICATION BOTTLENECK: FROM FULL-DIMENSIONAL TO COMPRESSED COMMUNICATION

Another useful ingredient in distributed optimization is *gradient compression*, where each client sends a compressed or quantized version of their update $\Delta_{t,m}$ instead of the full update vector, potentially saving communication bandwidth by sending fewer bits over the network. There are many operators that can be used for compressing the update vectors: stochastic quantization (Alistarh et al., 2017), random sparsification (Wangni et al., 2018; Stich et al., 2018), and others (Tang et al., 2020). In this work we consider compression operators satisfying the following assumption:

**Assumption 1.** *A compression operator is an operator* $\mathcal{Q} : \mathbb{R}^d \to \mathbb{R}^d$ *such that for some* $\omega > 0$, *the relations*

$$\mathbb{E}\left[\mathcal{Q}(x)\right] = x \qquad and \qquad \mathbb{E}\left[\|\mathcal{Q}(x) - x\|^2\right] \leq \omega\|x\|^2$$

*hold for* $x \in \mathbb{R}^d$.

Unbiased compressors can reduce the number of bits clients communicate per round, but also increases the variance of the stochastic gradients used slowing down overall convergence, see e.g. (Khirirat et al., 2018, Theorem 5.2) and (Stich, 2020, Theorem 1). By using control iterates, Mishchenko et al. (2019b) developed DIANA—an algorithm that can reduce the variance due to gradient compression with unbiased compression operators, and thus ensure fast convergence. DIANA has been extended and analyzed in many settings (Horváth et al., 2019; Stich, 2020; Safaryan et al., 2021) and forms an important tool in our arsenal for using gradient compression.

### 1.3 COMMUNICATION BOTTLENECK: FROM WITH REPLACEMENT TO WITHOUT REPLACEMENT SAMPLING

The algorithmic framework of generalized FedAvg (Algorithm 1) requires specifying a client update method that is used locally on each client. The typical choice is stochastic gradient descent (SGD), where at each time step we sample $j$ from $\{1, \ldots, n\}$ uniformly at random and then do a gradient descent step using the stochastic gradient $\nabla f_m^j(x_{t,m}^i)$, resulting in the client update:

$$\text{ClientUpdate}(x_{t,m}^i, \gamma, m, i) = x_{t,m}^i - \gamma\nabla f_m^j(x_{t,m}^i).$$

This procedure thus uses *with-replacement sampling* in order to select the stochastic gradient used at each local step from the dataset on node $m$. In contrast, we can use *without-replacement sampling* to select the gradients: that is, at the beginning of each *epoch* we choose a permutation $\pi_1, \pi_2, \ldots, \pi_n$ of $\{1, 2, \ldots, n\}$ and do the $i$-th update using the $\pi_i$-ith gradient:

$$\text{ClientUpdate}(x_{t,m}^i, \gamma, m, i) = x_{t,m}^i - \gamma\nabla f_m^{\pi_i}(x_{t,m}^i).$$

Without-replacement sampling SGD, also known as Random Reshuffling (RR), typically achieves better asymptotic convergence rates compared to with-replacement SGD and can improve upon it in many settings as shown by recent theoretical progress (Mishchenko et al., 2020; Ahn et al., 2020; Rajput et al., 2020; Safran and Shamir, 2021). While with-replacement SGD achieves an error proportional to $\mathcal{O}\left(\frac{1}{T}\right)$ after $T$ steps (Stich, 2019), Random Reshuffling achieves an error of $\mathcal{O}\left(\frac{n}{T^2}\right)$ after $T$ steps, faster than SGD when the number of steps $T$ is large.

The success of RR in the single-machine setting has inspired several recent methods that use it as a local update method as part of distributed training: Mishchenko et al. (2021) developed a distributed variant of random reshuffling, FedRR. FedRR fits into the framework of Algorithm 1 and uses RR as a local client update method in lieu of SGD. They show that FedRR can improve upon the convergence of Local SGD when the number of local steps is fixed as the local dataset size, i.e. when $H = n$. Yun et al. (2021) study the same method under the name Local RR under a more restrictive assumption of bounded inter-machine gradient deviation and show that by varying $H$ to be smaller than $n$ better rates can be obtained in this setting than the rates of Mishchenko et al. (2021). Other work has explored more such combinations between RR and distributed training algorithms (Huang et al., 2021; Malinovsky et al., 2022; Horváth et al., 2022).

### 1.4 THREE TRICKS FOR ACHIEVING COMMUNICATION EFFICIENCY

To summarize, we have at our disposal the following tricks and techniques for achieving communication efficiency in distributed training: (a) Local steps, (b) Gradient compression, and (c) Random Reshuffling. Each has found its use in federated learning and poses its own challenges, requiring special analysis or bias/variance-reduction techniques to achieve the best theoretical convergence rates and practical performance. Client heterogeneity causes local methods (with or without random reshuffling) to be biased, hence requiring bias-reduction techniques (Karimireddy et al., 2020; Murata and Suzuki, 2021) or decoupling local and server stepsizes (Malinovsky et al., 2022). Gradient compression reduces the number of bits clients have to send per round, but causes an increase in variance, and we hence also need variance-reduction techniques to achieve better convergence rates

under gradient compression (Mishchenko et al., 2019b; Stich and Karimireddy, 2019). However, it is not clear apriori *how these techniques should be combined to improve the convergence speed*, and this is our starting point.

## 1.5 CONTRIBUTIONS

In this paper, we aim to develop methods for federated optimization that combine gradient compression, random reshuffling, and/or local steps. While each of these techniques can aid in reducing the communication complexity of distributed optimization, their combination is under-explored. Thus our goal is to design methods that improve upon existing algorithms in convergence rates and in practice. We summarize our contributions as:

◇ **The issue: naïve combination has no improvements.** As a natural step towards our goal, we start with non-local methods and propose a new algorithm, Q-RR (Algorithm 2), that combines random reshuffling with gradient compression at every communication round. However, for Q-RR our theoretical results do not show any improvement upon QSGD when the compression level is reasonable. Moreover, we observe similar performance of Q-RR and QSGD in various numerical experiments. Therefore, we conclude that this phenomenon is not an artifact of our analysis but rather an issue of Q-RR: communication compression adds an additional noise that dominates the one coming from the stochastic gradients sampling.

◇ **The remedy: reduction of compression variance.** To remove the additional variance added by the compression and unleash the potential of Random Reshuffling in distributed learning with compression, we propose DIANA-RR (Algorithm 3), a combination of Q-RR and the DIANA algorithm. We derive the convergence rates of the new method and show that it improves upon the convergence rates of Q-RR, QSGD, and DIANA. We point out that to achieve such results we use $n$ shift-vectors per worker in DIANA-RR unlike DIANA that uses only 1 shift-vector.

◇ **Extensions to the local steps.** Inspired by the NASTYA algorithm of Malinovsky et al. (2022), we propose a variant of NASTYA, Q-NASTYA (Algorithm 4), that naïvely mixes quantization, local steps with random reshuffling, and uses different local and server stepsizes. Although it improves in per-round communication cost over NASTYA but, similar to Q-RR, we show that Q-NASTYA suffers from added variance due to gradient quantization. To overcome this issue, we propose another algorithm, DIANA-NASTYA (Algorithm 5), that adds DIANA-style variance reduction to Q-NASTYA and removes the additional variance.

Finally, to illustrate our theoretical findings we conduct experiments on federated linear regression tasks.

## 1.6 RELATED WORK

Federated optimization has been the subject of intense study, with many open questions even in the setting when all clients have identical data (Woodworth et al., 2020b;a; 2021). The FedAvg algorithm (also known as Local SGD) has also been a subject of intense study, with tight bounds obtained only very recently by Glasgow et al. (2022). It is now understood that using many local steps adds bias to distributed SGD, and hence several methods have been developed to mitigate it, e.g. (Karimireddy et al., 2020; Murata and Suzuki, 2021), see the work of Gorbunov et al. (2021) for a unifying lens on many variants of Local SGD. Note that despite the bias, even vanilla FedAvg/Local SGD still reduces the overall communication overhead in practice (Ortiz et al., 2021).

There are several methods that combine compression or quantization and local steps: both Basu et al. (2019) and Reisizadeh et al. (2020) combined Local SGD with quantization and sparsification, and Haddadpour et al. (2021) later improved their results using a gradient tracking method, achieving linear convergence under strong convexity. In parallel, Mitra et al. (2021) also developed a variance-reduced method, FedLin, that achieves linear convergence under strong convexity despite using local steps and compression. The paper most related to our work is (Malinovsky and Richtárik, 2022) in which the authors combine *iterate* compression, random reshuffling, and local steps. We study *gradient* compression instead, which is a more common form of compression in both theory and practice (Kairouz et al., 2019). We compare our results against (Malinovsky and Richtárik, 2022) and show we obtain better rates compared to their work.

---

**Algorithm 2** Q-RR: Distributed Random Reshuffling with Quantization

---

**Input:** $x_0$ – starting point, $\gamma > 0$ – stepsize
1: **for** $t = 0, 1, \ldots, T - 1$ **do**
2:     Receive $x_t$ from the server and set $x_{t,m}^0 = x_t$
3:     Sample random permutation of $[n]$: $\pi_m = (\pi_m^0, \ldots, \pi_m^{n-1})$
4:     **for** $i = 0, 1, \ldots, n - 1$ **do**
5:         **for** $m = 1, \ldots, M$ in parallel **do**
6:             Receive $x_t^i$ from the server, compute and send $\mathcal{Q}\left(\nabla f_m^{\pi_m^i}(x_t^i)\right)$ back
7:         **end for**
8:         Compute and send $x_t^{i+1} = x_t^i - \gamma \frac{1}{M} \sum_{m=1}^M \mathcal{Q}\left(\nabla f_m^{\pi_m^i}(x_t^i)\right)$ to the workers
9:     **end for**
10:     $x_{t+1} = x_t^n$
11: **end for**
**Output:** $x_T$

---

## 2    Algorithms and convergence theory

We will primarily consider the setting of strongly-convex and smooth optimization. We assume that the average function $f$ is strongly convex:

**Assumption 2.** *Function $f : \mathbb{R}^d \to \mathbb{R}$ is $\mu$-strongly convex, i.e., for all $x, y \in \mathbb{R}^d$,*

$$f(x) - f(y) - \langle \nabla f(y), x - y \rangle \geq \frac{\mu}{2} \|x - y\|^2, \tag{3}$$

*and functions $f_1^i, f_2^i, \ldots, f_M^i : \mathbb{R}^d \to \mathbb{R}$ are convex for all $i = 1, \ldots, n$.*

Examples of objectives satisfying Assumption 2 include $\ell_2$-regularized linear and logistic regression. Throughout the paper, we assume that $f$ has the unique minimizer $x_* \in \mathbb{R}^d$. We also use the assumption that each individual loss $f_m^i$ is smooth, i.e. has Lipschitz-continuous first-order derivatives:

**Assumption 3.** *Function $f_m^i : \mathbb{R}^d \to \mathbb{R}$ is $L_{i,m}$-smooth for every $i \in [n]$ and $m \in [M]$, i.e., for all $x, y \in \mathbb{R}^d$ and for all $m \in [M]$ and $i \in [n]$,*

$$\|\nabla f_m^i(x) - \nabla f_m^i(y)\| \leq L_{i,m}\|x - y\|. \tag{4}$$

*We denote the maximal smoothness constant as $L_{\max} \stackrel{\text{def}}{=} \max_{i,m} L_{i,m}$.*

For some methods, we shall additionally impose the assumption that *each* function is strongly convex:

**Assumption 4.** *Each function $f_m^i : \mathbb{R}^d \to \mathbb{R}$ is $\widetilde{\mu}$-strongly convex.*

The *Bregman divergence* associated with a convex function $h$ is defined for all $x, y \in \mathbb{R}^d$ as

$$D_h(x, y) \stackrel{\text{def}}{=} h(x) - h(y) - \langle \nabla h(y), x - y \rangle .$$

Note that the inequality (3) defining strong convexity can be compactly written as $D_f(x, y) \geq \frac{\mu}{2}\|x - y\|^2$.

### 2.1   Algorithm Q-RR

The first method we introduce is Q-RR (Algorithm 2). Q-RR is a straightforward combination of distributed random reshuffling and gradient quantization. This method can be seen as the stochastic without-replacement analogue of the distributed quantized gradient method of Khirirat et al. (2018).

We shall the use the notion of *shuffling radius* defined by Mishchenko et al. (2021) for the analysis of distributed methods with random reshuffling:

**Definition 2.1.** *Define the iterate sequence $x_\star^{i+1} = x_\star^i - \frac{\gamma}{M} \sum_{m=1}^M \nabla f_m^{\pi_m^i}(x_\star)$. Then the shuffling radius is the quantity*

$$\sigma_{rad}^2 \stackrel{\text{def}}{=} \max_i \left\{ \frac{1}{\gamma^2 M} \sum_{m=1}^M \mathbb{E} D_{f_m^{\pi^i}}(x_\star^i, x_\star) \right\} .$$

We now state the main convergence theorem for Algorithm 2:

**Theorem 2.1.** *Let Assumptions 1, 3, 4 hold and let the stepsize satisfy $0 < \gamma \leq \frac{1}{\left(1+2\frac{\omega}{M}\right)L_{\max}}$. Then, for all $T \geq 0$ the iterates produced by* Q-RR *(Algorithm 2) satisfy*

$$\mathbb{E}\|x_T - x_\star\|^2 \leq (1 - \gamma\widetilde{\mu})^{nT}\|x_0 - x_\star\|^2 + \frac{2\gamma^2\sigma_{rad}^2}{\widetilde{\mu}} + \frac{2\gamma\omega}{\widetilde{\mu}M}(\zeta_\star^2 + \sigma_\star^2), \tag{5}$$

*where $\zeta_\star^2 \overset{\text{def}}{=} \frac{1}{M}\sum_{m=1}^{M}\|\nabla f_m(x_\star)\|^2$, and $\sigma_\star^2 \overset{\text{def}}{=} \frac{1}{Mn}\sum_{m=1}^{M}\sum_{i=1}^{n}\|\nabla f_m^i(x_\star) - \nabla f_m(x_\star)\|^2$.*

All proofs are relegated to the appendix. By choosing the stepsize $\gamma$ properly, we can obtain the communication complexity (number of communication rounds) needed to find an $\varepsilon$-approximate solution as follows:

**Corollary 1.** *Under the same conditions as Theorem 2.1 and for Algorithm 2, there exists a stepsize $\gamma > 0$ such that the number of communication rounds $nT$ to find a solution with accuracy $\varepsilon > 0$ (i.e. $\mathbb{E}\|x_T - x_*\|^2 \leq \epsilon$) is equal to $\widetilde{\mathcal{O}}\left(\left(1 + \frac{\omega}{M}\right)\frac{L_{\max}}{\widetilde{\mu}} + \frac{\omega(\zeta_\star^2 + \sigma_\star^2)}{M\widetilde{\mu}^2\varepsilon} + \frac{\sigma_{rad}}{\sqrt{\widetilde{\mu}^3\varepsilon}}\right)$, where $\widetilde{\mathcal{O}}(\cdot)$ hides constants and logarithmic factors.*

The complexity of Quantized SGD (QSGD) is (Gorbunov et al., 2020): $\widetilde{\mathcal{O}}\left(\left(1 + \frac{\omega}{M}\right)\frac{L_{\max}}{\mu} + \frac{(\omega\zeta_\star^2 + (1+\omega)\sigma_\star^2)}{M\mu^2\varepsilon}\right)$. For simplicity, let us neglect the differences between $\mu$ and $\widetilde{\mu}$. First, when $\omega = 0$ we recover the complexity of FedRR (Mishchenko et al., 2021) which is known to be better than the one of SGD as long as $\varepsilon$ is sufficently small as we have $n\mu\sigma_\star^2/8 \leq \sigma_{rad}^2 \leq nL\sigma_\star^2/4$ from (Mishchenko et al., 2021). Next, when $M = 1$ and $\omega = 0$ (single node, no compression) our results recovers the rate of RR (Mishchenko et al., 2020).

However, it is more interesting to compare Q-RR and QSGD when $M > 1$ and $\omega > 1$, which is typically the case. In these settings, Q-RR and QSGD have *the same complexity* since the $\mathcal{O}(1/\varepsilon)$ term dominates the $\mathcal{O}(1/\sqrt{\varepsilon})$ one if $\varepsilon$ is sufficiently small. That is, the derived result for Q-RR has no advantages over the known one for QSGD unless $\omega$ is very small, which means that there is almost no compression at all. We also observe this phenomenon in the experiments.

The main reason for that is the variance appearing due to compression. Indeed, even if the current point is the solution of the problem ($x_t^i = x_*$), the update direction $-\gamma\frac{1}{M}\sum_{m=1}^{M}\mathcal{Q}\left(\nabla f_m^{\pi_m^i}(x_t^i)\right)$ has the compression variance

$$\mathbb{E}_{\mathcal{Q}}\left[\left\|\frac{\gamma}{M}\sum_{m=1}^{M}\left(\mathcal{Q}(\nabla f_m^{\pi_m^i}(x_*)) - \nabla f_m^{\pi_m^i}(x_*)\right)\right\|^2\right] \leq \frac{\gamma^2\omega}{M^2}\sum_{m=1}^{M}\|\nabla f_m^{\pi_m^i}(x_*)\|^2.$$

This upper bound is tight and non-zero in general. Moreover, it is proportional to $\gamma^2$ that creates the term proportional to $\gamma$ in (5) like in the convergence results for QSGD/SGD, while the RR-variance is proportional to $\gamma^2$ in the same bound. Therefore, during the later stages of the convergence Q-RR behaves similarly to QSGD when we decrease the stepsize.

## 2.2 ALGORITHM DIANA-RR

To reduce the additional variance caused by compression, we apply DIANA-style shift sequences (Mishchenko et al., 2019b; Horváth et al., 2019). Thus we obtain DIANA-RR (Algorithm 3). We notice that unlike DIANA, DIANA-RR has $n$ shift-vectors on each node.

**Theorem 2.2.** *Let Assumptions 1, 3, 4 hold and suppose that the stepsizes satisfy $\gamma \leq \min\left\{\frac{\alpha}{2n\widetilde{\mu}}, \frac{1}{\left(1+\frac{6\omega}{M}\right)L_{\max}}\right\}$, and $\alpha \leq \frac{1}{1+\omega}$. Define the following Lyapunov function for every $t \geq 0$*

$$\Psi_{t+1} \overset{\text{def}}{=} \|x_{t+1} - x_\star\|^2 + \frac{4\omega\gamma^2}{\alpha M^2}\sum_{m=1}^{M}\sum_{j=0}^{n-1}(1 - \gamma\mu)^j\|\Delta_{t+1,m}^j\|^2, \tag{6}$$

---

**Algorithm 3** DIANA-RR

---

**Input:** $x_0$ – starting point, $\{h_{0,m}^i\}_{m,i=1,1}^{M,n}$ – initial shift-vectors, $\gamma > 0$ – stepsize, $\alpha > 0$ – stepsize for learning the shifts

1: **for** $t = 0, 1, \ldots, T - 1$ **do**
2:     Receive $x_t$ from the server and set $x_{t,m}^0 = x_t$
3:     Sample random permutation of $[n]$: $\pi_m = (\pi_m^0, \ldots, \pi_m^{n-1})$
4:     **for** $i = 0, 1, \ldots, n - 1$ **do**
5:         **for** $m = 1, 2, \ldots, M$ in parallel **do**
6:             Receive $x_t^i$ from the server, compute and send $\mathcal{Q}\left(\nabla f_m^{\pi_m^i}(x_t^i) - h_{t,m}^{\pi_m^i}\right)$ back
7:             Set $\hat{g}_{t,m}^{\pi_m^i} = h_{t,m}^{\pi_m^i} + \mathcal{Q}\left(\nabla f_m^{\pi_m^i}(x_{t,m}^i) - h_{t,m}^{\pi_m^i}\right)$
8:             Set $h_{t+1,m}^{\pi_m^i} = h_{t,m}^{\pi_m^i} + \alpha\mathcal{Q}\left(\nabla f_m^{\pi_m^i}(x_{t,m}^i) - h_{t,m}^{\pi_m^i}\right)$
9:         **end for**
10:         Compute $x_t^{i+1} = x_t^i - \gamma\frac{1}{M}\sum_{m=1}^M \hat{g}_{t,m}^{\pi_m^i}$ and send $x_t^{i+1}$ to the workers
11:     **end for**
12:     $x_{t+1} = x_t^n$
13: **end for**
**Output:** $x_T$

---

where $\Delta_{t+1,m}^j = h_{t+1,m}^{\pi_m^j} - \nabla f_m^{\pi_m^j}(x_\star)$ *Then, for all $T \geq 0$ the iterates produced by* DIANA-RR *(Algorithm 3) satisfy*

$$\mathbb{E}\left[\Psi_T\right] \leq (1 - \gamma\widetilde{\mu})^{nT}\Psi_0 + \frac{2\gamma^2\sigma_{rad}^2}{\widetilde{\mu}}$$

**Corollary 2.** *Under the same conditions as Theorem 2.2 and for Algorithm 3, there exists stepsizes $\gamma, \alpha > 0$ such that the number of communication rounds $nT$ to find a solution with accuracy $\varepsilon > 0$ is*

$$\widetilde{\mathcal{O}}\left(n(1 + \omega) + \left(1 + \frac{\omega}{M}\right)\frac{L_{\max}}{\widetilde{\mu}} + \frac{\sigma_{rad}}{\sqrt{\varepsilon\widetilde{\mu}^3}}\right).$$

Unlike Q-RR/QSGD/DIANA, DIANA-RR does not have a $\widetilde{\mathcal{O}}(1/\varepsilon)$-term, which makes it superior to Q-RR/QSGD/DIANA for small enough $\varepsilon$. However, the complexity of DIANA-RR has an additive $\widetilde{\mathcal{O}}(n(1 + \omega))$ term arising due to learning the shifts $\{h_{t,m}^i\}_{m\in[M],i\in[n]}$. Nevertheless, this additional term is not the dominating one when $\varepsilon$ is small enough. Next, we elaborate a bit more on the comparison between DIANA and DIANA-R. That is, DIANA has $\widetilde{\mathcal{O}}\left(\left(1 + \frac{\omega}{M}\right)\frac{L_{\max}}{\mu} + \frac{(1+\omega)\sigma_\star^2}{M\mu^2\varepsilon}\right)$ complexity (Gorbunov et al., 2020). Neglecting the differences between $\mu$ and $\widetilde{\mu}$, $L_{\max}$ and $L_{\max}$, we observe a similar relation between DIANA-RR and DIANA as between RR and SGD: instead of the term $\mathcal{O}((1+\omega)\sigma_\star^2/(M\mu^2\varepsilon))$ appearing in the complexity of DIANA, DIANA-RR has $\mathcal{O}(\sigma_{rad}/\sqrt{\varepsilon\widetilde{\mu}^3})$ term much better depending on $\varepsilon$. To the best of our knowledge, our result is the only known one establishing the theoretical superiority of RR to regular SGD in the context of distributed learning with gradient compression. Moreover, when $\omega = 0$ (no compression) we recover the rate of FedRR and when additionally $M = 1$ (single worker) we recover the rate of RR.

## 2.3 ALGORITHM Q-NASTYA

By adding local steps to Q-RR, we can do enable each client to do more local work and only communicate once per epoch rather than at each iteration of every epoch. We follow the framework of the NASTYA algorithm (Malinovsky et al., 2022) and extend it by allowing for quantization, resulting in Q-NASTYA (Algorithm 4).

**Theorem 2.3.** *Let Assumptions 1, 2, 3 hold. Let the stepsizes $\gamma, \eta$ satisfy $0 < \eta \leq \frac{1}{16L_{\max}\left(1 + \frac{\omega}{M}\right)}$, $0 < \gamma \leq \frac{1}{5nL_{\max}}$. Then, for all $T \geq 0$ the iterates produced by* Q-NASTYA *(Algorithm 4) satisfy*

$$\mathbb{E}\left[\|x_T - x_\star\|^2\right] \leq \left(1 - \frac{\eta\mu}{2}\right)^T\|x_0 - x_\star\|^2 + 8\frac{\eta\omega}{\mu M}\zeta_\star^2 + \frac{9}{2}\frac{\gamma^2 nL_{\max}}{\mu}\left((n+1)\zeta_\star^2 + \sigma_\star^2\right).$$

---

**Algorithm 4** Q-NASTYA

---

**Input:** $x_0$ – starting point, $\gamma > 0$ – local stepsize, $\eta > 0$ – global stepsize
1: **for** $t = 0, 1, \ldots, T - 1$ **do**
2:     **for** $m \in [M]$ in parallel **do**
3:         Receive $x_t$ from the server and set $x_{t,m}^0 = x_t$
4:         Sample random permutation of $[n]$: $\pi_m = (\pi_m^0, \ldots, \pi_m^{n-1})$
5:         **for** $i = 0, 1, \ldots, n - 1$ **do**
6:             Set $x_{t,m}^{i+1} = x_{t,m}^i - \gamma \nabla f_m^{\pi_m^i}(x_{t,m}^i)$
7:         **end for**
8:         Compute $g_{t,m} = \frac{1}{\gamma n}\left(x_t - x_{t,m}^n\right)$ and send $\mathcal{Q}_t(g_{t,m})$ to the server
9:     **end for**
10:    Compute $g_t = \frac{1}{M}\sum_{m=1}^M \mathcal{Q}_t(g_{t,m})$
11:    Compute $x_{t+1} = x_t - \eta g_t$ and send $x_{t+1}$ to the workers
12: **end for**
**Output:** $x_T$

---

**Corollary 3.** *Under the same conditions as Theorem 2.3 and for Algorithm 4, there exist stepsizes* $\gamma = \eta/n$ *and* $\eta > 0$ *such that the number of communication rounds* $T$ *to find a solution with accuracy* $\varepsilon > 0$ *is* $\widetilde{\mathcal{O}}\left(\frac{L_{\max}}{\mu}\left(1 + \frac{\omega}{M}\right) + \frac{\omega}{M}\frac{\zeta_\star^2}{\varepsilon\mu^3} + \sqrt{\frac{L_{\max}}{\varepsilon\mu^3}}\sqrt{\zeta_\star^2 + \frac{\sigma_\star^2}{n}}\right)$. *If* $\gamma \to 0$, *one can choose* $\eta > 0$ *such that the above complexity bound improves to* $\widetilde{\mathcal{O}}\left(\frac{L_{\max}}{\mu}\left(1 + \frac{\omega}{M}\right) + \frac{\omega}{M}\frac{\zeta_\star^2}{\varepsilon\mu^3}\right)$.

We emphasize several differences with the known theoretical results. First, the FedCOM method of Haddadpour et al. (2021) was analyzed in the homogeneous setting only, i.e., $f_m(x) = f(x)$ for all $m \in [M]$, which is an unrealistic assumption for FL applications. In contrast, our result holds in the fully heterogeneous case. Next, the analysis of FedPAQ of Reisizadeh et al. (2020) uses a bounded variance assumption, which is also known to be restrictive. Nevertheless, let us compare to their result. Reisizadeh et al. (2020) derive the following complexity for their method: $\widetilde{\mathcal{O}}\left(\frac{L_{\max}}{\mu}\left(1 + \frac{\omega}{M}\right) + \frac{\omega}{M}\frac{\sigma^2}{\mu^2\varepsilon} + \frac{\sigma^2}{M\mu^2\varepsilon}\right)$. This result is inferior to the one we show for Q-NASTYA: when $\omega$ is small, the main term in the complexity bound of FedPAQ is $\widetilde{\mathcal{O}}\left(1/\varepsilon\right)$, while for Q-NASTYA the dominating term is of the order $\widetilde{\mathcal{O}}\left(1/\sqrt{\varepsilon}\right)$ (when $\omega$ and $\varepsilon$ are sufficiently small). We also highlight that FedCRR (Malinovsky and Richtárik, 2022) does not converge if $\omega > M^2\gamma\mu\varepsilon/(2\|x_{*,m}^n\|^2)$, while Q-NASTYA does for any $\omega \geq 0$. Finally, when $\omega = 0$ (no compression) we recover NASTYA as a special case, and using $\gamma = \eta/n$, we recover the rate of FedRR (Mishchenko et al., 2021).

### 2.4 ALGORITHM DIANA-NASTYA

As in the case of Q-RR, the complexity bound for Q-NASTYA includes a $\widetilde{\mathcal{O}}(\omega/\varepsilon)$ term, appearing due to quantization noise. To reduce it, we apply DIANA-style correction sequences, which leads to a new method for which we coin the name DIANA-NASTYA (Algorithm 5).

**Theorem 2.4.** *Let Assumptions 1, 2, 3 hold. Suppose the stepsizes* $\gamma$, $\eta$, $\alpha$ *satisfy* $0 < \gamma \leq \frac{1}{16L_{\max}n}$, $0 < \eta \leq \min\left\{\frac{\alpha}{2\mu}, \frac{1}{16L_{\max}\left(1+\frac{9\omega}{M}\right)}\right\}$, *and* $\alpha \leq \frac{1}{1+\omega}$. *Define the following Lyapunov function:*

$$\Psi_{t+1} \stackrel{\text{def}}{=} \|x_{t+1} - x_\star\|^2 + \frac{8\omega\eta^2}{\alpha M^2}\sum_{m=1}^M \|h_{t+1,m} - h_m^\star\|^2. \tag{7}$$

*Then, for all* $T \geq 0$ *the iterates produced by* DIANA-NASTYA *(Algorithm 5) satisfy*

$$\mathbb{E}\left[\Psi_T\right] \leq \left(1 - \frac{\eta\mu}{2}\right)^T \Psi_0 + \frac{9}{2}\frac{\gamma^2 nL}{\mu}\left((n+1)\zeta_\star^2 + \sigma_\star^2\right). \tag{8}$$

**Corollary 4.** *Under the same conditions as Theorem 2.4 and for Algorithm 5, there exist stepsizes* $\gamma = \eta/n$, $\eta > 0$, $\alpha > 0$ *such that the number of communication rounds* $T$ *to find a solution with*

---

**Algorithm 5** DIANA-NASTYA

---

**Input:** $x_0$ – starting point, $\{h_{0,m}\}_{m=1}^M$ – initial shift-vectors, $\gamma > 0$ – local stepsize, $\eta > 0$ – global stepsize, $\alpha > 0$ – stepsize for learning the shifts
1: **for** $t = 0, 1, \ldots, T-1$ **do**
2:     **for** $m = 1, \ldots, M$ in parallel **do**
3:         Receive $x_t$ from the server and set $x_{t,m}^0 = x_t$
4:         Sample random permutation of $[n]$: $\pi_m = (\pi_m^0, \ldots, \pi_m^{n-1})$
5:         **for** $i = 0, 1, \ldots, n-1$ **do**
6:             Set $x_{t,m}^{i+1} = x_{t,m}^i - \gamma \nabla f_m^{\pi_m^i}(x_{t,m}^i)$
7:         **end for**
8:         Compute $g_{t,m} = \frac{1}{\gamma n}\left(x_t - x_{t,m}^n\right)$ and send $\mathcal{Q}_t\left(g_{t,m} - h_{t,m}\right)$ to the server
9:         Set $h_{t+1,m} = h_{t,m} + \alpha \mathcal{Q}_t\left(g_{t,m} - h_{t,m}\right)$
10:        Set $\hat{g}_{t,m} = h_{t,m} + \mathcal{Q}_t\left(g_{t,m} - h_{t,m}\right)$
11:     **end for**
12:    $h_{t+1} = \frac{1}{M}\sum_{m=1}^M h_{t+1,m} = h_t + \frac{\alpha}{M}\sum_{m=1}^M \mathcal{Q}_t\left(g_{t,m} - h_{t,m}\right)$
13:    $\hat{g}_t = \frac{1}{M}\sum_{m=1}^M \hat{g}_{t,m} = h_t + \frac{1}{M}\sum_{m=1}^M \mathcal{Q}_t\left(g_{t,m} - h_{t,m}\right)$
14:    $x_{t+1} = x_t - \eta\hat{g}_t$
15: **end for**
**Output:** $x_T$

---

*accuracy $\varepsilon > 0$ is $\widetilde{\mathcal{O}}\left(\omega + \frac{L_{\max}}{\mu}\left(1 + \frac{\omega}{M}\right) + \sqrt{\frac{L_{\max}}{\varepsilon\mu^3}}\sqrt{\zeta_\star^2 + \frac{\sigma_\star^2}{n}}\right)$. If $\gamma \to 0$, one can choose $\eta > 0$ such that the above complexity bound improves to $\widetilde{\mathcal{O}}\left(\omega + \frac{L_{\max}}{\mu}\left(1 + \frac{\omega}{M}\right)\right)$.*

In contrast to Q-NASTYA, DIANA-NASTYA does not suffer from the $\widetilde{\mathcal{O}}(1/\varepsilon)$ term in the complexity bound. This shows the superiority of DIANA-NASTYA to Q-NASTYA. Next, FedCRR-VR (Malinovsky and Richtárik, 2022) has the rate $\widetilde{\mathcal{O}}\left(\frac{(\omega+1)\left(1-\frac{1}{\kappa}\right)^n}{\left(1-\left(1-\frac{1}{\kappa}\right)^n\right)^2} + \frac{\sqrt{\kappa}(\zeta_\star + \sigma_\star)}{\mu\sqrt{\varepsilon}}\right)$, which depends on $\widetilde{\mathcal{O}}\left(1/\sqrt{\varepsilon}\right)$. However, the first term is close to $\widetilde{\mathcal{O}}\left((\omega+1)\kappa^2\right)$ for a large condition number. FedCRR-VR-2 utilizes variance reduction technique from Malinovsky et al. (2021) and it allows to get rid of permutation variance. This method has $\widetilde{\mathcal{O}}\left(\frac{(\omega+1)\left(1-\frac{1}{\kappa\sqrt{\kappa n}}\right)^{\frac{n}{2}}}{\left(1-\left(1-\frac{1}{\kappa\sqrt{\kappa n}}\right)^{\frac{n}{2}}\right)^2} + \frac{\sqrt{\kappa}\zeta_\star}{\mu\sqrt{\varepsilon}}\right)$ complexity, but it requires additional assumption on number of functions $n$ and thus not directly comparable with our result. Note that if we have no compression ($\omega = 0$), DIANA-NASTYA recovers rate of NASTYA.

## 3 EXPERIMENTS

We evaluated our methods for solving logistic regression problems and training neural networks in three parts: (i) Comparison of the proposed non-local methods with existing baselines; (ii) Comparison of the proposed local methods with existing baselines; (iii) Comparison of the proposed non-local methods in training ResNet-18 on CIFAR10.

In our first experiment (refer to Figure 1a), we compared non-local algorithms named Q-RR and DIANA-RR with their corresponding classical baselines, QSGD (Alistarh et al., 2017) and DIANA (Mishchenko et al., 2019b), that use a with-replacement mini-batch SGD estimator. In the second experiment (see Figure 1b), we assessed local methods named Q-NASTYA and DIANA-NASTYA, along with other baselines, FedCOM (Haddadpour et al., 2021), and FedPAQ (Reisizadeh et al., 2020).

For the third set of experiments, we focused on training the ResNet-18 model on the CIFAR10 dataset Krizhevsky and Hinton (2009). We tested Q-RR, QSGD, DIANA, and DIANA-RR in the distributed training of ResNet-18 on CIFAR10 (see Figure 2). These experiments were conducted using the FL_PyTorch simulator (Burlachenko et al., 2021).

To adhere to space limitations, we have provided the experimental results and details in Appendix A.

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

# A EXPERIMENTAL DETAILS

In this section, we provide missing details on the experimental setting from Section 3. The codes are provided in the following anonymous repository: https://anonymous.4open.science/r/diana_rr-[]B0A5.

## A.1 LOGISTIC REGRESSION

To confirm our theoretical results we conducted several numerical experiments on binary classification problem with L2 regularized logistic regression of the form

$$\min_{x \in \mathbb{R}^d} \left[ f(x) \stackrel{\text{def}}{=} \frac{1}{M} \sum_{m=1}^{M} \frac{1}{n_m} \sum_{i=1}^{n_m} f_{m,i} \right], \tag{9}$$

where $f_{m,i} \stackrel{\text{def}}{=} \log\left(1 + \exp(-y_{mi} a_{mi}^\top x)\right) + \lambda \|x\|_2^2$ $(a_{mi}, y_{mi}) \in \mathbb{R}^d \times \in \{-1, 1\}, i = 1, \dots, n_m$ are the training data samples stored on machines $m = 1, \dots, M$, and $\lambda > 0$ is a regularization parameter. In all experiments, for each method, we used the largest stepsize allowed by its theory multiplied by some individually tuned constant multiplier. For better parallelism, each worker $m$ uses mini-batches of size $\approx 0.1 n_m$. In all algorithms, as a compression operator $\mathcal{Q}$, we use Rand-$k$ (Beznosikov et al., 2020) with fixed compression ratio $k/d \approx 0.02$, where $d$ is the number of features in the dataset.

**Hardware and Software.** All algorithms were written in Python 3.8. We used three different CPU cluster node types:

1. AMD EPYC 7702 64-Core;
2. Intel(R) Xeon(R) Gold 6148 CPU @ 2.40GHz;
3. Intel(R) Xeon(R) Gold 6248 CPU @ 2.50GHz.

**Datasets.** The datasets were taken from open LibSVM library Chang and Lin (2011), sorted in ascending order of labels, and equally split among 20 machines \clients\workers. The remaining part of size $N - 20 \cdot \lfloor N/20 \rfloor$ was assigned to the last worker, where $N = \sum_{m=1}^{M} n_m$ is the total size of the dataset. A summary of the splitting and the data samples distribution between clients can be found in Tables 1, 2, 3, 4.

Table 1: Summary of the datasets and splitting of the data samples among clients.

| Dataset | $M$ | $N$ (dataset size) | $d$ (# of features) | $n_m$ (# of datasamples per client) |
|---|---|---|---|---|
| mushrooms | 20 | 8120 | 112 | 406 |
| w8a | 20 | 49749 | 300 | 2487 |
| a9a | 20 | 32560 | 123 | 1628 |

Table 2: Partition of the mushrooms dataset among clients.

| Client's № | # of datasamples of class "-1" | # of datasamples of class "+1" |
|---|---|---|
| $1 - 9$ | 406 | 0 |
| 10 | 262 | 144 |
| $11 - 19$ | 0 | 406 |
| 20 | 0 | 410 |

**Hyperparameters.** Regularization parameter $\lambda$ was chosen individually for each dataset to guarantee the condition number $L/\mu$ to be approximately $10^4$, where $L$ and $\mu$ are the smoothness and strong-convexity constants of function $f$. For the chosen logistic regression problem of the form

Table 3: Partition of the `w8a` dataset among clients.

| Client's № | # of datasamples of class "-1" | # of datasamples of class "+1" |
|---|---|---|
| $1-19$ | 2487 | 0 |
| 20 | 1017 | 1479 |

Table 4: Partition of the `a9a` dataset among clients.

| Client's № | # of datasamples of class "-1" | # of datasamples of class "+1" |
|---|---|---|
| $1-14$ | 1628 | 0 |
| 15 | 1328 | 300 |
| $16-19$ | 0 | 1628 |
| 20 | 0 | 1629 |

(9), smoothness and strong convexity constants $L$, $L_m$, $L_{i,m}$, $\mu$, $\widetilde{\mu}$ of functions $f$, $f_m$ and $f_m^i$ were computed explicitly as

$$
\begin{aligned}
L &= \lambda_{\max}\left(\frac{1}{M}\sum_{m=1}^{M}\frac{1}{4n_m}\mathbf{A}_m^\top\mathbf{A}_m + 2\lambda\mathbf{I}\right) \\
L_m &= \lambda_{\max}\left(\frac{1}{4n_m}\mathbf{A}_m^\top\mathbf{A}_m + 2\lambda\mathbf{I}\right) \\
L_{i,m} &= \lambda_{\max}\left(\frac{1}{4}a_{mi}a_{mi}^\top + 2\lambda\mathbf{I}\right) \\
\mu &= 2\lambda \\
\widetilde{\mu} &= 2\lambda,
\end{aligned}
$$

where $\mathbf{A}_m$ is the dataset associated with client $m$, and $a_{mi}$ is the $i$-th row of data matrix $\mathbf{A}_m$. In general, the fact that $f$ is $L$-smooth with

$$
L \le \frac{1}{M}\sum_{m=1}^{M}\frac{1}{n_m}\sum_{i=1}^{n_m}L_{i,m}
$$

follows from the $L_{i,m}$-smoothness of $f_m^i$ (see Assumption 3).

In all algorithms, as a compression operator $\mathcal{Q}$, we use Rand-$k$ as a canonical example of unbiased compressor with relatively bounded variance, and fix the compression parameter $k = \lfloor 0.02d \rfloor$, where $d$ is the number of features in the dataset.

In addition, in all algorithms, for all clients $m = 1, \ldots, M$, we set the batch size for the SGD estimator to be $b_m = \lfloor 0.1n_m \rfloor$, where $n_m$ is the size of the local dataset.

The summary of the values $L$, $L_m$, $L_{i,m}$ $L_{\max}$, $\mu$, $b_m$ and $k$ for each dataset can be found in Table 5.

Table 5: Summary of the hyperparameters.

| Dataset | $L$ | $L_{\max}$ | $\mu$ | $\lambda$ | $k$ | $b_m$ (batchsize) |
|---|---|---|---|---|---|---|
| mushrooms | 2.59 | 5.25 | $2.58 \cdot 10^{-4}$ | $1.29 \cdot 10^{-4}$ | 2 | 40 |
| w8a | 0.66 | 28.5 | $6.6 \cdot 10^{-5}$ | $3.3 \cdot 10^{-5}$ | 6 | 248 |
| a9a | 1.57 | 3.5 | $1.57 \cdot 10^{-4}$ | $7.85 \cdot 10^{-5}$ | 2 | 162 |

In all experiments, we follow constant stepsize strategy within the whole iteration procedure. For each method, we set the largest possible stepsize predicted by its theory multiplied by some individually tuned constant multiplier. For a more detailed explanation of the tuning routine, see Sections A.1.1 and A.1.2.

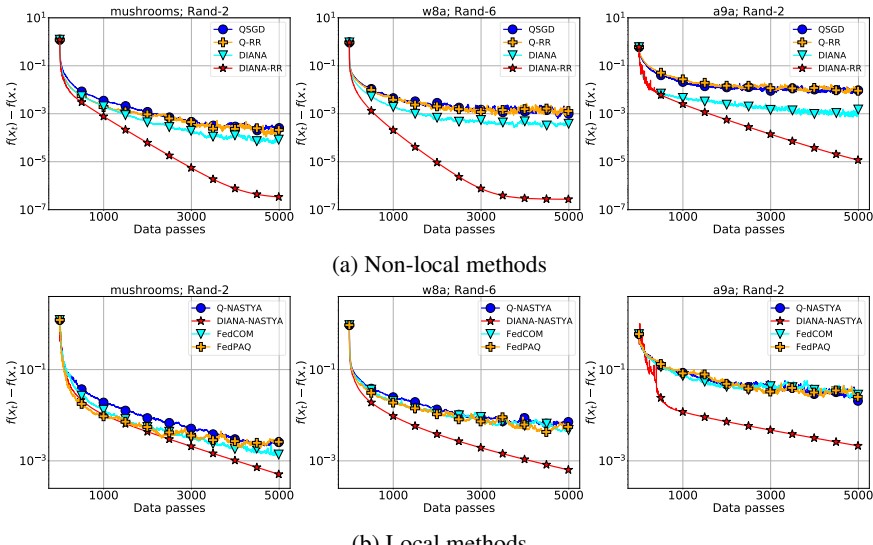

(a) Non-local methods

(b) Local methods

Figure 1: The comparison of the four proposed methods (Q-NASTYA, DIANA-NASTYA, Q-RR, DIANA-RR) and existing baselines (FedCOM, FedPAQ) with tuned stepsizes and Rand-$k$ compressor.

**SGD implementation.** We considered two approaches to minibatching: random reshuffling and with-replacement sampling. In the first, all clients $m = 1, \ldots, M$ independently permute their local datasets and pass through them within the next subsequent $\lfloor \frac{n_m}{b_m} \rfloor$ steps. In our implementations of Q-RR, Q-NASTYA and DIANA-NASTYA, all clients permuted their datasets in the beginning of every new epoch, whereas for the DIANA-RR method they do so only once in the beginning of the iteration procedure. Second approach of minibatching is called with-replacement sampling, and it requires every client to draw $b_m$ data samples from the local dataset uniformly at random. We used this strategy in the baseline algorithms (QSGD, DIANA, FedCOM and FedPAQ) we compared our proposed methods to.

**Experimental setup.** To compare the performance of methods within the whole optimization process, we track the functional suboptimality metric $f(x_t) - f(x_\star)$ that was recomputed after each epoch. For each dataset, the value $f(x_\star)$ was computed once at the preprocessing stage with $10^{-16}$ tolerance via conjugate gradient method. We terminate our algorithms after performing 5000 epochs.

### A.1.1 EXPERIMENT 1: COMPARISON OF THE PROPOSED NON-LOCAL METHODS WITH EXISTING BASELINES

In our first experiment (see Figure 1a), we compare Q-RR and DIANA-RR with corresponding classical baselines (QSGD (Alistarh et al., 2017), DIANA (Mishchenko et al., 2019b)) that use a with-replacement mini-batch SGD estimator. Figure 1a illustrates that Q-RR experiences similar behavior as QSGD both losing in speed to DIANA method in all considered datasets. However, DIANA-RR shows the best rate among all considered non-local methods, efficiently reducing the variance, and achieving the lowest functional sub-optimality tolerance. The results observed in numerical experiments are in perfect correspondence with the derived theory.

For each of the considered non-local methods, we take the stepsize as the largest one predicted by the theory premultiplied by the individually tuned constant factor from the set $\{0.000975, 0.00195, 0.0039, 0.0078, 0.0156, 0.0312, 0.0625, 0.125, 0.25, 0.5, 1, 2, 4, 8, 16, 32, 64, 128, 256, 512, 1024, 2048, 4096\}$.

Therefore, for each local method on every dataset, we performed 20 launches to find the stepsize multiplier showing the best convergence behavior (the fastest reaching the lowest possible level of functional suboptimality $f(x_t) - f(x_\star)$).

Theoretical stepsizes for methods Q-RR and DIANA-RR are provided by the Theorems 2.1 and 2.2, whereas stepsizes for QSGD and DIANA were taken from the paper Gorbunov et al. (2020).

### A.1.2 Experiment 2: Comparison of the Proposed Local Methods with Existing Baselines

The second experiment shows that DIANA-based method can significantly outperform in practice when one applies it to local methods as well. In particular, whereas Q-NASTYA shows comparative behavior as existing methods FedCOM (Haddadpour et al., 2021), FedPAQ (Reisizadeh et al., 2020) in all considered datasets, DIANA-NASTYA noticeably outperforms other methods.

In this set of experiments, we tuned stepsizes similarly to the non-local methods. However, for algorithms Q-NASTYA, DIANA-NASTYA, and FedCOM we needed to independently adjust the client and server stepsizes, leading to a more extensive tunning routine.

As before, for each local method on every dataset, tuned client and server stepsizes are defined by the theoretical one and adjusted constant multiplier. Theoretical stepsizes for methods Q-NASTYA and DIANA-NASTYA are given by the Theorems 2.3 and 2.4, whereas FedCOM and FedPAQ stepsizes were taken from the papers by Haddadpour et al. (2021) and Reisizadeh et al. (2020) respectively. We now list all the considered multipliers of client and server stepsizes for every method (i.e. $\gamma$ and $\eta$ respectively):

- Q-NASTYA:
  - Multipliers for $\gamma$ : $\{0.000975, 0.00195, 0.0039, 0.0078, 0.0156, 0.0312, 0.0625, 0.125, 0.25, 0.5, 1, 2, 4, 8, 16, 32, 64, 128\}$;
  - Multipliers for $\eta$ : $\{0.0039, 0.0078, 0.0156, 0.0312, 0.0625, 0.125, 0.25, 0.5, 1, 2, 4, 8, 16, 32, 64, 128\}$.
- DIANA-NASTYA:
  - Multipliers for $\gamma$ and $\eta$ : $\{0.000975, 0.00195, 0.0039, 0.0078, 0.0156, 0.0312, 0.0625, 0.125, 0.25, 0.5, 1, 2, 4, 8, 16, 32, 64, 128\}$;
- FedCOM:
  - Multipliers for $\gamma$ : $\{0.0312, 0.0625, 0.125, 0.25, 0.5, 1, 2, 4, 8, 16, 32, 64, 128, 256, 512, 1024, 2048, 4096, 8192, 16384, 32768 \}$;
  - Multipliers for $\eta$ : $\{0.000975, 0.00195, 0.0039, 0.0078, 0.0156, 0.0312, 0.0625, 0.125, 0.25, 0.5, 1, 2, 4, 8, 16, 32, 64, 128\}$.
- FedPAQ:
  - Multipliers for $\gamma$ : $\{0.00195, 0.0039, 0.0078, 0.0156, 0.0312, 0.0625, 0.125, 0.25, 0.5, 1, 2, 4, 8, 16, 32, 64, 128, 256, 512, 1024, 2048, 4096, 8192, 16384, 32768, 65536, 131072, 262144, 524288, 1048576 \}$.

For example, to find the best pair $(\gamma, \eta)$ for FedCOM method on each dataset, we performed 378 launches. A similar subroutine was executed for all algorithms on all datasets independently.

### A.2 Training Deep Neural Network model: ResNet-18 on CIFAR-10

Since random reshuffling is a very popular technique in training neural networks, it is natural to test the proposed methods on such problems. Therefore, in the second set of experiments, we consider training `ResNet-18` (He et al., 2016) model on the `CIFAR10` dataset Krizhevsky and Hinton (2009). To conduct these experiments we use `FL_PyTorch` simulator (Burlachenko et al., 2021).

The main goal of this experiment is to verify the phenomenon observed in Experiment 1 on the training of a deep neural network. That is, we tested Q-RR, QSGD, DIANA, and DIANA-RR in the distributed training of `ResNet-18` on `CIFAR10`, see Figure 2. As in the logistic regression experiments, we observe that (i) Q-RR and QSGD behave similarly and (ii) DIANA-RR outperforms DIANA.

To illustrate the behavior of the proposed methods in training Deep Neural Networks (DNN), we consider the `ResNet-18` (He et al., 2016) model. This model is used for image classification,

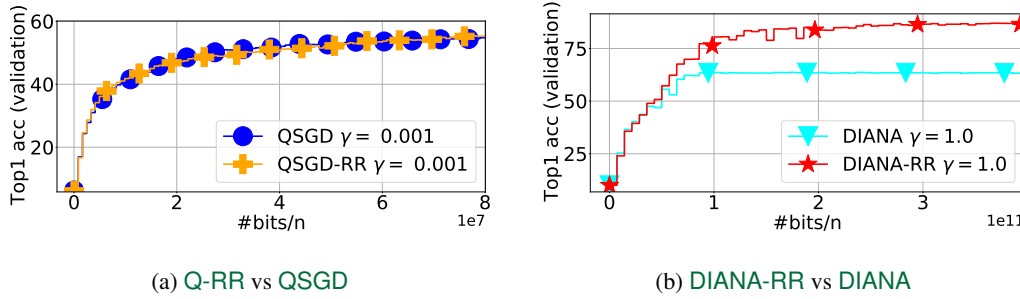

(a) Q-RR vs QSGD

(b) DIANA-RR vs DIANA

Figure 2: The comparison of Q-RR, QSGD, DIANA, and DIANA-RR on the task of training `ResNet-18` on `CIFAR-10` with $n = 10$ workers. Top-1 accuracy on test set is reported. Stepsizes were tuned and workers used Rand-$k$ compressor with $k/d \approx 0.05$.

feature extraction for image segmentation, object detection, image embedding, and image captioning. We train all layers of `ResNet-18` model meaning that the dimension of the optimization problem equals $d = 11,173,962$. During the training, the `ResNet-18` model normalizes layer inputs via exploiting 20 Batch Normalization (Ioffe and Szegedy, 2015) layers that are applied directly before nonlinearity in the computation graph of this model. Batch normalization (BN) layers add 9600 trainable parameters to the model. Besides trainable parameters, a BN layer has its internal state that is used for computing the running mean and variance of inputs due to its own specific regime of working. We use *He* initialization (He et al., 2015).

### A.2.1 Computing Environment

We performed numerical experiments on a server-grade machine running Ubuntu 18.04 and Linux Kernel v5.4.0, equipped with 16-cores (2 sockets by 16 cores per socket) 3.3 GHz Intel Xeon, and four NVIDIA A100 GPU with 40GB of GPU memory. The distributed environment is simulated in Python 3.9 via using the software suite `FL_PyTorch` (Burlachenko et al., 2021) that serves for carrying complex Federate Learning experiments. `FL_PyTorch` allowed us to simulate the distributed environment in the local machine. Besides storing trainable parameters per client, this simulator stores all not trainable parameters including BN statistics per client.

### A.2.2 Loss Function

Training of `ResNet-18` can be formalized as problem (1) with the following choice of $f_m^i$

$$f_m(x) = \frac{1}{|n_m|} \sum_{j=1}^{|n_m|} CE(b^{(j)}, g(a^{(j)}, x)), \tag{10}$$

where $CE(p, q) \stackrel{\text{def}}{=} -\sum_{k=1}^{\#\text{classes}} p_i \cdot \log(q_i)$ with agreement $0 \cdot \log(0) = 0$ is a standard cross-entropy loss, function $g : \mathbb{R}^{28 \times 28} \times \mathbb{R}^d \to [0, 1]^{\#\text{classes}}$ is a neural network taking image $a^{(j)}$ and vector of parameters $x$ as an input and returning a vector in probability simplex, and $n_m$ is the size of the dataset on worker $m$.

### A.2.3 Dataset and Metric

In our experiments, we used `CIFAR10` dataset Krizhevsky and Hinton (2009). The dataset consists of input variables $a_i \in \mathbb{R}^{28 \times 28 \times 3}$, and response variables $b_i \in \{0, 1\}^{10}$ and is used for training 10-way classification. The sizes of training and validation set are $5 \times 10^4$ and $10^4$ respectively. The training set is partitioned heterogeneously across 10 clients. To measure the performance, we evaluate the loss function value $f(x)$, norm of the gradient $\|\nabla f(x)\|_2$ and the Top-1 accuracy of the obtained model as a function of passed epochs and the normalized number of bits sent from clients to the server.

### A.2.4 TUNING PROCESS

In this set of experiments, we tested QSGD (Alistarh et al., 2017), Q-RR (Algorithm 2), DI-ANA (Mishchenko et al., 2019a) and DIANA-RR (Algorithm 3) algorithms. For all algorithms, we tuned the strategy $\in \{A, B, C\}$ of decaying stepsize model via selecting the best in terms of the norm of the full gradient on the train set in the final iterate produced after 20000 rounds. The stepsize policies are described below.

A. Stepsizes decaying as inverse square root of the number epochs

$$
\gamma_e = \begin{cases} \gamma_{init} \cdot \dfrac{1}{\sqrt{e - s + 1}}, & \text{if } e \geq s, \\ \gamma_{init}, & \text{if } e < s, \end{cases}
$$

where $\gamma_e$ denotes the stepsize used during epoch $e + 1$, $s$ is a fixed shift.

B. Stepsizes decaying as inverse of number epochs

$$
\gamma = \begin{cases} \gamma_{init} \cdot \dfrac{1}{e - s + 1}, & \text{if } e \geq s, \\ \gamma_{init}, & \text{if } e < s. \end{cases}
$$

C. Fixed stepsize

$$
\gamma = \gamma_{init}.
$$

We say that the algorithm passed $e$ epochs if the total number of computed gradient oracles lies between $e \sum_{m=1}^{M} n_m$ and $(e + 1) \sum_{m=1}^{M} n_m$. For each algorithm the used stepsize $\gamma_{init}$ and shift parameter $s$ were tuned via selecting from the following sets:

$$
\gamma_{init} \in \gamma_{set} \stackrel{\text{def}}{=} \{4.0, 3.75, 3.00, 2.5, 2.00, 1.25, 1.0, 0.75, 0.5, 0.25, \\ 0.2, 0.1, 0.06, 0.03, 0.01, 0.003, 0.001, 0.0006\}.
$$

$$
s \in s_{set} \stackrel{\text{def}}{=} \{50, 100, 200, 500, 1000\}.
$$

In all tested methods, clients independently apply Rand-$k$ compression with carnality $k = \lfloor 0.05d \rfloor$. Computation for all gradient oracles is carried out in single precision float (fp32) arithmetic.

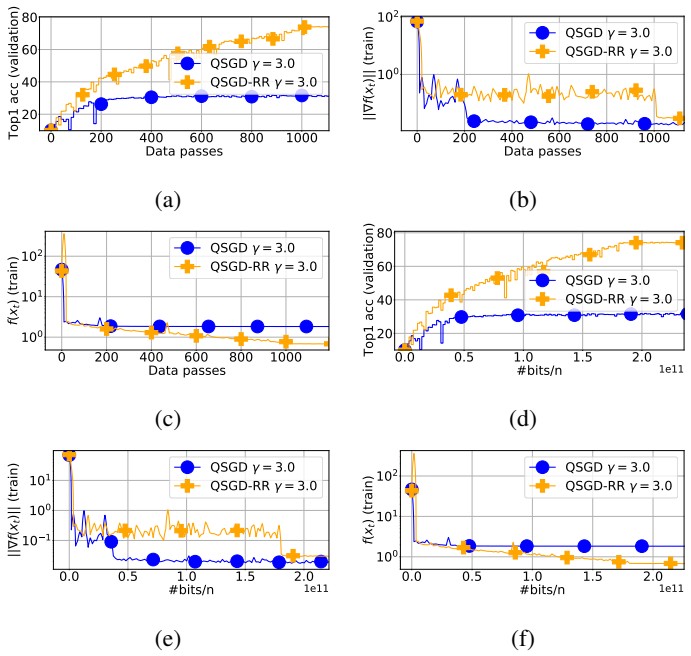

Figure 3: Comparison of QSGD and Q-RR in the training of ResNet-18 on CIFAR-10, with $n = 10$ workers. Here (a) and (d) show Top-1 accuracy on test set, (b) and (e) – norm of full gradient on the train set, (c) and (f) – loss function value on the train set. Stepsizes and decay shift has been tuned from $s_{set}$ and $\gamma_{set}$ based on minimum achievable value of loss function on the train set.

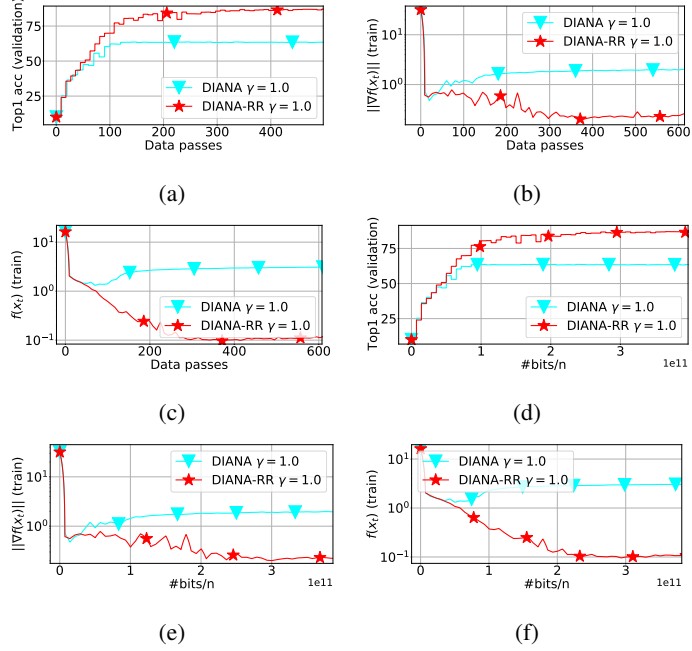

Figure 4: Comparison of DIANA and DIANA-RR in the training of ResNet-18 on CIFAR-10, with $n = 10$ workers. Here (a) and (d) show Top-1 accuracy on test set, (b) and (e) – norm of full gradient on the train set, (c) and (f) – loss function value on the train set. Stepsizes and decay shift has been tuned from $s_{set}$ and $\gamma_{set}$ based on minimum achievable value of loss function on the train set. For both algorithms stepsize is fixed. For both algorithms stepsize is decaying according to srategy $B$.

A.2.5   OPTIMIZATION-BASED FINE-TUNING FOR PRETRAINED RESNET−18.

In this setting, we trained ResNet−18 image classification in a distributed way across $n = 10$ clients. In this experiment, we have trained only the last linear layer.

Next, we have turned off batch normalization. Turning off batch normalization implies that the computation graph of NN $g(a, x)$ with weights of NN denoted as $x$ is a deterministic function and does not include any internal state.

The loss function is a standard cross-entropy loss augmented with extra $\ell_2$-regularization $\alpha\|x\|^2/2$ with $\alpha = 0.0001$. Initially used weights of NN are pretrained parameters after training the model on ImageNet.

The dataset distribution across clients has been set in a heterogeneous manner via presorting dataset $D$ by label class and after this, it was split across 10 clients.

The comparison of stepsizes policies used in QSGD and Q-RR is presented in Figure 6. The behavior of the algorithms with best tuned step sizes is presented in Figure 5. These results demonstrate that in this setting there is no real benefit of using Q-RR in comparison to QSGD.

A.2.6   EXPERIMENTS

The comparison of QSGD and Q-RR is presented in Figure 3. In particular, Figures 3b and 3e show that in terms of the convergence to stationary points both algorithms exhibit similar behavior. However, Q-RR has better generalization and in fact, converges to the better loss function value. This experiment demonstrates that Q-RR with manually tuned stepsize can be better compared to QSGD in terms of the final quality of obtained Deep Learning model. For QSGD the tuned meta parameters are: $\gamma_{init} = 3.0, s = 200$, strategy $= B$. For QSGD-RR tuned meta parameters are: $\gamma_{init} = 3.0$, $s = 1000$, strategy $= B$.

The results of comparison of DIANA and DIANA-RR are presented in Figure 4. For DIANA the tuned meta parameters are: $\gamma_{init} = 1.0, s = 0$, strategy $= C$ and for DIANA-RR tuned meta parameters are: $\gamma_{init} = 1.0$, $s = 0$, strategy $= C$. These results show that DIANA-RR outperforms DIANA in terms of the all reported metrics.

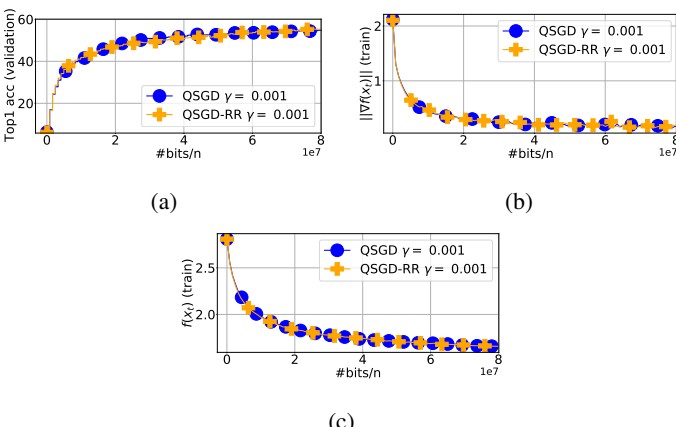

Figure 5: Comparison of QSGD and Q-RR in the training of the last linear layer of ResNet−18 on CIFAR−10, with $n = 10$ workers. Here (a) shows Top-1 accuracy on test set, (b) – norm of full gradient on the train set, (c) – loss function value on the train set. Stepsizes and decay shift has been tuned from $s_{set}$ and $\gamma_{set}$ based on minimum achievable value of loss function on the train set. Both algorithms used fixed stepsize during training.

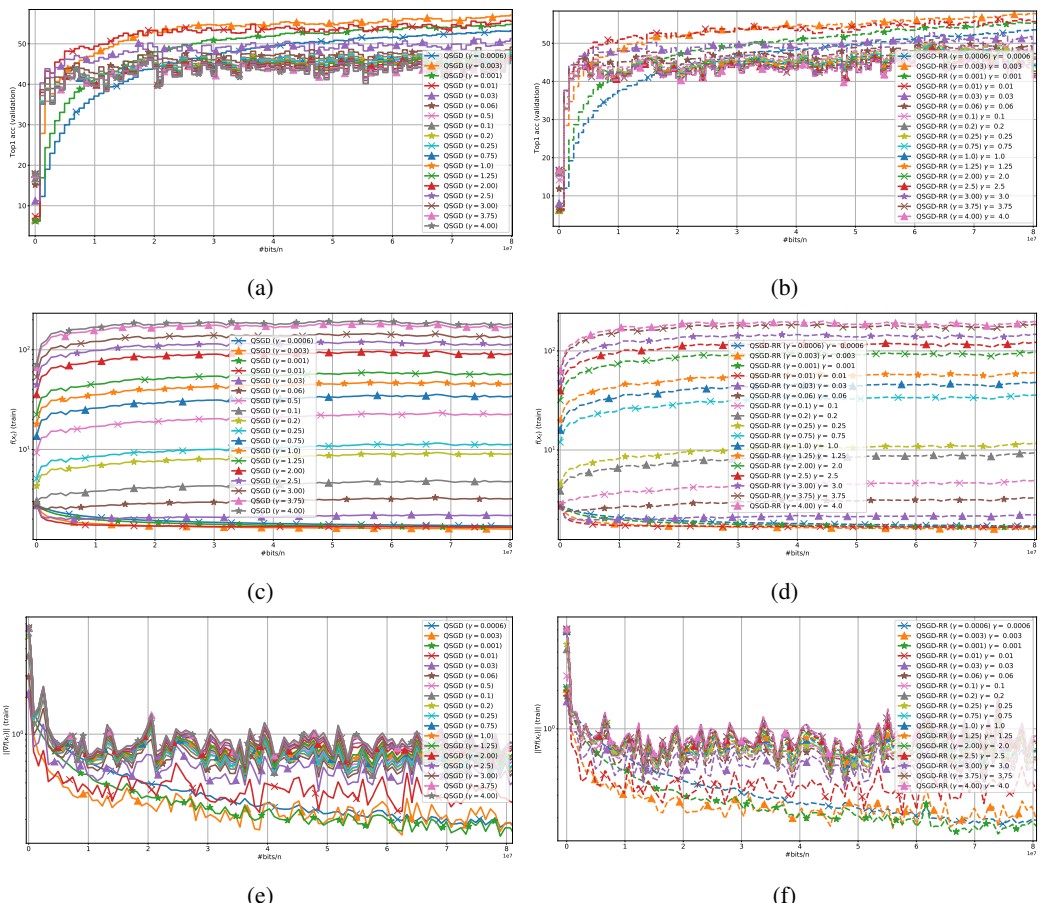

(a)          (b)

(c)          (d)

(e)          (f)

Figure 6: Comparison of QSGD and Q-RR in the training of the last linear layer of `ResNet-18` on `CIFAR-10`, with $n = 10$ workers. Here (a) and (b) show Top-1 accuracy on test set, (c) and (d) – loss function value on the train set, (e) and (f) – norm of full gradient on the train set. Stepsizes and decay shift has been tuned from $s_{set}$ and $\gamma_{set}$ based on minimum achievable value of loss function on the train set. During training stepsize was fixed. Batch Normalization was turned off.

# B MISSING PROOFS FOR Q-RR

In the main part of the paper, we intoduce Assumptions 3 and 4 for the analysis of Q-RR and DIANA-RR. These assumptions can be refined as follows.

**Assumption 5.** *Function* $f^{\pi^i} = \frac{1}{M}\sum_{i=1}^{M} f_m^{\pi_m^i} : \mathbb{R}^d \to \mathbb{R}$ *is* $\widetilde{L}$*-smooth for all sets of permutations* $\pi = (\pi_1, \ldots, \pi_m)$ *from* $[n]$ *and all* $i \in [n]$*, i.e.,*

$$\max_{i \in [n], \pi} \|\nabla f^{\pi^i}(x) - \nabla f^{\pi^i}(y)\| \le \widetilde{L}\|x - y\| \quad \forall x, y \in \mathbb{R}^d.$$

**Assumption 6.** *Function* $f^{\pi^i} = \frac{1}{M}\sum_{i=1}^{M} f_m^{\pi_m^i} : \mathbb{R}^d \to \mathbb{R}$ *is* $\widetilde{\mu}$*-strongly convex for all sets of permutations* $\pi = (\pi_1, \ldots, \pi_m)$ *from* $[n]$ *and all* $i \in [n]$*, i.e.,*

$$\min_{i \in [n], \pi} \left\{ f^{\pi^i}(x) - f^{\pi^i}(y) - \langle \nabla f^{i,\pi}(y), x - y \rangle \right\} \ge \frac{\widetilde{\mu}}{2}\|x - y\|^2 \quad \forall x, y \in \mathbb{R}^d.$$

*Moreover, functions* $f_1^i, f_2^i, \ldots, f_M^i : \mathbb{R}^d \to \mathbb{R}$ *are convex for all* $i = 1, \ldots, n$.

We notice that Assumptions 3 and 4 imply Assumptions 5 and 6. In the proofs of the results for Q-RR and DIANA-RR, we use Assumptions 5 in addition to Assumptions 3 and we use Assumption 6 instead of Assumption 4.

## B.1 PROOF OF THEOREM 2.1

For convenience, we restate the theorem below.

**Theorem B.1** (Theorem 2.1). *Let Assumptions 1, 3, 5, 6 hold and* $0 < \gamma \le \frac{1}{\widetilde{L} + 2\frac{\omega}{M}L_{\max}}$. *Then, for all* $T \ge 0$ *the iterates produced by* Q-RR *satisfy*

$$\mathbb{E}\|x_T - x_\star\|^2 \le (1 - \gamma\widetilde{\mu})^{nT}\|x_0 - x_\star\|^2 + \frac{2\gamma^2\sigma_{rad}^2}{\widetilde{\mu}} + \frac{2\gamma\omega}{\widetilde{\mu}M}\left(\zeta_\star^2 + \sigma_\star^2\right),$$

*where* $\zeta_\star^2 = \frac{1}{M}\sum_{m=1}^{M} \|\nabla f_m(x_\star)\|^2$, *and* $\sigma_\star^2 = \frac{1}{Mn}\sum_{m=1}^{M}\sum_{i=1}^{n} \|\nabla f_m^i(x^\star) - \nabla f_m(x^\star)\|^2$.

*Proof.* Using $x_\star^{i+1} = x_\star^i - \frac{\gamma}{M}\sum_{m=1}^{M} \nabla f_m^{\pi_m^i}(x_\star)$ and line 7 of Algorithm 2, we get

$$\|x_t^{i+1} - x_\star^{i+1}\|^2 = \left\| x_t^i - x_\star^i - \gamma\frac{1}{M}\sum_{m=1}^{M}\left(\mathcal{Q}\left(\nabla f_m^{\pi_m^i}(x_t^i)\right) - \nabla f_m^{\pi_m^i}(x_\star)\right) \right\|^2$$

$$= \|x_t^i - x_\star^i\|^2 - 2\gamma\left\langle \frac{1}{M}\sum_{m=1}^{M}\left(\mathcal{Q}\left(\nabla f_m^{\pi_m^i}(x_t^i)\right) - \nabla f_m^{\pi_m^i}(x_\star)\right), x_t^i - x_\star^i \right\rangle$$

$$+ \gamma^2 \left\| \frac{1}{M}\sum_{m=1}^{M}\left(\mathcal{Q}\left(\nabla f_m^{\pi_m^i}(x_t^i)\right) - \nabla f_m^{\pi_m^i}(x_\star)\right) \right\|^2.$$

Taking the expectation w.r.t. $\mathcal{Q}$, we obtain

$$\mathbb{E}_{\mathcal{Q}}\left[\|x_t^{i+1} - x_\star^{i+1}\|^2\right] = \|x_t^i - x_\star^i\|^2 - 2\gamma\left\langle \frac{1}{M}\sum_{m=1}^{M}\left(\nabla f_m^{\pi_m^i}(x_t^i) - \nabla f_m^{\pi_m^i}(x_\star)\right), x_t^i - x_\star^i \right\rangle$$

$$+ \gamma^2\mathbb{E}_{\mathcal{Q}}\left[\left\| \frac{1}{M}\sum_{m=1}^{M}\left(\mathcal{Q}\left(\nabla f_m^{\pi_m^i}(x_t^i)\right) - \nabla f_m^{\pi_m^i}(x_\star)\right) \right\|^2\right].$$

In view of Assumption 1 and $\mathbb{E}_\xi\|\xi - c\|^2 = \mathbb{E}_\xi\|\xi - \mathbb{E}_\xi\xi\|^2 + \|\mathbb{E}_\xi\xi - c\|^2$, we have

$$
\begin{aligned}
\mathbb{E}_{\mathcal{Q}}\left[\|x_t^{i+1} - x_\star^{i+1}\|^2\right] &= \|x_t^i - x_\star^i\|^2 - \frac{2\gamma}{M}\sum_{m=1}^{M}\left\langle \nabla f_m^{\pi_m^i}(x_t^i) - \nabla f_m^{\pi_m^i}(x_\star), x_t^i - x_\star^i\right\rangle \\
&\quad + \gamma^2\mathbb{E}_{\mathcal{Q}}\left[\left\|\frac{1}{M}\sum_{m=1}^{M}\left(\mathcal{Q}\left(\nabla f_m^{\pi_m^i}(x_t^i)\right) - \nabla f_m^{\pi_m^i}(x_t^i)\right)\right\|^2\right] \\
&\quad + \gamma^2\left\|\frac{1}{M}\sum_{m=1}^{M}\left(\nabla f_m^{\pi_m^i}(x_t^i) - \nabla f_m^{\pi_m^i}(x_\star)\right)\right\|^2 \\
&\leq \|x_t^i - x_\star^i\|^2 - \frac{2\gamma}{M}\sum_{m=1}^{M}\left\langle \nabla f_m^{\pi_m^i}(x_t^i) - \nabla f_m^{\pi_m^i}(x_\star), x_t^i - x_\star^i\right\rangle \\
&\quad + \gamma^2\left\|\frac{1}{M}\sum_{m=1}^{M}\left(\nabla f_m^{\pi_m^i}(x_t^i) - \nabla f_m^{\pi_m^i}(x_\star)\right)\right\|^2 \\
&\quad + \frac{\gamma^2\omega}{M^2}\sum_{m=1}^{M}\left\|\nabla f_m^{\pi_m^i}(x_t^i)\right\|^2,
\end{aligned}
$$

where in the last step we apply independence of $\mathcal{Q}\left(\nabla f_m^{\pi_m^i}(x_t^i)\right)$ for $m \in [M]$. Next, we use three-point identity[1] and obtain

$$
\begin{aligned}
\mathbb{E}_{\mathcal{Q}}\left[\|x_t^{i+1} - x_\star^{i+1}\|^2\right] &\leq \|x_t^i - x_\star^i\|^2 \\
&\quad - \frac{2\gamma}{M}\sum_{m=1}^{M}\left(D_{f_m^{\pi_m^i}}(x_\star, x_t^i) + D_{f_m^{\pi_m^i}}(x_t^i, x_\star) - D_{f_m^{\pi_m^i}}(x_\star, x_\star)\right) \\
&\quad + \gamma^2\left\|\frac{1}{M}\sum_{m=1}^{M}\left(\nabla f_m^{\pi_m^i}(x_t^i) - \nabla f_m^{\pi_m^i}(x_\star)\right)\right\|^2 \\
&\quad + \frac{\gamma^2\omega}{M^2}\sum_{m=1}^{M}\left\|\nabla f_m^{\pi_m^i}(x_t^i)\right\|^2.
\end{aligned}
$$

Applying $\widetilde{L}$-smoothness and convexity of $\frac{1}{M}\sum_{m=1}^{m}f_m^{\pi_m^i}$, $\widetilde{\mu}$-strong convexity of $\frac{1}{M}\sum_{m=1}^{m}f_m^{\pi_m^i}$, and $L_{\max}$-smoothness and convexity of $f_m^i$, we get

$$
\begin{aligned}
\mathbb{E}_{\mathcal{Q}}\left[\|x_t^{i+1} - x_\star^{i+1}\|^2\right] &\leq (1 - \gamma\widetilde{\mu})\|x_t^i - x_\star^i\|^2 - 2\gamma\left(1 - \widetilde{L}\gamma\right)\frac{1}{M}\sum_{m=1}^{M}D_{f_m^{\pi_m^i}}(x_t^i, x_\star) \\
&\quad + 2\gamma\frac{1}{M}\sum_{m=1}^{M}D_{f_m^{\pi_m^i}}(x_\star^i, x_\star) + \frac{\gamma^2\omega}{M^2}\sum_{m=1}^{M}\left\|\nabla f_m^{\pi_m^i}(x_t^i)\right\|^2 \\
&\leq (1 - \gamma\widetilde{\mu})\|x_t^i - x_\star^i\|^2 - 2\gamma\left(1 - \widetilde{L}\gamma\right)\frac{1}{M}\sum_{m=1}^{M}D_{f_m^{\pi_m^i}}(x_t^i, x_\star) \\
&\quad + 2\gamma\frac{1}{M}\sum_{m=1}^{M}D_{f_m^{\pi_m^i}}(x_\star^i, x_\star) + \frac{2\gamma^2\omega}{M^2}\sum_{m=1}^{M}\left\|\nabla f_m^{\pi_m^i}(x_\star)\right\|^2 \\
&\quad + \frac{2\gamma^2\omega}{M^2}\sum_{m=1}^{M}\left\|\nabla f_m^{\pi_m^i}(x_t^i) - \nabla f_m^{\pi_m^i}(x_\star)\right\|^2.
\end{aligned}
$$

---

[1] For any differentiable function $f : \mathbb{R}^d \to \mathbb{R}^d$ we have: $\langle \nabla f(x) - \nabla f(y), x - z\rangle = D_f(z, x) + D_f(x, y) - D_f(z, y)$.

So, we get

$$
\begin{aligned}
\mathbb{E}_{\mathcal{Q}}\left[\|x_t^{i+1} - x_\star^{i+1}\|^2\right] &\leq (1 - \gamma\widetilde{\mu})\left\|x_t^i - x_\star^i\right\|^2 + \frac{2\gamma^2\omega}{M^2}\sum_{m=1}^{M}\left\|\nabla f_m^{\pi_m^i}(x_\star)\right\|^2 \\
&\quad + \frac{2\gamma}{M}\sum_{m=1}^{M} D_{f_m^{\pi_m^i}}(x_\star^i, x_\star) \\
&\quad - 2\gamma\left(1 - \gamma\left(\widetilde{L} + \frac{2\omega L_{\max}}{M}\right)\right)\frac{1}{M}\sum_{m=1}^{M} D_{f_m^{\pi_m^i}}(x_t^i, x_\star).
\end{aligned}
$$

Taking the full expectation and using a definition of shuffle radius, $0 < \gamma \leq \frac{1}{\left(\widetilde{L} + 2\frac{\omega}{M}L_{\max}\right)}$, and $D_{f_m^{\pi_m^i}}(x_t^i, x_\star) \geq 0$, we obtain

$$
\begin{aligned}
\mathbb{E}\left[\|x_t^{i+1} - x_\star^{i+1}\|^2\right] &\leq (1 - \gamma\widetilde{\mu})\,\mathbb{E}\left[\left\|x_t^i - x_\star^i\right\|^2\right] + 2\gamma^3\sigma_{\text{rad}}^2 + \frac{2\gamma^2\omega}{M^2}\sum_{m=1}^{M}\mathbb{E}\left[\left\|\nabla f_m^{\pi_m^i}(x_\star)\right\|^2\right] \\
&= (1 - \gamma\widetilde{\mu})\,\mathbb{E}\left[\left\|x_t^i - x_\star^i\right\|^2\right] + 2\gamma^3\sigma_{\text{rad}}^2 + \frac{2\gamma^2\omega}{M^2 n}\sum_{m=1}^{M}\sum_{j=1}^{n}\left\|\nabla f_m^j(x_\star)\right\|^2 \\
&\leq (1 - \gamma\widetilde{\mu})\,\mathbb{E}\left[\left\|x_t^i - x_\star^i\right\|^2\right] + 2\gamma^3\sigma_{\text{rad}}^2 + \frac{2\gamma^2\omega}{M}\left(\zeta_\star^2 + \sigma_\star^2\right).
\end{aligned}
$$

Unrolling the recurrence in $i$, we derive

$$
\begin{aligned}
\mathbb{E}\left[\|x_{t+1} - x_\star\|^2\right] &\leq (1 - \gamma\widetilde{\mu})^n\,\mathbb{E}\left[\|x_t - x_\star\|^2\right] + 2\gamma^3\sigma_{\text{rad}}^2\sum_{j=0}^{n-1}(1 - \gamma\widetilde{\mu})^j \\
&\quad + \frac{2\gamma^2\omega}{M}\left(\zeta_\star^2 + \sigma_\star^2\right)\sum_{j=0}^{n-1}(1 - \gamma\widetilde{\mu})^j.
\end{aligned}
$$

Unrolling the recurrence in $t$, we derive

$$
\begin{aligned}
\mathbb{E}\left[\|x_T - x_\star\|^2\right] &\leq (1 - \gamma\widetilde{\mu})^{nT}\|x_0 - x_\star\|^2 + 2\gamma^3\sigma_{\text{rad}}^2\sum_{t=0}^{T-1}(1 - \gamma\widetilde{\mu})^{nt}\sum_{j=0}^{n-1}(1 - \gamma\widetilde{\mu})^j \\
&\quad + \frac{2\gamma^2\omega}{M}\left(\zeta_\star^2 + \sigma_\star^2\right)\sum_{j=0}^{nT-1}(1 - \gamma\widetilde{\mu})^{nt}\sum_{j=0}^{n-1}(1 - \gamma\widetilde{\mu})^j.
\end{aligned}
$$

Since $\sum_{j=0}^{nT-1}(1 - \gamma\widetilde{\mu})^j \leq \frac{1}{\gamma\widetilde{\mu}}$, we get the result. $\qquad\square$

**Corollary 5.** *Let the assumptions of Theorem B.1 hold and*

$$
\gamma = \min\left\{\frac{1}{\widetilde{L} + 2\frac{\omega}{M}L_{\max}}, \sqrt{\frac{\varepsilon\widetilde{\mu}}{6\sigma_{rad}^2}}, \frac{\varepsilon\widetilde{\mu}M}{6\omega\left(\zeta_\star^2 + \sigma_\star^2\right)}\right\}. \tag{11}
$$

*Then,* Q-RR *finds a solution with accuracy $\varepsilon > 0$ after the following number of communication rounds:*

$$
\widetilde{\mathcal{O}}\left(\frac{\widetilde{L}}{\widetilde{\mu}} + \frac{\omega}{M}\frac{L_{\max}}{\widetilde{\mu}} + \frac{\omega}{M}\frac{\zeta_\star^2 + \sigma_\star^2}{\varepsilon\widetilde{\mu}^2} + \frac{\sigma_{rad}}{\sqrt{\varepsilon\widetilde{\mu}^3}}\right).
$$

*Proof.* Theorem B.1 implies

$$
\mathbb{E}\|x_T - x_\star\|^2 \leq (1 - \gamma\widetilde{\mu})^{nT}\|x_0 - x_\star\|^2 + \frac{2\gamma^2\sigma_{\text{rad}}^2}{\widetilde{\mu}} + \frac{2\gamma\omega}{\widetilde{\mu}M}\left(\zeta_\star^2 + \sigma_\star^2\right). \tag{12}
$$

To estimate the number of communication rounds required to find a solution with accuracy $\varepsilon > 0$, we need to upper-bound each term from the right-hand side by $\varepsilon/3$. Thus, we get additional conditions on $\gamma$:

$$\frac{2\gamma^2 \sigma_{\text{rad}}^2}{\widetilde{\mu}} < \frac{\varepsilon}{3}, \quad \frac{2\gamma\omega}{\widetilde{\mu}M}\left(\zeta_\star^2 + \sigma_\star^2\right) < \frac{\varepsilon}{3}$$

and also the upper bound on the number of communication rounds $nT$

$$nT = \widetilde{\mathcal{O}}\left(\frac{1}{\gamma\widetilde{\mu}}\right).$$

Substituting (11), we get a final result. □

## B.2 Non-Strongly Convex Summands

In this section, we provide the analysis of Q-RR without using Assumptions 4, 6. Before we move one to the proofs, we would like to emphasize that

$$x_t^{i+1} = x_t^i - \gamma \frac{1}{M} \sum_{m=1}^{M} \mathcal{Q}\left(\nabla f_m^{\pi_m^i}(x_t^i)\right).$$

Then we have

$$x_{t+1} = x_t - \gamma \sum_{i=0}^{n-1} \frac{1}{M} \sum_{m=1}^{M} \mathcal{Q}\left(\nabla f_m^{\pi_m^i}(x_t^i)\right) = x_t - \tau \frac{1}{Mn} \sum_{i=0}^{n-1} \sum_{m=1}^{M} \mathcal{Q}\left(\nabla f_m^{\pi_m^i}(x_t^i)\right),$$

where $\tau = \gamma n$. For convenience, we denote

$$g_t = \frac{1}{Mn} \sum_{i=0}^{n-1} \sum_{m=1}^{M} \mathcal{Q}\left(\nabla f_m^{\pi_m^i}(x_t^i)\right)$$

allowing to write the update rule as $x_{t+1} = x_t - \tau g_t$.

**Lemma B.1** (Lemma 1 from (Malinovsky et al., 2022)). *For any $k \in [n]$, let $\xi_{\pi_1}, \ldots, \xi_{\pi_k}$ be sampled uniformly without replacement from a set of vectors $\{\xi_1, \ldots, \xi_n\}$ and $\bar{\xi}_\pi$ be their average. Then, it holds*

$$\mathbb{E}\bar{\xi}_\pi = \bar{\xi}, \quad \mathbb{E}\left[\|\bar{\xi}_\pi - \bar{\xi}\|^2\right] = \frac{n-k}{k(n-1)}\sigma^2, \tag{13}$$

*where $\bar{\xi} = \frac{1}{n}\sum_{i=1}^{n}\xi_i$, $\bar{\xi}_\pi = \frac{1}{k}\sum_{i=1}^{k}\xi_{\pi_i}$, $\sigma^2 = \frac{1}{n}\sum_{i=1}^{n}\|\xi_i - \bar{\xi}\|^2$*

**Lemma B.2.** *Under Assumptions 1, 2, 3, 5, the following inequality holds*

$$\mathbb{E}_{\mathcal{Q}}\left[-2\tau\langle g_t, x_t - x_\star\rangle\right] \leq -\frac{\tau\mu}{2}\|x_t - x_\star\|^2 - \tau(f(x_t) - f(x_\star)) + \frac{\tau\widetilde{L}}{n}\sum_{i=0}^{n-1}\|x_t^i - x_t\|^2.$$

*Proof.* Using that $\mathbb{E}_{\mathcal{Q}}[g_t] = \frac{1}{Mn}\sum_{i=0}^{n-1}\sum_{m=1}^{M}\nabla f_m^{\pi_m^i}(x_t^i)$ and definition of $h^\star$, we get

$$
\begin{aligned}
-2\tau\mathbb{E}_{\mathcal{Q}}\left[\langle g_t, x_t - x_\star\rangle\right] &= -\frac{1}{Mn}\sum_{i=0}^{n-1}\sum_{m=1}^{M}\left\langle\nabla f_m^{\pi_m^i}(x_t^i), x_t - x_\star\right\rangle \\
&= -\frac{1}{Mn}\sum_{m=1}^{M}\sum_{i=0}^{n-1}\left\langle\nabla f_m^{\pi_m^i}(x_t^i) - \nabla f_m^{\pi_m^i}(x_\star), x_t - x_\star\right\rangle.
\end{aligned}
$$

Using three-point identity, we obtain

$$
\begin{aligned}
-2\tau\mathbb{E}_{\mathcal{Q}}\left[\langle g_t, x_t - x_\star\rangle\right] &= -\frac{2\tau}{Mn}\sum_{m=1}^{M}\sum_{i=0}^{n-1}\left(D_{f_m^{\pi_m^i}}(x_t, x_\star) + D_{f_m^{\pi_m^i}}(x_\star, x_t^i) - D_{f_m^{\pi_m^i}}(x_t, x_t^i)\right) \\
&= -2\tau D_f(x_t, x_\star) - \frac{2\tau}{n}\sum_{i=0}^{n-1}D_{f^{\pi^i}}(x_\star, x_t^i) + \frac{2\tau}{n}\sum_{i=0}^{n-1}D_{f^{\pi^i}}(x_t, x_t^i) \\
&\leq -2\tau D_f(x_t, x_\star) + \frac{\tau\widetilde{L}}{n}\sum_{i=0}^{n-1}\|x_t^i - x_t\|^2,
\end{aligned}
$$

where in the last inequality we apply $\widetilde{L}$-smoothness and convexity of each function $f^{\pi^i}$. Finally, using $\mu$-strong convexity of $f$, we finish the proof of the lemma.

$\square$

**Lemma B.3.** *Under Assumptions 1, 2, 3, 5, the following inequality holds*

$$
\begin{aligned}
\mathbb{E}_{\mathcal{Q}}\left[\|g_t\|^2\right] &\leq 2\widetilde{L}\left(\widetilde{L} + \frac{\omega}{Mn}L_{\max}\right)\frac{1}{n}\sum_{i=0}^{n-1}\mathbb{E}\left[\|x_t^i - x_t\|^2\right] + \frac{4\omega}{Mn}\left(\zeta_\star^2 + \sigma_\star^2\right) \\
&\quad + 8\left(\widetilde{L} + \frac{\omega}{Mn}L_{\max}\right)\left(f(x_t) - f(x_\star)\right).
\end{aligned}
$$

*Proof.* Taking the expectation w.r.t. $\mathcal{Q}$ and using variance decomposition $\mathbb{E}\left[\|\xi\|^2\right] = \mathbb{E}\left[\|\xi - \mathbb{E}\left[\xi\right]\|^2\right] + \|\mathbb{E}\xi\|^2$, we get

$$
\begin{aligned}
\mathbb{E}_{\mathcal{Q}}\left[\|g_t\|^2\right] &= \mathbb{E}_{\mathcal{Q}}\left[\left\|\frac{1}{Mn}\sum_{i=0}^{n-1}\sum_{m=1}^{M}\mathcal{Q}\left(\nabla f_m^{\pi_m^i}(x_t^i)\right)\right\|^2\right] \\
&= \mathbb{E}_{\mathcal{Q}}\left[\left\|\frac{1}{Mn}\sum_{i=0}^{n-1}\sum_{m=1}^{M}\left(\mathcal{Q}\left(\nabla f_m^{\pi_m^i}(x_t^i)\right) - \nabla f_m^{\pi_m^i}(x_t^i)\right)\right\|^2\right] \\
&\quad + \left\|\frac{1}{Mn}\sum_{i=0}^{n-1}\sum_{m=1}^{M}\nabla f_m^{\pi_m^i}(x_t^i)\right\|^2.
\end{aligned}
$$

Next, Assumption 1 and conditional independence of $\mathcal{Q}\left(\nabla f_m^{\pi_m^i}(x_t^i)\right)$ for $m = 1, \ldots, M, i = 0, \ldots, n-1$ imply

$$
\begin{aligned}
\mathbb{E}_{\mathcal{Q}}\left[\|g_t\|^2\right] &= \frac{1}{M^2 n^2}\sum_{i=0}^{n-1}\sum_{m=1}^{M}\mathbb{E}_{\mathcal{Q}}\left[\left\|\mathcal{Q}\left(\nabla f_m^{\pi_m^i}(x_t^i)\right) - \nabla f_m^{\pi_m^i}(x_t^i)\right\|^2\right] \\
&\quad + \left\|\frac{1}{Mn}\sum_{i=0}^{n-1}\sum_{m=1}^{M}\nabla f_m^{\pi_m^i}(x_t^i)\right\|^2 \\
&\leq \frac{\omega}{M^2 n^2}\sum_{i=0}^{n-1}\sum_{m=1}^{M}\left\|\nabla f_m^{\pi_m^i}(x_t^i)\right\|^2 + \left\|\frac{1}{Mn}\sum_{i=0}^{n-1}\sum_{m=1}^{M}\nabla f_m^{\pi_m^i}(x_t^i)\right\|^2 \\
&\leq \frac{2\omega}{M^2 n^2}\sum_{i=0}^{n-1}\sum_{m=1}^{M}\left\|\nabla f_m^{\pi_m^i}(x_t^i) - \nabla f_m^{\pi_m^i}(x_t)\right\|^2 + \frac{2\omega}{M^2 n^2}\sum_{i=0}^{n-1}\sum_{m=1}^{M}\left\|\nabla f_m^{\pi_m^i}(x_t)\right\|^2 \\
&\quad + 2\left\|\frac{1}{Mn}\sum_{i=0}^{n-1}\sum_{m=1}^{M}\left(\nabla f_m^{\pi_m^i}(x_t^i) - \nabla f_m^{\pi_m^i}(x_t)\right)\right\|^2 \\
&\quad + 2\left\|\frac{1}{Mn}\sum_{i=0}^{n-1}\sum_{m=1}^{M}\nabla f_m^{\pi_m^i}(x_t)\right\|^2.
\end{aligned}
$$

Using $L_{\max}$-smoothness and convexity of $f_m^i$ and $\widetilde{L}$-smoothness and convexity of $f^{\pi^i} = \frac{1}{M}\sum_{m=1}^{M} f_m^{\pi_m^i}$, we derive

$$
\begin{aligned}
\mathbb{E}_{\mathcal{Q}}\left[\|g_t\|^2\right] &\leq \frac{4\omega}{M^2 n^2} L_{\max} \sum_{i=0}^{n-1}\sum_{m=1}^{M} D_{f_m^{\pi_m^i}}(x_t^i, x_t) + \frac{2\omega}{M^2 n^2}\sum_{i=0}^{n-1}\sum_{m=1}^{M}\left\|\nabla f_m^{\pi_m^i}(x_t)\right\|^2 \\
&\quad + 4\widetilde{L}\frac{1}{n}\sum_{i=0}^{n-1} D_{f^{\pi^i}}(x_t^i, x_t) + 2\left\|\nabla f(x_t)\right\|^2 \\
&\leq 4\left(\widetilde{L} + \frac{\omega}{Mn}L_{\max}\right)\frac{1}{n}\sum_{i=0}^{n-1} D_{f^{\pi^i}}(x_t^i, x_t) + \frac{4\omega}{M^2 n^2}\sum_{i=0}^{n-1}\sum_{m=1}^{M}\left\|\nabla f_m^{\pi_m^i}(x_\star)\right\|^2 \\
&\quad + \frac{4\omega}{M^2 n^2}\sum_{i=0}^{n-1}\sum_{m=1}^{M}\left\|\nabla f_m^{\pi_m^i}(x_t) - \nabla f_m^{\pi_m^i}(x_\star)\right\|^2 + 2\left\|\nabla f(x_t) - \nabla f(x_\star)\right\|^2 \\
&\leq 2\widetilde{L}\left(\widetilde{L} + \frac{\omega}{Mn}L_{\max}\right)\frac{1}{n}\sum_{i=0}^{n-1}\left\|x_t^i - x_t\right\|^2 + \frac{4\omega}{M^2 n^2}\sum_{i=0}^{n-1}\sum_{m=1}^{M}\left\|\nabla f_m^{\pi_m^i}(x_\star)\right\|^2 \\
&\quad + \frac{8\omega}{M^2 n^2}L_{\max}\sum_{i=0}^{n-1}\sum_{m=1}^{M} D_{f_m^{\pi_m^i}}(x_t, x_\star) + 4\widetilde{L}\left(f(x_t) - f(x_\star)\right).
\end{aligned}
$$

Taking the full expectation, we obtain

$$
\begin{aligned}
\mathbb{E}\left[\|g_t\|^2\right] &\leq 2\widetilde{L}\left(\widetilde{L} + \frac{\omega}{Mn}L_{\max}\right)\frac{1}{n}\sum_{i=0}^{n-1}\mathbb{E}\left[\left\|x_t^i - x_t\right\|^2\right] + \frac{4\omega}{M^2 n^2}\sum_{i=0}^{n-1}\sum_{m=1}^{M}\mathbb{E}\left[\left\|\nabla f_m^{\pi_m^i}(x_\star)\right\|^2\right] \\
&\quad + \left(4\widetilde{L} + \frac{8\omega}{Mn}L_{\max}\right)\mathbb{E}\left[f(x_t) - f(x_\star)\right] \\
&= 2\widetilde{L}\left(\widetilde{L} + \frac{\omega}{Mn}L_{\max}\right)\frac{1}{n}\sum_{i=0}^{n-1}\mathbb{E}\left[\left\|x_t^i - x_t\right\|^2\right] + \frac{4\omega}{Mn}\left(\zeta_\star^2 + \sigma_\star^2\right) \\
&\quad + \left(4\widetilde{L} + \frac{8\omega}{Mn}L_{\max}\right)\mathbb{E}\left[f(x_t) - f(x_\star)\right].
\end{aligned}
$$

$\square$

**Lemma B.4.** *Let Assumptions 1, 2, 3, 5 hold and* $\tau \leq \frac{1}{2\sqrt{\widetilde{L}\left(\widetilde{L} + \frac{\omega}{Mn}L_{\max}\right)}}$. *Then, the following inequality holds*

$$
\begin{aligned}
\frac{1}{n}\sum_{i=0}^{n-1}\mathbb{E}\left[\|x_t^i - x_t\|^2\right] &\leq 24\tau^2\left(\widetilde{L} + \frac{\omega}{Mn}L_{\max}\right)\mathbb{E}\left[f(x_t) - f(x_\star)\right] \\
&\quad + 8\tau^2\frac{\omega}{Mn}\left(\zeta_\star^2 + \sigma_\star^2\right) + 8\tau^2\frac{\sigma_{\star,n}^2}{n},
\end{aligned}
$$

*where* $\sigma_{\star,n}^2 = \frac{1}{n}\sum_{i=1}^{n}\|\nabla f^i(x_\star)\|^2$, $f^i(x) = \frac{1}{M}\sum_{m=1}^{M} f_m^i(x)$, $i \in [n]$.

*Proof.* Since $x_t^i = x_t - \frac{\tau}{Mn} \sum_{m=1}^{M} \sum_{j=0}^{i-1} \mathcal{Q}\left(\nabla f_m^{\pi_m^j}(x_t^j)\right)$, we have

$$
\begin{aligned}
\mathbb{E}_{\mathcal{Q}}\left[\|x_t^i - x_t\|^2\right] &= \tau^2 \mathbb{E}_{\mathcal{Q}}\left[\left\|\frac{1}{Mn} \sum_{m=1}^{M} \sum_{j=0}^{i-1} \mathcal{Q}\left(\nabla f_m^{\pi_m^j}(x_t^j)\right)\right\|^2\right] \\
&= \tau^2 \mathbb{E}_{\mathcal{Q}}\left[\left\|\frac{1}{Mn} \sum_{m=1}^{M} \sum_{j=0}^{i-1}\left(\mathcal{Q}\left(\nabla f_m^{\pi_m^j}(x_t^j)\right) - \nabla f_m^{\pi_m^j}(x_t^j)\right)\right\|^2\right] \\
&\quad + \tau^2\left\|\frac{1}{Mn} \sum_{m=1}^{M} \sum_{j=0}^{i-1} \nabla f_m^{\pi_m^j}(x_t^j)\right\|^2 \\
&\leq \frac{\tau^2}{M^2 n^2} \sum_{m=1}^{M} \sum_{j=0}^{i-1} \mathbb{E}_{\mathcal{Q}}\left[\left\|\mathcal{Q}\left(\nabla f_m^{\pi_m^j}(x_t^j)\right) - \nabla f_m^{\pi_m^j}(x_t^j)\right\|^2\right] \\
&\quad + \tau^2\left\|\frac{1}{Mn} \sum_{m=1}^{M} \sum_{j=0}^{i-1} \nabla f_m^{\pi_m^j}(x_t^j)\right\|^2.
\end{aligned}
$$

Using Assumption 1, $\widetilde{L}$-smoothness and convexity of $f^{\pi^i} = \frac{1}{M} \sum_{m=1}^{M} f_m^{\pi_m^i}$ and $L_{\max}$-smoothness and convexity of $f_m^i$, we obtain

$$
\begin{aligned}
\mathbb{E}_{\mathcal{Q}}\left[\|x_t^i - x_t\|^2\right] &\leq \frac{\tau^2 \omega}{M^2 n^2} \sum_{m=1}^{M} \sum_{j=0}^{i-1}\left\|\nabla f_m^{\pi_m^j}(x_t^j)\right\|^2 + \tau^2\left\|\frac{1}{Mn} \sum_{m=1}^{M} \sum_{j=0}^{i-1} \nabla f_m^{\pi_m^j}(x_t^j)\right\|^2 \\
&\leq \frac{2\tau^2 \omega}{M^2 n^2} \sum_{m=1}^{M} \sum_{j=0}^{i-1}\left\|\nabla f_m^{\pi_m^j}(x_t^j) - \nabla f_m^{\pi_m^j}(x_t)\right\|^2 + 2\tau^2\left\|\frac{1}{n} \sum_{j=0}^{i-1} \nabla f^{\pi^j}(x_t)\right\|^2 \\
&\quad + 2\tau^2\left\|\frac{1}{n} \sum_{j=0}^{i-1}\left(\nabla f^{\pi^j}(x_t^j) - \nabla f^{\pi^j}(x_t)\right)\right\|^2 \qquad (14) \\
&\quad + \frac{2\tau^2 \omega}{M^2 n^2} \sum_{m=1}^{M} \sum_{j=0}^{i-1}\left\|\nabla f_m^{\pi_m^j}(x_t)\right\|^2 \\
&\leq \frac{4\tau^2 \omega}{M^2 n^2} \sum_{m=1}^{M} \sum_{j=0}^{n-1} L_{\max} D_{f_m^{\pi_m^j}}(x_t^j, x_t) + 2\tau^2\left\|\frac{1}{n} \sum_{j=0}^{i-1} \nabla f^{\pi^j}(x_t)\right\|^2 \\
&\quad + 4\widetilde{L}\tau^2 \frac{1}{n} \sum_{j=0}^{n-1} D_{f^{\pi^j}}(x_t^j, x_t) + \frac{2\tau^2 \omega}{M^2 n^2} \sum_{m=1}^{M} \sum_{j=0}^{n-1}\left\|\nabla f_m^{\pi_m^j}(x_t)\right\|^2 \\
&= 4\tau^2\left(\widetilde{L} + \frac{\omega}{Mn} L_{\max}\right) \frac{1}{n} \sum_{j=0}^{n-1} D_{f^{\pi^j}}(x_t^j, x_t) \\
&\quad + 2\tau^2\left\|\frac{1}{n} \sum_{j=0}^{i-1} \nabla f^{\pi^j}(x_t)\right\|^2 + \frac{2\tau^2 \omega}{M^2 n^2} \sum_{m=1}^{M} \sum_{j=0}^{n-1}\left\|\nabla f_m^{\pi_m^j}(x_t)\right\|^2. \qquad (15)
\end{aligned}
$$

Next, we need to estimate the second term from the previous inequality. Taking the full expectation and using Lemma B.1 and using new notation $\sigma_t^2 = \frac{1}{n}\sum_{j=1}^n \mathbb{E}[\|\nabla f^j(x_t) - \nabla f(x_t)\|^2]$, we get

$$
\begin{aligned}
\mathbb{E}\left[\left\|\frac{1}{n}\sum_{j=0}^{i-1}\nabla f^{\pi^j}(x_t)\right\|^2\right] &= \frac{i^2}{n^2}\mathbb{E}\left[\|\nabla f(x_t)\|^2\right] + \frac{i^2}{n^2}\mathbb{E}\left[\left\|\frac{1}{i}\sum_{j=0}^{i-1}\left(\nabla f^{\pi^j}(x_t) - \nabla f(x_t)\right)\right\|^2\right] \\
&\leq \frac{i^2}{n^2}\mathbb{E}\left[\|\nabla f(x_t)\|^2\right] + \frac{i^2}{n^3}\frac{n-i}{i(n-1)}\sum_{j=1}^n \mathbb{E}\left[\|\nabla f^j(x_t) - \nabla f(x_t)\|^2\right] \\
&\leq \mathbb{E}\left[\|\nabla f(x_t)\|^2\right] + \frac{1}{n}\sigma_t^2. \quad (16)
\end{aligned}
$$

Taking the full expectation from (15) and using (16), we obtain

$$
\begin{aligned}
\mathbb{E}\left[\|x_t^i - x_t\|^2\right] &\leq 4\tau^2\left(\widetilde{L} + \frac{\omega}{Mn}L_{\max}\right)\sum_{j=0}^{n-1}\mathbb{E}\left[D_{f^{\pi^j}}(x_t^j, x_t)\right] \\
&\quad + 2\tau^2\mathbb{E}\left[\|\nabla f(x_t)\|^2\right] + \frac{2\tau^2}{n}\sigma_t^2 + \frac{2\tau^2\omega}{M^2n^2}\sum_{m=1}^M\sum_{j=0}^{n-1}\mathbb{E}\left[\left\|\nabla f_m^{\pi_m^j}(x_t)\right\|^2\right].
\end{aligned}
$$

Using $\widetilde{L}$-smoothness of $f^{\pi^j}$, we get

$$
\begin{aligned}
\mathbb{E}\left[\|x_t^i - x_t\|^2\right] &\leq 2\widetilde{L}\tau^2\left(\widetilde{L} + \frac{\omega}{Mn}L_{\max}\right)\sum_{j=0}^{n-1}\mathbb{E}\left[\|x_t^j - x_t\|^2\right] \\
&\quad + 2\tau^2\mathbb{E}\left[\|\nabla f(x_t)\|^2\right] + \frac{2\tau^2}{n}\sigma_t^2 + \frac{2\tau^2\omega}{M^2n^2}\sum_{m=1}^M\sum_{j=0}^{n-1}\mathbb{E}\left[\left\|\nabla f_m^{\pi_m^j}(x_t)\right\|^2\right].
\end{aligned}
$$

Since $\tau \leq \frac{1}{2\sqrt{\widetilde{L}\left(\widetilde{L}+\frac{\omega}{Mn}L_{\max}\right)}}$, we have

$$
\begin{aligned}
\mathbb{E}\left[\|x_t^i - x_t\|^2\right] &\leq 2\left(1 - 2\widetilde{L}\tau^2\left(\widetilde{L} + \frac{\omega}{Mn}L_{\max}\right)\right)\sum_{j=0}^{n-1}\mathbb{E}\left[\|x_t^j - x_t\|^2\right] \\
&\leq 4\tau^2\mathbb{E}\left[\|\nabla f(x_t)\|^2\right] + \frac{4\tau^2}{n}\sigma_t^2 + \frac{4\tau^2\omega}{M^2n^2}\sum_{m=1}^M\sum_{j=0}^{n-1}\mathbb{E}\left[\left\|\nabla f_m^{\pi_m^j}(x_t)\right\|^2\right] \\
&\leq \frac{8\tau^2\omega}{M^2n^2}\sum_{m=1}^M\sum_{j=0}^{n-1}\mathbb{E}\left[\left\|\nabla f_m^{\pi_m^j}(x_t) - \nabla f_m^{\pi_m^j}(x_\star)\right\|^2\right] \\
&\quad + \frac{8\tau^2\omega}{M^2n^2}\sum_{m=1}^M\sum_{j=0}^{n-1}\mathbb{E}\left[\left\|\nabla f_m^{\pi_m^j}(x_\star)\right\|^2\right] + 4\tau^2\mathbb{E}\left[\|\nabla f(x_t) - \nabla f(x_\star)\|^2\right] \\
&\quad + \frac{4\tau^2}{n}\left(\frac{1}{n}\sum_{j=1}^n\mathbb{E}\left[\|\nabla f^j(x_t)\|\right] - \mathbb{E}\left[\|\nabla f(x_t)\|^2\right]\right) \\
&\leq \frac{8\tau^2\omega}{M^2n^2}\sum_{m=1}^M\sum_{j=0}^{n-1}\mathbb{E}\left[\left\|\nabla f_m^{\pi_m^j}(x_t) - \nabla f_m^{\pi_m^j}(x_\star)\right\|^2\right] \\
&\quad + \frac{8\tau^2\omega}{M^2n^2}\sum_{m=1}^M\sum_{j=0}^{n-1}\mathbb{E}\left[\left\|\nabla f_m^{\pi_m^j}(x_\star)\right\|^2\right] + 8\tau^2\mathbb{E}\left[\|\nabla f(x_t) - \nabla f(x_\star)\|^2\right] \\
&\quad + \frac{8\tau^2}{n^2}\sum_{j=1}^n\mathbb{E}\left[\|\nabla f^j(x_t) - \nabla f^j(x_\star)\|^2\right] + \frac{8\tau^2}{n^2}\sum_{j=1}^n\mathbb{E}\left[\|\nabla f^j(x_\star)\|^2\right].
\end{aligned}
$$

Summing from $i = 0$ to $n - 1$ and using $\widetilde{L}$-smoothness of $f^i$ and $L_{\max}$-smoothness of $f_m^i$, we obtain

$$
\begin{aligned}
\frac{1}{n} \sum_{i=0}^{n-1} \mathbb{E} \left[ \|x_t^i - x_t\|^2 \right] &\leq \frac{16\tau^2\omega}{Mn} L_{\max} \mathbb{E} \left[ f(x_t) - f(x_\star) \right] + \frac{16\tau^2}{n} \widetilde{L} \mathbb{E} \left[ f(x_t) - f(x_\star) \right] \\
&\quad + \frac{8\tau^2\omega}{Mn} \left( \zeta_\star^2 + \sigma_\star^2 \right) + \frac{8\tau^2}{n} \sigma_{\star,n}^2 + 8\tau^2 \widetilde{L} \mathbb{E} \left[ f(x_t) - f(x_\star) \right].
\end{aligned}
$$

$\square$

**Theorem B.2.** *Let Assumptions 1, 2, 3, 5 hold and stepsize $\gamma$ satisfy*

$$
0 < \gamma \leq \frac{1}{16n \left( \widetilde{L} + \frac{\omega}{Mn} L_{\max} \right)}. \tag{17}
$$

*Then, for all $T \geq 0$ the iterates produced by* Q-RR *satisfy*

$$
\begin{aligned}
\mathbb{E} \left[ \|x_T - x_\star\|^2 \right] &\leq \left( 1 - \frac{n\gamma\mu}{2} \right)^T \|x_0 - x_\star\|^2 + 18\frac{\gamma^2 n\widetilde{L}}{\mu} \left( \frac{\omega}{M}(\zeta_\star^2 + \sigma_\star^2) + \sigma_{\star,n}^2 \right) \\
&\quad + 8\frac{\gamma\omega}{\mu M}(\zeta_\star^2 + \sigma_\star^2),
\end{aligned}
$$

*where*

$$
\sigma_{\star,n}^2 = \frac{1}{n} \sum_{i=1}^{n} \|\nabla f^i(x_\star)\|^2. \tag{18}
$$

*Proof.* Taking expectation w.r.t. $\mathcal{Q}$ and using Lemma B.3, we get

$$
\begin{aligned}
\mathbb{E}_{\mathcal{Q}} \left[ \|x_{t+1} - x_\star\|^2 \right] &= \|x_t - x_\star\|^2 - 2\tau \mathbb{E}_{\mathcal{Q}} \left[ \langle g_t, x_t - x_\star \rangle \right] + \tau^2 \mathbb{E}_{\mathcal{Q}} \left[ \|g^t\|^2 \right] \\
&\leq \|x_t - x_\star\|^2 - 2\tau \mathbb{E}_{\mathcal{Q}} \left[ \langle g^t, x_t - x_\star \rangle \right] \\
&\quad + 2\tau^2 \widetilde{L} \left( \widetilde{L} + \frac{\omega}{Mn} L_{\max} \right) \frac{1}{n} \sum_{i=0}^{n-1} \mathbb{E} \left[ \|x_t^i - x_t\|^2 \right] \\
&\quad + 8\tau^2 \left( \widetilde{L} + \frac{\omega}{Mn} L_{\max} \right) (f(x_t) - f(x_\star)) + \frac{4\tau^2\omega}{Mn}(\zeta_\star^2 + \sigma_\star^2).
\end{aligned}
$$

Using Lemma B.2, we obtain

$$
\begin{aligned}
\mathbb{E}_{\mathcal{Q}} \left[ \|x_{t+1} - x_\star\|^2 \right] &\leq \|x_t - x_\star\|^2 \\
&\quad - \frac{\tau\mu}{2} \|x_t - x_\star\|^2 - \tau(f(x_t) - f(x_\star)) + \frac{\tau\widetilde{L}}{n} \sum_{i=0}^{n-1} \|x_t^i - x_t\|^2 \\
&\quad + 2\tau^2 \widetilde{L} \left( \widetilde{L} + \frac{\omega}{Mn} L_{\max} \right) \frac{1}{n} \sum_{i=0}^{n-1} \mathbb{E} \left[ \|x_t^i - x_t\|^2 \right] \\
&\quad + 8\tau^2 \left( \widetilde{L} + \frac{\omega}{Mn} L_{\max} \right) (f(x_t) - f(x_\star)) + \frac{4\tau^2\omega}{Mn}(\zeta_\star^2 + \sigma_\star^2) \\
&\leq \left( 1 - \frac{\tau\mu}{2} \right) \|x_t - x_\star\|^2 \\
&\quad - \tau \left( 1 - 8\tau \left( \widetilde{L} + \frac{\omega}{Mn} L_{\max} \right) \right) (f(x_t) - f(x_\star)) \\
&\quad + \tau\widetilde{L} \left( 1 + 2\tau \left( \widetilde{L} + \frac{\omega}{Mn} L_{\max} \right) \right) \frac{1}{n} \sum_{i=0}^{n-1} \mathbb{E} \left[ \|x_t^i - x_t\|^2 \right] \\
&\quad + \frac{4\tau^2\omega}{Mn}(\zeta_\star^2 + \sigma_\star^2).
\end{aligned}
$$

Next, we take the full expectation and apply Lemma B.4:

$$
\begin{aligned}
\mathbb{E}\left[\|x_{t+1} - x_\star\|^2\right] &\leq \left(1 - \frac{\tau\mu}{2}\right)\mathbb{E}\left[\|x_t - x_\star\|^2\right] \\
&- \tau\left(1 - 8\tau\left(\widetilde{L} + \frac{\omega}{Mn}L_{\max}\right)\right)\mathbb{E}\left[f(x_t) - f(x_\star)\right] \\
&+ 24\tau^3\widetilde{L}\left(1 + 2\tau\left(\widetilde{L} + \frac{\omega}{Mn}L_{\max}\right)\right)\left(\widetilde{L} + \frac{\omega}{Mn}L_{\max}\right)(f(x_t) - f(x_\star)) \\
&+ 8\tau^3\widetilde{L}\left(1 + 2\tau\left(\widetilde{L} + \frac{\omega}{Mn}L_{\max}\right)\right)\left(\frac{\omega}{Mn}(\zeta_\star^2 + \sigma_\star^2) + \frac{\sigma_{\star,n}^2}{n}\right) + \frac{4\tau^2\omega}{Mn}(\zeta_\star^2 + \sigma_\star^2).
\end{aligned}
$$

Using (17), we derive

$$
\begin{aligned}
\mathbb{E}\left[\|x_{t+1} - x_\star\|^2\right] &\leq \left(1 - \frac{\tau\mu}{2}\right)\mathbb{E}\left[\|x_t - x_\star\|^2\right] \\
&+ 9\tau^3\widetilde{L}\left(\frac{\omega}{Mn}(\zeta_\star^2 + \sigma_\star^2) + \frac{\sigma_{\star,n}^2}{n}\right) + \frac{4\tau^2\omega}{Mn}(\zeta_\star^2 + \sigma_\star^2)
\end{aligned}
$$

Recursively unrolling the inequality, substituting $\tau = n\gamma$ and using $\sum_{t=0}^{+\infty}\left(1 - \frac{\tau\mu}{2}\right)^t \leq \frac{2}{\mu\tau}$, we get the result. $\qquad\square$

**Corollary 6.** *Let the assumptions of Theorem B.2 hold and*

$$
\gamma = \min\left\{\frac{1}{16n\left(\widetilde{L} + \frac{\omega}{Mn}L_{\max}\right)}, \sqrt{\frac{\varepsilon\mu}{8^2 n\widetilde{L}}}\left(\frac{\omega}{M}\Delta_\star^2 + \sigma_{\star,n}^2\right)^{-\frac{1}{2}}, \frac{\varepsilon\mu M}{24\omega\Delta_\star^2}\right\}, \tag{19}
$$

*where $\Delta_\star^2 = \zeta_\star^2 + \sigma_\star^2$. Then,* Q-RR *finds a solution with accuracy $\varepsilon > 0$ after the following number of communication rounds:*

$$
\widetilde{\mathcal{O}}\left(\frac{n\widetilde{L}}{\mu} + \frac{\omega}{M}\frac{L_{\max}}{\mu} + \frac{\omega}{M}\frac{\zeta_\star^2 + \sigma_\star^2}{\varepsilon\mu^2} + \sqrt{\frac{n\widetilde{L}}{\varepsilon\mu^3}}\sqrt{\frac{\omega}{M}\left(\zeta_\star^2 + \sigma_\star^2\right) + \sigma_{\star,n}^2}\right).
$$

*Proof.* Theorem B.2 implies

$$
\begin{aligned}
\mathbb{E}\left[\|x_T - x_\star\|^2\right] &\leq \left(1 - \frac{n\gamma\mu}{2}\right)^T\|x_0 - x_\star\|^2 + 18\frac{\gamma^2 n\widetilde{L}}{\mu}\left(\frac{\omega}{M}\left(\zeta_\star^2 + \sigma_\star^2\right) + \sigma_{\star,n}^2\right) \\
&+ 8\frac{\gamma\omega}{\mu M}\left(\zeta_\star^2 + \sigma_\star^2\right).
\end{aligned}
$$

To estimate the number of communication rounds required to find a solution with accuracy $\varepsilon > 0$, we need to upper bound each term from the right-hand side by $\varepsilon/3$. Thus, we get additional conditions on $\gamma$:

$$
18\frac{\gamma^2 n\widetilde{L}}{\mu}\left(\frac{\omega}{M}\left(\zeta_\star^2 + \sigma_\star^2\right) + \sigma_{\star,n}^2\right) < \frac{\varepsilon}{3}, \quad 8\frac{\gamma\omega}{\mu M}\left(\zeta_\star^2 + \sigma_\star^2\right) < \frac{\varepsilon}{3},
$$

and also the upper bound on the number of communication rounds $nT$

$$
nT = \widetilde{\mathcal{O}}\left(\frac{1}{\gamma\mu}\right).
$$

Substituting (19) in the previous equation, we get the result. $\qquad\square$

## C    MISSING PROOFS FOR DIANA-RR

### C.1    PROOF OF THEOREM 2.2

**Lemma C.1.** *Let Assumptions 1, 3, 5, 6 hold and $\alpha \leq \frac{1}{1+\omega}$. Then, the iterates of DIANA-RR satisfy*

$$\frac{1}{M} \sum_{m=1}^{M} \mathbb{E}_{\mathcal{Q}} \left[ \|h_{t+1,m}^{\pi_m^i} - \nabla f_m^{\pi_m^i}(x_\star)\|^2 \right] \leq \frac{1-\alpha}{M} \sum_{m=1}^{M} \|h_{t,m}^{\pi_m^i} - \nabla f_m^{\pi_m^i}(x_\star)\|^2$$
$$+ \frac{2\alpha L_{\max}}{M} \sum_{m=1}^{M} D_{f_m^{\pi_m^i}}(x_t^i, x_\star).$$

*Proof.* Taking expectation w.r.t. $\mathcal{Q}$, we obtain

$$\begin{aligned}
\mathbb{E}_{\mathcal{Q}} \left[ \|h_{t+1,m}^{\pi_m^i} - \nabla f_m^{\pi_m^i}(x_\star)\|^2 \right] &= \mathbb{E}_{\mathcal{Q}} \left[ \|h_{t,m}^{\pi_m^i} + \alpha \mathcal{Q}(\nabla f_m^{\pi_m^i}(x_t^i) - h_{t,m}^{\pi_m^i}) - \nabla f_m^{\pi_m^i}(x_\star)\|^2 \right] \\
&= \|h_{t,m}^{\pi_m^i} - \nabla f_m^{\pi_m^i}(x_\star)\|^2 \\
&\quad + 2\alpha \mathbb{E}_{\mathcal{Q}} \left[ \left\langle \mathcal{Q}(\nabla f_m^{\pi_m^i}(x_t^i) - h_{t,m}^{\pi_m^i}), h_{t,m}^{\pi_m^i} - \nabla f_m^{\pi_m^i}(x_\star) \right\rangle \right] \\
&\quad + \alpha^2 \mathbb{E}_{\mathcal{Q}} \left[ \|\mathcal{Q}(\nabla f_m^{\pi_m^i}(x_t^i) - h_{t,m}^{\pi_m^i})\|^2 \right] \\
&= \|h_{t,m}^{\pi_m^i} - \nabla f_m^{\pi_m^i}(x_\star)\|^2 \\
&\quad + 2\alpha \left\langle \nabla f_m^{\pi_m^i}(x_t^i) - h_{t,m}^{\pi_m^i}, h_{t,m}^{\pi_m^i} - \nabla f_m^{\pi_m^i}(x_\star) \right\rangle \\
&\quad + \alpha^2 \mathbb{E}_{\mathcal{Q}} \left[ \|\mathcal{Q}(\nabla f_m^{\pi_m^i}(x_t^i) - h_{t,m}^{\pi_m^i})\|^2 \right].
\end{aligned}$$

Assumption 1, $L_{\max}$-smoothness and convexity of $f_m^i$ and $\alpha \leq 1/(1+\omega)$ imply

$$\begin{aligned}
\mathbb{E}_{\mathcal{Q}} \left[ \|h_{t+1,m}^{\pi_m^i} - \nabla f_m^{\pi_m^i}(x_\star)\|^2 \right] &\leq \|h_{t,m}^{\pi_m^i} - \nabla f_m^{\pi_m^i}(x_\star)\|^2 \\
&\quad + 2\alpha \left\langle \nabla f_m^{\pi_m^i}(x_t^i) - h_{t,m}^{\pi_m^i}, h_{t,m}^{\pi_m^i} - \nabla f_m^{\pi_m^i}(x_\star) \right\rangle \\
&\quad + \alpha^2(1+\omega) \|\nabla f_m^{\pi_m^i}(x_t^i) - h_{t,m}^{\pi_m^i}\|^2 \\
&\leq \|h_{t,m}^{\pi_m^i} - \nabla f_m^{\pi_m^i}(x_\star)\|^2 \\
&\quad + \alpha \left\langle \nabla f_m^{\pi_m^i}(x_t^i) - h_{t,m}^{\pi_m^i}, h_{t,m}^{\pi_m^i} + \nabla f_m^{\pi_m^i}(x_t^i) - 2\nabla f_m^{\pi_m^i}(x_\star) \right\rangle \\
&\leq \|h_{t,m}^{\pi_m^i} - \nabla f_m^{\pi_m^i}(x_\star)\|^2 \\
&\quad + \alpha \|\nabla f_m^{\pi_m^i}(x_t^i) - \nabla f_m^{\pi_m^i}(x_\star)\|^2 - \alpha \|h_{t,m}^{\pi_m^i} - \nabla f_m^{\pi_m^i}(x_\star)\|^2 \\
&\leq (1-\alpha) \|h_{t,m}^{\pi_m^i} - \nabla f_m^{\pi_m^i}(x_\star)\|^2 \\
&\quad + \alpha \|\nabla f_m^{\pi_m^i}(x_t^i) - \nabla f_m^{\pi_m^i}(x_\star)\|^2 \qquad (20) \\
&\leq (1-\alpha) \|h_{t,m}^{\pi_m^i} - \nabla f_m^{\pi_m^i}(x_\star)\|^2 + 2\alpha L_{\max} D_{f_m^{\pi_m^i}}(x_t^i, x_\star).
\end{aligned}$$

Summing up the above inequality for $m = 1, \ldots, M$, we get the result. $\qquad \square$

**Theorem C.1.** *Let Assumptions 1, 3, 5, 6 hold and $0 < \gamma \leq \min\left\{ \frac{\alpha}{2n\widetilde{\mu}}, \frac{1}{\widetilde{L} + \frac{6\omega}{M}L_{\max}} \right\}$, $\alpha \leq \frac{1}{1+\omega}$. Then, for all $T \geq 0$ the iterates produced by DIANA-RR satisfy*

$$\mathbb{E}[\Psi_T] \leq (1 - \gamma\widetilde{\mu})^{nT} \Psi_0 + \frac{2\gamma^2 \sigma_{rad}^2}{\widetilde{\mu}},$$

*where $\Psi_t$ is defined in (6).*

*Proof.* Using $x_\star^{i+1} = x_\star^i - \frac{\gamma}{M} \sum_{m=1}^{M} \nabla f_m^{\pi_m^i}(x_\star)$ and line 9 of Algorithm 3, we derive

$$
\begin{aligned}
\|x_t^{i+1} - x_\star^{i+1}\|^2 &= \left\| x_t^i - x_\star^i - \gamma \frac{1}{M} \sum_{m=1}^{M} \left( \hat{g}_{t,m}^{\pi_m^i} - \nabla f_m^{\pi_m^i}(x_\star) \right) \right\|^2 \\
&= \|x_t^i - x_\star^i\|^2 - \frac{2\gamma}{M} \sum_{m=1}^{M} \left\langle \left( \hat{g}_{t,m}^{\pi_m^i} - \nabla f_m^{\pi_m^i}(x_\star) \right), x_t^i - x_\star^i \right\rangle \\
&\quad + \gamma^2 \left\| \frac{1}{M} \sum_{m=1}^{M} \left( \hat{g}_{t,m}^{\pi_m^i} - \nabla f_m^{\pi_m^i}(x_\star) \right) \right\|^2 .
\end{aligned}
$$

Taking expectation w.r.t. $\mathcal{Q}$ and using $\mathbb{E}\|\xi - c\|^2 = \mathbb{E}\|\xi - \mathbb{E}\xi\|^2 + \|\mathbb{E}\xi - c\|^2$, we obtain

$$
\begin{aligned}
\mathbb{E}_{\mathcal{Q}} \left[ \|x_t^{i+1} - x_\star^{i+1}\|^2 \right] &= \|x_t^i - x_\star^i\|^2 - \frac{2\gamma}{M} \sum_{m=1}^{M} \left\langle \nabla f_m^{\pi_m^i}(x_t^i) - \nabla f_m^{\pi_m^i}(x_\star), x_t^i - x_\star^i \right\rangle \\
&\quad + \gamma^2 \mathbb{E}_{\mathcal{Q}} \left[ \left\| \frac{1}{M} \sum_{m=1}^{M} \left( \mathcal{Q}\left( \nabla f_m^{\pi_m^i}(x_t^i) - h_{t,m}^{\pi_m^i} \right) + h_{t,m}^{\pi_m^i} - \nabla f_m^{\pi_m^i}(x_\star) \right) \right\|^2 \right] \\
&\leq \|x_t^i - x_\star^i\|^2 - \frac{2\gamma}{M} \sum_{m=1}^{M} \left\langle \nabla f_m^{\pi_m^i}(x_t^i) - \nabla f_m^{\pi_m^i}(x_\star), x_t^i - x_\star^i \right\rangle \\
&\quad + \gamma^2 \mathbb{E}_{\mathcal{Q}} \left[ \left\| \frac{1}{M} \sum_{m=1}^{M} \left( \mathcal{Q}\left( \nabla f_m^{\pi_m^i}(x_t^i) - h_{t,m}^{\pi_m^i} \right) - \nabla f_m^{\pi_m^i}(x_t^i) + h_{t,m}^{\pi_m^i} \right) \right\|^2 \right] \\
&\quad + \gamma^2 \left\| \frac{1}{M} \sum_{m=1}^{M} \left( \nabla f_m^{\pi_m^i}(x_\star) - \nabla f_m^{\pi_m^i}(x_t^i) \right) \right\|^2 .
\end{aligned}
$$

Independence of $\mathcal{Q}\left( \nabla f_m^{\pi_m^i}(x_t^i) - h_{t,m}^{\pi_m^i} \right)$, $m \in [M]$, assumption 1, and three-point identity imply

$$
\begin{aligned}
\mathbb{E}_{\mathcal{Q}} \left[ \|x_t^{i+1} - x_\star^{i+1}\|^2 \right] &\leq \|x_t^i - x_\star^i\|^2 \\
&\quad - \frac{2\gamma}{M} \sum_{m=1}^{M} \left( D_{f_m^{\pi_m^i}}(x_\star^i, x_t^i) + D_{f_m^{\pi_m^i}}(x_t^i, x_\star) - D_{f_m^{\pi_m^i}}(x_\star^i, x_\star) \right) \\
&\quad + \frac{\gamma^2 \omega}{M^2} \sum_{m=1}^{M} \left\| \nabla f_m^{\pi_m^i}(x_t^i) - h_{t,m}^{\pi_m^i} \right\|^2 \\
&\quad + \gamma^2 \left\| \frac{1}{M} \sum_{m=1}^{M} \left( \nabla f_m^{\pi_m^i}(x_\star) - \nabla f_m^{\pi_m^i}(x_t^i) \right) \right\|^2 \\
&\leq \|x_t^i - x_\star^i\|^2 \\
&\quad - \frac{2\gamma}{M} \sum_{m=1}^{M} \left( D_{f_m^{\pi_m^i}}(x_\star^i, x_t^i) + D_{f_m^{\pi_m^i}}(x_t^i, x_\star) - D_{f_m^{\pi_m^i}}(x_\star^i, x_\star) \right) \\
&\quad + \frac{2\gamma^2 \omega}{M} \frac{1}{M} \sum_{m=1}^{M} \left\| \nabla f_m^{\pi_m^i}(x_t^i) - \nabla f_m^{\pi_m^i}(x_\star) \right\|^2 \\
&\quad + \gamma^2 \left\| \frac{1}{M} \sum_{m=1}^{M} \left( \nabla f_m^{\pi_m^i}(x_\star) - \nabla f_m^{\pi_m^i}(x_t^i) \right) \right\|^2 \\
&\quad + \frac{2\gamma^2 \omega}{M^2} \sum_{m=1}^{M} \left\| h_{t,m}^{\pi_m^i} - \nabla f_m^{\pi_m^i}(x_\star) \right\|^2 .
\end{aligned}
$$

Using $L_{\max}$-smoothness and $\mu$-strong convexity of functions $f_m^i$ and $\widetilde{L}$-smoothness and $\widetilde{\mu}$-strong convexity of $f^{\pi^i} = \frac{1}{M} \sum_{i=1}^M f_m^{\pi_m^i}$, we obtain

$$
\begin{aligned}
\mathbb{E}_{\mathcal{Q}}\left[\|x_t^{i+1} - x_\star^{i+1}\|^2\right] \leq{} & (1 - \gamma\widetilde{\mu})\left\|x_t^i - x_\star^i\right\|^2 \\
& -2\gamma\left(1 - \gamma\left(\widetilde{L} + \frac{2\omega}{M}L_{\max}\right)\right)\frac{1}{M}\sum_{m=1}^M D_{f_m^{\pi_m^i}}(x_t^i, x_\star) \\
& +\frac{2\gamma}{M}\sum_{m=1}^M D_{f_m^{\pi_m^i}}(x_\star^i, x_\star) + \frac{2\gamma^2\omega}{M^2}\sum_{m=1}^M\left\|h_{t,m}^{\pi_m^i} - \nabla f_m^{\pi_m^i}(x_\star)\right\|^2.
\end{aligned}
$$

Taking the full expectation and using Defenition 2, we derive

$$
\begin{aligned}
\mathbb{E}\left[\|x_t^{i+1} - x_\star^{i+1}\|^2\right] \leq{} & (1 - \gamma\widetilde{\mu})\mathbb{E}\left[\left\|x_t^i - x_\star^i\right\|^2\right] \\
& -2\gamma\left(1 - \gamma\left(\widetilde{L} + \frac{2\omega}{M}L_{\max}\right)\right)\frac{1}{M}\sum_{m=1}^M \mathbb{E}\left[D_{f_m^{\pi_m^i}}(x_t^i, x_\star)\right] \\
& +2\gamma^3\sigma_{\mathrm{rad}}^2 + \frac{2\gamma^2\omega}{M^2}\sum_{m=1}^M \mathbb{E}\left[\left\|h_{t,m}^{\pi_m^i} - \nabla f_m^{\pi_m^i}(x_\star)\right\|^2\right].
\end{aligned}
$$

Recursively unrolling the inequality, we get

$$
\begin{aligned}
\mathbb{E}\left[\|x_{t+1} - x_\star\|^2\right] \leq{} & (1 - \gamma\widetilde{\mu})^n\mathbb{E}\left[\left\|x_t - x_\star\right\|^2\right] \\
& +\frac{2\gamma^2\omega}{M^2}\sum_{m=1}^M\sum_{j=0}^{n-1}(1 - \gamma\widetilde{\mu})^j\mathbb{E}\left[\left\|h_{t,m}^{\pi_m^i} - \nabla f_m^{\pi_m^i}(x_\star)\right\|^2\right] \\
& -2\gamma\left(1 - \gamma\left(\widetilde{L} + \frac{2\omega}{M}L_{\max}\right)\right)\frac{1}{M}\sum_{m=1}^M\sum_{j=0}^{n-1}(1 - \gamma\widetilde{\mu})^j\mathbb{E}\left[D_{f_m^{\pi_m^i}}(x_t^i, x_\star)\right] \\
& +2\gamma^3\sigma_{\mathrm{rad}}^2\sum_{j=0}^{n-1}(1 - \gamma\widetilde{\mu})^j.
\end{aligned}
$$

Next, we apply (6) and Lemma C.1:

$$
\begin{aligned}
\mathbb{E}\left[\Psi_{t+1}\right] \leq{} & (1 - \gamma\widetilde{\mu})^n\mathbb{E}\left[\left\|x_t - x_\star\right\|^2\right] + 2\gamma^3\sigma_{\mathrm{rad}}^2\sum_{j=0}^{n-1}(1 - \gamma\widetilde{\mu})^j \\
& +\left(c(1-\alpha) + \frac{2\omega}{M}\right)\frac{\gamma^2}{M}\sum_{m=1}^M\sum_{j=0}^{n-1}(1 - \gamma\widetilde{\mu})^j\mathbb{E}\left[\left\|h_{t,m}^{\pi_m^i} - \nabla f_m^{\pi_m^i}(x_\star)\right\|^2\right] \\
& -2\gamma\left(1 - c\gamma\alpha L_{\max} - \gamma\left(\widetilde{L} + \frac{2\omega}{M}L_{\max}\right)\right)\frac{1}{M}\sum_{m=1}^M\sum_{j=0}^{n-1}(1 - \gamma\widetilde{\mu})^j\mathbb{E}\left[D_{f_m^{\pi_m^i}}(x_\star^i, x_\star)\right],
\end{aligned}
$$

where $c = \frac{4\omega}{\alpha M^2}$. Using $\alpha \leq \frac{1}{1+\omega}$ and $\gamma \leq \min\left\{\frac{\alpha}{2n\mu}, \frac{1}{(\widetilde{L}+6\omega/ML_{\max})}\right\}$, we obtain

$$
\begin{aligned}
\mathbb{E}\left[\Psi_{t+1}\right] &\leq (1-\gamma\widetilde{\mu})^n \mathbb{E}\left[\|x_t - x_\star\|^2\right] \\
&\quad + \left(1 - \frac{\alpha}{2}\right)\frac{4\omega\gamma^2}{\alpha M^2}\sum_{m=1}^{M}\sum_{j=0}^{n-1}(1-\gamma\widetilde{\mu})^j \mathbb{E}\left[\left\|h_{t,m}^{\pi_m^i} - \nabla f_m^{\pi_m^i}(x_\star)\right\|^2\right] \\
&\quad + 2\gamma^2\sigma_{\text{rad}}^3\sum_{j=0}^{n-1}(1-\gamma\widetilde{\mu})^j \\
&\leq \max\left\{(1-\gamma\widetilde{\mu})^n, \left(1-\frac{\alpha}{2}\right)\right\}\mathbb{E}\left[\Psi_t\right] \\
&\quad + 2\gamma^2\sigma_{\text{rad}}^3\sum_{j=0}^{n-1}(1-\gamma\widetilde{\mu})^j \\
&\leq (1-\gamma\widetilde{\mu})^n\mathbb{E}\left[\Psi_t\right] + 2\gamma^3\sigma_{\text{rad}}^2\sum_{j=0}^{n-1}(1-\gamma\widetilde{\mu})^j.
\end{aligned}
$$

Recursively rewriting the inequality, we obtain

$$
\begin{aligned}
\mathbb{E}\left[\Psi_T\right] &\leq (1-\gamma\widetilde{\mu})^{nT}\Psi_0 + 2\gamma^3\sigma_{\text{rad}}^2\sum_{t=0}^{T-1}(1-\gamma\widetilde{\mu})^{tn}\sum_{j=0}^{n-1}(1-\gamma\widetilde{\mu})^j \\
&\leq (1-\gamma\widetilde{\mu})^{nT}\Psi_0 + 2\gamma^3\sigma_{\text{rad}}^2\sum_{k=0}^{nT-1}(1-\gamma\widetilde{\mu})^k
\end{aligned}
$$

Using that $\sum_{k=0}^{+\infty}\left(1 - \frac{\gamma\widetilde{\mu}}{2}\right)^k \leq \frac{2}{\widetilde{\mu}\gamma}$, we finish proof. $\qquad\square$

**Corollary 7.** *Let the assumptions of Theorem C.1 hold, $\alpha = \frac{1}{1+\omega}$ and*

$$
\gamma = \min\left\{\frac{\alpha}{2n\widetilde{\mu}}, \frac{1}{\widetilde{L} + \frac{6\omega}{M}L_{\max}}, \frac{\sqrt{\varepsilon\widetilde{\mu}}}{2\sigma_{rad}}\right\}. \tag{21}
$$

*Then DIANA-RR finds a solution with accuracy $\varepsilon > 0$ after the following number of communication rounds:*

$$
\widetilde{\mathcal{O}}\left(n(1+\omega) + \frac{\widetilde{L}}{\widetilde{\mu}} + \frac{\omega}{M}\frac{L_{\max}}{\widetilde{\mu}} + \frac{\sigma_{rad}}{\sqrt{\varepsilon\widetilde{\mu}^3}}\right).
$$

*Proof.* Theorem C.1 implies

$$
\mathbb{E}\left[\Psi_T\right] \leq (1-\gamma\widetilde{\mu})^{nT}\Psi_0 + \frac{2\gamma^2\sigma_{\text{rad}}^2}{\widetilde{\mu}}.
$$

To estimate the number of communication rounds required to find a solution with accuracy $\varepsilon > 0$, we need to upper bound each term from the right-hand side by $\frac{\varepsilon}{2}$. Thus, we get an additional condition on $\gamma$:

$$
\frac{2\gamma^2\sigma_{\text{rad}}^2}{\widetilde{\mu}} < \frac{\varepsilon}{2},
$$

and also the upper bound on the number of communication rounds $nT$

$$
nT = \widetilde{\mathcal{O}}\left(\frac{1}{\gamma\mu}\right).
$$

Substituting (21) in the previous equation, we get the result. $\qquad\square$

## C.2 Non-Strongly Convex Summands

In this section, we provide the analysis of DIANA-RR without using Assumptions 4, 6. We emphasize that $x_t^{i+1} = x_t^i - \gamma \frac{1}{M} \sum_{m=1}^{M} \hat{g}_{t,m}^{\pi_m^i}$. Then we have

$$x_{t+1} = x_t - \gamma \sum_{i=0}^{n-1} \frac{1}{M} \sum_{m=1}^{M} \hat{g}_{t,m}^{\pi_m^i} = x_t - \tau \frac{1}{Mn} \sum_{i=0}^{n-1} \sum_{m=1}^{M} \hat{g}_{t,m}^{\pi_m^i}.$$

We denote $\hat{g}_t = \frac{1}{Mn} \sum_{i=0}^{n-1} \sum_{m=1}^{M} \hat{g}_{t,m}^{\pi_m^i}$.

**Lemma C.2.** *Let Assumptions 1, 2, 3, 5 hold. Then, the following inequality holds*

$$-2\tau \mathbb{E}_{\mathcal{Q}} \left[ \langle \hat{g}_t - h_\star, x_t - x_\star \rangle \right] \leq -\frac{\tau \mu}{2} \|x_t - x_\star\|^2 - \tau \left( f(x_t) - f(x_\star) \right) + \tau \widetilde{L} \frac{1}{n} \sum_{i=1}^{n-1} \|x_t - x_t^i\|^2,$$

*where $h^\star = \nabla f(x_\star) = 0$.*

*Proof.* Since $h^\star = \nabla f(x_\star) = 0$, the proof of Lemma C.2 is identical to the proof of Lemma B.2. □

**Lemma C.3.** *Let Assumptions 1, 2, 3, 5 hold. Then, the following inequality holds*

$$
\begin{aligned}
\mathbb{E}_{\mathcal{Q}} \left[ \|\hat{g}_t - h_\star\|^2 \right] &\leq 2\widetilde{L} \left( \widetilde{L} + \frac{\omega}{Mn} L_{\max} \right) \frac{1}{n} \sum_{i=0}^{n-1} \|x_t^i - x_t\|^2 + 8 \left( \widetilde{L} + \frac{\omega}{Mn} L_{\max} \right) (f(x_t) - f(x_\star)) \\
&\quad + \frac{4\omega}{M^2 n^2} \sum_{i=0}^{n-1} \sum_{m=1}^{M} \|h_{t,m}^{\pi_m^i} - \nabla f_m^{\pi_m^i}(x_\star)\|^2
\end{aligned}
$$

*Proof.* Taking expectation w.r.t. $\mathcal{Q}$, we get

$$
\begin{aligned}
\mathbb{E}_{\mathcal{Q}} \left[ \|\hat{g}_t - h_\star\|^2 \right] &= \mathbb{E}_{\mathcal{Q}} \left[ \left\| \frac{1}{Mn} \sum_{i=0}^{n-1} \sum_{m=1}^{M} \hat{g}_{t,m}^{\pi_m^i} - h_\star \right\|^2 \right] \\
&= \mathbb{E}_{\mathcal{Q}} \left[ \left\| \frac{1}{Mn} \sum_{i=0}^{n-1} \sum_{m=1}^{M} \left( h_{t,m}^{\pi_m^i} + \mathcal{Q} \left( \nabla f_m^{\pi_m^i}(x_t^i) - h_{t,m}^{\pi_m^i} \right) \right) - h_\star \right\|^2 \right] \\
&= \mathbb{E}_{\mathcal{Q}} \left[ \left\| \frac{1}{Mn} \sum_{i=0}^{n-1} \sum_{m=1}^{M} \left( h_{t,m}^{\pi_m^i} - \nabla f_m^{\pi_m^i}(x_t^i) + \mathcal{Q} \left( \nabla f_m^{\pi_m^i}(x_t^i) - h_{t,m}^{\pi_m^i} \right) \right) \right\|^2 \right] \\
&\quad + \left\| \frac{1}{Mn} \sum_{i=0}^{n-1} \sum_{m=1}^{M} \nabla f_m^{\pi_m^i}(x_t^i) - h_\star \right\|^2.
\end{aligned}
$$

Independence of $\mathcal{Q} \left( \nabla f_m^{\pi_m^i}(x_t^i) - h_{t,m}^{\pi_m^i} \right)$, $m \in [M]$ and Assumption 1 imply

$$
\begin{aligned}
\mathbb{E}_{\mathcal{Q}} \left[ \|\hat{g}_t - h_\star\|^2 \right] &= \frac{1}{M^2 n^2} \sum_{i=0}^{n-1} \sum_{m=1}^{M} \mathbb{E}_{\mathcal{Q}} \left[ \left\| h_{t,m}^{\pi_m^i} - \nabla f_m^{\pi_m^i}(x_t^i) + \mathcal{Q} \left( \nabla f_m^{\pi_m^i}(x_t^i) - h_{t,m}^{\pi_m^i} \right) \right\|^2 \right] \\
&\quad + \left\| \frac{1}{Mn} \sum_{i=0}^{n-1} \sum_{m=1}^{M} \nabla f_m^{\pi_m^i}(x_t^i) - h_\star \right\|^2 \\
&\leq \frac{\omega}{M^2 n^2} \sum_{i=0}^{n-1} \sum_{m=1}^{M} \left\| \nabla f_m^{\pi_m^i}(x_t^i) - h_{t,m}^{\pi_m^i} \right\|^2 + \left\| \frac{1}{n} \sum_{i=0}^{n-1} \nabla f^{\pi^i}(x_t^i) - h_\star \right\|^2 \\
&\leq \frac{2\omega}{M^2 n^2} \sum_{i=0}^{n-1} \sum_{m=1}^{M} \left\| \nabla f_m^{\pi_m^i}(x_t^i) - \nabla f_m^{\pi_m^i}(x_t) \right\|^2 + \frac{2}{n} \sum_{i=0}^{n-1} \left\| \nabla f^{\pi^i}(x_t^i) - \nabla f^{\pi^i}(x_t) \right\|^2 \\
&\quad + \frac{2\omega}{M^2 n^2} \sum_{i=0}^{n-1} \sum_{m=1}^{M} \left\| h_{t,m}^{\pi_m^i} - \nabla f_m^{\pi_m^i}(x_t) \right\|^2 + 2 \left\| \frac{1}{n} \sum_{i=0}^{n-1} \nabla f^{\pi^i}(x_t) - h_\star \right\|^2.
\end{aligned}
$$

Using $L_{\max}$-smoothness and convexity of $f_m^i$ and $\widetilde{L}$-smoothness and convexity of $f^{\pi^i}$, we obtain

$$
\begin{aligned}
\mathbb{E}_{\mathcal{Q}}\left[\|\hat{g}_t - h_\star\|^2\right] &\leq \frac{4\omega L_{\max}}{M^2 n^2} \sum_{i=0}^{n-1}\sum_{m=1}^{M} D_{f_m^{\pi_m^i}}(x_t^i, x_t) + \frac{4\widetilde{L}}{n}\sum_{i=0}^{n-1} D_{f^{\pi^i}}(x_t^i, x_t) \\
&\quad + \frac{4\omega}{M^2 n^2}\sum_{i=0}^{n-1}\sum_{m=1}^{M}\left\|h_{t,m}^{\pi_m^i} - \nabla f_m^{\pi_m^i}(x_\star)\right\|^2 + 4\widetilde{L}\left(f(x_t) - f(x_\star)\right) \\
&\quad + \frac{4\omega}{M^2 n^2}\sum_{i=0}^{n-1}\sum_{m=1}^{M}\left\|\nabla f_m^{\pi_m^i}(x_t) - \nabla f_m^{\pi_m^i}(x_\star)\right\|^2 \\
&\leq 2\widetilde{L}\left(\widetilde{L} + \frac{\omega}{Mn}L_{\max}\right)\frac{1}{n}\sum_{i=0}^{n-1}\|x_t^i - x_t\|^2 + 4\widetilde{L}\left(f(x_t) - f(x_\star)\right) \\
&\quad + \frac{8\omega}{Mn}L_{\max}\frac{1}{Mn}\sum_{i=0}^{n-1}\sum_{m=1}^{M} D_{f_m^{\pi_m^i}}(x_t, x_\star) \\
&\quad + \frac{4\omega}{M^2 n^2}\sum_{i=0}^{n-1}\sum_{m=1}^{M}\left\|h_{t,m}^{\pi_m^i} - \nabla f_m^{\pi_m^i}(x_\star)\right\|^2.
\end{aligned}
$$

$\square$

**Lemma C.4.** *Let $\alpha \leq \frac{1}{1+\omega}$ and Assumptions 1, 2, 3, 5 hold. Then, the iterates produced by* DIANA-RR *satisfy*

$$
\begin{aligned}
\frac{1}{Mn}\sum_{i=0}^{n-1}\sum_{m=1}^{M}\mathbb{E}_{\mathcal{Q}}\left[\|h_{t+1,m}^{\pi_m^i} - \nabla f_m^{\pi_m^i}(x_\star)\|^2\right] &\leq \frac{1-\alpha}{Mn}\sum_{i=0}^{n-1}\sum_{m=1}^{M}\|h_{t,m}^{\pi_m^i} - \nabla f_m^{\pi_m^i}(x_\star)\|^2 \\
&\quad + \frac{2\alpha\widetilde{L}L_{\max}}{n}\sum_{i=0}^{n-1}\|x_t^i - x_t\|^2 \\
&\quad + 4\alpha L_{\max}\left(f(x_t) - f(x^\star)\right).
\end{aligned}
$$

*Proof.* Fist of all, we introduce new notation: $\mathcal{H}_t = \frac{1}{Mn}\sum_{i=0}^{n-1}\sum_{m=1}^{M}\mathbb{E}_{\mathcal{Q}}\left[\|h_{t,m}^{\pi_m^i} - \nabla f_m^{\pi_m^i}(x_\star)\|^2\right]$. Using (20) and summing it up for $i = 0, \ldots, n-1$, we obtain

$$
\begin{aligned}
\mathcal{H}_{t+1} &\leq \frac{1-\alpha}{Mn}\sum_{i=0}^{n-1}\sum_{m=1}^{M}\|h_{t,m}^{\pi_m^i} - \nabla f_m^{\pi_m^i}(x_\star)\|^2 + \frac{\alpha}{Mn}\sum_{i=0}^{n-1}\sum_{m=1}^{M}\|\nabla f_m^{\pi_m^i}(x_t^i) - \nabla f_m^{\pi_m^i}(x_\star)\|^2 \\
&\leq \frac{1-\alpha}{Mn}\sum_{i=0}^{n-1}\sum_{m=1}^{M}\|h_{t,m}^{\pi_m^i} - \nabla f_m^{\pi_m^i}(x_\star)\|^2 + \frac{2\alpha}{Mn}\sum_{i=0}^{n-1}\sum_{m=1}^{M}\|\nabla f_m^{\pi_m^i}(x_t^i) - \nabla f_m^{\pi_m^i}(x_t)\|^2 \\
&\quad + \frac{2\alpha}{Mn}\sum_{i=0}^{n-1}\sum_{m=1}^{M}\|\nabla f_m^{\pi_m^i}(x_t) - \nabla f_m^{\pi_m^i}(x_\star)\|^2.
\end{aligned}
$$

Next, we apply $L_{\max}$-smoothness and convexity of $f_m^i$ and $\widetilde{L}$-smoothness and convexity of $f^{\pi^i}$:

$$
\begin{aligned}
\mathcal{H}_{t+1} &\leq \frac{1-\alpha}{Mn}\sum_{i=0}^{n-1}\sum_{m=1}^{M}\|h_{t,m}^{\pi_m^i} - \nabla f_m^{\pi_m^i}(x_\star)\|^2 + \frac{4\alpha}{Mn}L_{\max}\sum_{i=0}^{n-1}\sum_{m=1}^{M} D_{f_m^{\pi_m^i}}(x_t^i, x_t) \\
&\quad + \frac{4\alpha}{Mn}L_{\max}\sum_{i=0}^{n-1}\sum_{m=1}^{M} D_{f_m^{\pi_m^i}}(x_t, x_\star) \\
&\leq \frac{1-\alpha}{Mn}\sum_{i=0}^{n-1}\sum_{m=1}^{M}\|h_{t,m}^{\pi_m^i} - \nabla f_m^{\pi_m^i}(x_\star)\|^2 + \frac{2\alpha}{n}\widetilde{L}L_{\max}\sum_{i=0}^{n-1}\|x_t^i - x_t\|^2 \\
&\quad + \frac{4\alpha}{Mn}L_{\max}\sum_{i=0}^{n-1}\sum_{m=1}^{M} D_{f_m^{\pi_m^i}}(x_t, x_\star).
\end{aligned}
$$

$\square$

**Lemma C.5.** *Let Assumptions [1], [2], [3], [5] and $\tau \leq \frac{1}{2\sqrt{\widetilde{L}\left(\widetilde{L}+\frac{\omega}{Mn}L_{\max}\right)}}$. Then, the following inequality holds*

$$
\frac{1}{n}\sum_{i=0}^{n-1}\mathbb{E}\left[\|x_t^i - x_t\|^2\right] \leq 24\tau^2\left(\widetilde{L}+\frac{\omega}{Mn}L_{\max}\right)\mathbb{E}\left[f(x_t)-f(x_\star)\right] + 8\tau^2\frac{\sigma_{\star,n}^2}{n}
$$
$$
+ 8\frac{\tau^2\omega}{M^2n^2}\sum_{i=0}^{n-1}\sum_{m=1}^{M}\mathbb{E}\left[\|h_{t,m}^{\pi_m^i}-\nabla f_m^{\pi_m^i}(x_\star)\|^2\right],
$$

*where $\sigma_{\star,n}^2 = \frac{1}{n}\sum_{i=1}^{n}\|\nabla f^i(x_\star)\|^2$.*

*Proof.* Since $x_t^i = x_t - \frac{\tau}{Mn}\sum_{m=1}^{M}\sum_{j=0}^{i-1}\left(h_{t,m}^{\pi_m^j} + \mathcal{Q}\left(\nabla f_m^{\pi_m^j}(x_t^j) - h_{t,m}^{\pi_m^i}\right)\right)$, we have

$$
\mathbb{E}_{\mathcal{Q}}\left[\|x_t^i - x_t\|^2\right] = \tau^2\mathbb{E}_{\mathcal{Q}}\left[\left\|\frac{1}{Mn}\sum_{m=1}^{M}\sum_{j=0}^{i-1}\left(h_{t,m}^{\pi_m^j} + \mathcal{Q}\left(\nabla f_m^{\pi_m^j}(x_t^j) - h_{t,m}^{\pi_m^j}\right)\right)\right\|^2\right]
$$
$$
= \tau^2\mathbb{E}_{\mathcal{Q}}\left[\left\|\frac{1}{Mn}\sum_{m=1}^{M}\sum_{j=0}^{i-1}\left(h_{t,m}^{\pi_m^j} - \nabla f_m^{\pi_m^j}(x_t^j) + \mathcal{Q}\left(\nabla f_m^{\pi_m^i}(x_t^j) - h_{t,m}^{\pi_m^j}\right)\right)\right\|^2\right]
$$
$$
+ \tau^2\left\|\frac{1}{Mn}\sum_{m=1}^{M}\sum_{j=0}^{i-1}\nabla f_m^{\pi_m^j}(x_t^j)\right\|^2.
$$

Independence of $\mathcal{Q}\left(\nabla f_m^{\pi_m^i}(x_t^j) - h_{t,m}^{\pi_m^j}\right)$, $m \in [M]$ and Assumption [1] imply

$$
\mathbb{E}_{\mathcal{Q}}\left[\|x_t^i - x_t\|^2\right] = \frac{\tau^2}{M^2n^2}\sum_{m=1}^{M}\sum_{j=0}^{i-1}\mathbb{E}_{\mathcal{Q}}\left[\left\|h_{t,m}^{\pi_m^j} - \nabla f_m^{\pi_m^j}(x_t^j) + \mathcal{Q}\left(\nabla f_m^{\pi_m^i}(x_t^j) - h_{t,m}^{\pi_m^j}\right)\right\|^2\right]
$$
$$
+ \tau^2\left\|\frac{1}{Mn}\sum_{m=1}^{M}\sum_{j=0}^{i-1}\nabla f_m^{\pi_m^j}(x_t^j)\right\|^2
$$
$$
\leq \frac{\tau^2\omega}{M^2n^2}\sum_{m=1}^{M}\sum_{j=0}^{i-1}\left\|\nabla f^{\pi_m^j}(x_t^j) - h_{t,m}^{\pi_m^j}\right\|^2 + \tau^2\left\|\frac{1}{n}\sum_{j=0}^{i-1}\nabla f^{\pi^j}(x_t^j)\right\|^2
$$
$$
\leq \frac{2\tau^2\omega}{M^2n^2}\sum_{m=1}^{M}\sum_{j=0}^{n-1}\left\|\nabla f_m^{\pi_m^j}(x_t^j) - \nabla f_m^{\pi_m^j}(x_t)\right\|^2 + 2\tau^2\left\|\frac{1}{n}\sum_{j=0}^{i-1}\nabla f^{\pi^j}(x_t)\right\|^2
$$
$$
+ \frac{2\tau^2\omega}{M^2n^2}\sum_{m=1}^{M}\sum_{j=0}^{n-1}\left\|h_{t,m}^{\pi_m^j} - \nabla f_m^{\pi_m^j}(x_t)\right\|^2 + \frac{2\tau^2}{n}\sum_{j=0}^{n-1}\left\|\nabla f^{\pi^j}(x_t^j) - \nabla f^{\pi^j}(x_t)\right\|^2.
$$

Using $L_{\max}$-smoothness and convexity of $f_m^i$ and $\widetilde{L}$-smoothness and convexity of $f^{\pi^j}$, we obtain

$$
\begin{aligned}
\mathbb{E}_{\mathcal{Q}}\left[\|x_t^i - x_t\|^2\right] &\leq \frac{4\tau^2\omega}{M^2n^2}L_{\max}\sum_{m=1}^{M}\sum_{j=0}^{n-1} D_{f_m^{\pi_m^j}}(x_t^j, x_t) + 2\tau^2\left\|\frac{1}{n}\sum_{j=0}^{i-1}\nabla f^{\pi^j}(x_t)\right\|^2 \\
&\quad + \frac{2\tau^2\omega}{M^2n^2}\sum_{m=1}^{M}\sum_{j=0}^{n-1}\left\|h_{t,m}^{\pi_m^j} - \nabla f_m^{\pi_m^j}(x_t)\right\|^2 + \frac{2\tau^2\widetilde{L}^2}{n}\sum_{j=0}^{n-1}\left\|x_t^j - x_t\right\|^2 \\
&\leq 2\tau^2\widetilde{L}\left(\widetilde{L} + \frac{\omega}{Mn}L_{\max}\right)\frac{1}{n}\sum_{j=0}^{n-1}\|x_t^j - x_t\|^2 + 2\tau^2\left\|\frac{1}{n}\sum_{j=0}^{i-1}\nabla f^{\pi^j}(x_t)\right\|^2 \\
&\quad + \frac{2\tau^2\omega}{M^2n^2}\sum_{m=1}^{M}\sum_{j=0}^{n-1}\left\|h_{t,m}^{\pi_m^j} - \nabla f_m^{\pi_m^j}(x_t)\right\|^2.
\end{aligned}
$$

Taking the full expectation and using (16), we derive

$$
\begin{aligned}
\mathbb{E}\left[\|x_t^i - x_t\|^2\right] &\leq 2\tau^2\widetilde{L}\left(\widetilde{L} + \frac{\omega}{Mn}L_{\max}\right)\frac{1}{n}\sum_{j=0}^{n-1}\mathbb{E}\left[\|x_t^j - x_t\|^2\right] + 2\tau^2\mathbb{E}\left[\|\nabla f(x_t)\|^2\right] \\
&\quad + \frac{4\tau^2\omega}{M^2n^2}\sum_{m=1}^{M}\sum_{j=0}^{n-1}\mathbb{E}\left[\left\|h_{t,m}^{\pi_m^j} - \nabla f_m^{\pi_m^j}(x_\star)\right\|^2\right] + \frac{2\tau^2}{n}\mathbb{E}\left[\sigma_t^2\right] \\
&\quad + \frac{8\tau^2\omega}{M^2n^2}L_{\max}\sum_{m=1}^{M}\sum_{j=0}^{n-1}\mathbb{E}\left[D_{f_m^{\pi_m^j}}(x_t, x_\star)\right].
\end{aligned}
$$

Using $L_{\max}$-smoothness and convexity of $f_m^i$ and $\widetilde{L}$-smoothness and convexity of $f^{\pi^j}$, we obtain

$$
\begin{aligned}
\mathbb{E}\left[\|x_t^i - x_t\|^2\right] &\leq 2\tau^2\widetilde{L}\left(\widetilde{L} + \frac{\omega}{Mn}L_{\max}\right)\frac{1}{n}\sum_{j=0}^{n-1}\mathbb{E}\left[\|x_t^j - x_t\|^2\right] \\
&\quad + \frac{4\tau^2\omega}{M^2n^2}\sum_{m=1}^{M}\sum_{j=0}^{n-1}\mathbb{E}\left[\left\|h_{t,m}^{\pi_m^j} - \nabla f_m^{\pi_m^j}(x_\star)\right\|^2\right] + \frac{2\tau^2}{n}\mathbb{E}\left[\sigma_t^2\right] \\
&\quad + 4\tau^2\left(\widetilde{L} + \frac{2\omega}{M^2n^2}L_{\max}\right)\mathbb{E}\left[f(x_t) - f(x_\star)\right].
\end{aligned}
$$

Now we need to estimate $\frac{2\tau^2}{n}\mathbb{E}\left[\sigma_t^2\right]$. Due to $\mathbb{E}\left[\sigma_t^2\right] \leq \frac{1}{n}\sum_{i=1}^{n}\mathbb{E}\left[\|\nabla f^i(x_t)\|^2\right]$, we get

$$
\begin{aligned}
\frac{2\tau^2}{n}\mathbb{E}\left[\sigma_t^2\right] &\leq \frac{2\tau^2}{n^2}\sum_{j=1}^{n}\mathbb{E}\left[\|\nabla f^j(x_t)\|^2\right] \\
&\leq \frac{4\tau^2}{n^2}\sum_{j=1}^{n}\mathbb{E}\left[\|\nabla f^j(x_t) - \nabla f^j(x_\star)\|^2\right] + \frac{4\tau^2}{n^2}\sum_{j=1}^{n}\mathbb{E}\left[\|\nabla f^j(x_\star)\|^2\right] \\
&\leq \frac{8\tau^2}{n^2}\widetilde{L}\sum_{j=1}^{n}\mathbb{E}\left[D_{f^j}(x_t, x_\star)\right] + \frac{4\tau^2}{n^2}\sum_{j=1}^{n}\sigma_{n,\star}^2.
\end{aligned}
$$

Combining two previous inequalities, we get

$$
\begin{aligned}
\mathbb{E}\left[\|x_t^i - x_t\|^2\right] \;\le\; & 2\tau^2\widetilde{L}\left(\widetilde{L} + \frac{\omega}{Mn}L_{\max}\right)\frac{1}{n}\sum_{j=0}^{n-1}\mathbb{E}\left[\|x_t^j - x_t\|^2\right] \\
& + \frac{4\tau^2\omega}{M^2n^2}\sum_{m=1}^{M}\sum_{j=0}^{n-1}\mathbb{E}\left[\left\|h_{t,m}^{\pi_m^j} - \nabla f_m^{\pi_m^j}(x_\star)\right\|^2\right] \\
& + 4\tau^2\left(\widetilde{L} + \frac{2\omega}{M^2n^2}L_{\max}\right)\mathbb{E}\left[f(x_t) - f(x_\star)\right] \\
& + \frac{8\tau^2}{n}\widetilde{L}\mathbb{E}\left[f(x_t) - f(x_\star)\right] + \frac{4\tau^2}{n^2}\sum_{j=1}^{n}\sigma_{n,\star}^2.
\end{aligned}
$$

Summing from $i = 0$ to $n-1$ and using $\tau \le \dfrac{1}{2\sqrt{\widetilde{L}\left(\widetilde{L} + \frac{\omega}{Mn}L_{\max}\right)}}$, we obtain

$$
\begin{aligned}
\frac{1}{n}\sum_{i=0}^{n-1}\mathbb{E}\left[\|x_t^i - x_t\|^2\right] \;\le\; & 2\left(1 - 2\tau^2\widetilde{L}\left(\widetilde{L} + \frac{\omega}{Mn}L_{\max}\right)\right)\frac{1}{n}\sum_{i=0}^{n-1}\mathbb{E}\left[\|x_t^i - x_t\|^2\right] \\
\le\; & \frac{8\tau^2\omega}{M^2n^2}\sum_{m=1}^{M}\sum_{j=0}^{n-1}\mathbb{E}\left[\left\|h_{t,m}^{\pi_m^j} - \nabla f_m^{\pi_m^j}(x_\star)\right\|^2\right] \\
& + 8\tau^2\left(\widetilde{L} + \frac{2\omega}{M^2n^2}L_{\max}\right)\mathbb{E}\left[f(x_t) - f(x_\star)\right] \\
& + \frac{16\tau^2}{n}\widetilde{L}\mathbb{E}\left[f(x_t) - f(x_\star)\right] + \frac{8\tau^2}{n^2}\sum_{j=1}^{n}\sigma_{n,\star}^2.
\end{aligned}
$$

$\qquad\square$

We consider the following Lyapunov function:

$$
\Psi_{t+1} = \|x_{t+1} - x_\star\|^2 + \frac{c\tau^2}{Mn}\sum_{m=1}^{M}\sum_{j=0}^{n-1}\left\|h_{t+1,m}^{\pi_m^i} - \nabla f_m^{\pi_m^i}(x_\star)\right\|^2. \tag{22}
$$

**Theorem C.2.** *Let Assumptions 1, 2, 3, 5 hold and*

$$
\gamma \le \min\left\{\frac{\alpha}{n\mu},\; \frac{1}{12n\left(\widetilde{L} + \frac{11\omega}{Mn}L_{\max}\right)}\right\}, \quad \alpha \le \frac{1}{1+\omega}, \quad c = \frac{10\omega}{\alpha Mn}.
$$

*Then, for all $T \ge 0$ the iterates produced by* DIANA-RR *satisfy*

$$
\mathbb{E}\left[\Psi_T\right] \le \left(1 - \frac{n\gamma\mu}{2}\right)^T\Psi_0 + 20\frac{\gamma^2 n\widetilde{L}}{\mu}\sigma_{\star,n}^2.
$$

*Proof.* Taking expectation w.r.t. $\mathcal{Q}$ and using Lemma C.2, we get

$$
\begin{aligned}
\mathbb{E}_\mathcal{Q}\left[\|x_{t+1} - x_\star\|^2\right] \;=\; & \|x_t - \tau\hat{g}_t - x_\star + \tau h^\star\|^2 \\
=\; & \|x_t - x_\star\|^2 - 2\tau\mathbb{E}_\mathcal{Q}\left[\langle\hat{g}_t - h^\star, x_t - x_\star\rangle\right] + \tau^2\mathbb{E}_\mathcal{Q}\left[\|\hat{g}_t - h^\star\|^2\right] \\
\le\; & \|x_t - x_\star\|^2 - \frac{\tau\mu}{2}\|x_t - x_\star\|^2 + \tau^2\mathbb{E}_\mathcal{Q}\left[\|\hat{g}_t - h^\star\|^2\right] \\
& - \tau\left(f(x_t) - f(x_\star)\right) + \tau\widetilde{L}\frac{1}{n}\sum_{i=1}^{n-1}\|x_t - x_t^i\|^2.
\end{aligned}
$$

Next, due to Lemma C.3 we have

$$
\begin{aligned}
\mathbb{E}_{\mathcal{Q}}\left[\|x_{t+1}-x_\star\|^2\right] \leq\ & \left(1-\frac{\tau\mu}{2}\right)\|x_t-x_\star\|^2 - \tau\left(f(x_t)-f(x_\star)\right) + \tau\widetilde{L}\frac{1}{n}\sum_{i=1}^{n-1}\|x_t-x_t^i\|^2 \\
& +2\tau^2\widetilde{L}\left(\widetilde{L}+\frac{\omega}{Mn}L_{\max}\right)\frac{1}{n}\sum_{i=0}^{n-1}\|x_t^i-x_t\|^2 \\
& +8\tau^2\left(\widetilde{L}+\frac{\omega}{Mn}L_{\max}\right)\left(f(x_t)-f(x_\star)\right) \\
& +\frac{4\omega\tau^2}{M^2n^2}\sum_{i=0}^{n-1}\sum_{m=1}^{M}\|h_{t,m}^{\pi_m^i}-\nabla f_m^{\pi_m^i}(x_\star)\|^2 \\
\leq\ & \left(1-\frac{\tau\mu}{2}\right)\|x_t-x_\star\|^2 + \frac{4\omega\tau^2}{M^2n^2}\sum_{i=0}^{n-1}\sum_{m=1}^{M}\|h_{t,m}^{\pi_m^i}-\nabla f_m^{\pi_m^i}(x_\star)\|^2 \\
& -\tau\left(1-8\tau\left(\widetilde{L}+\frac{\omega}{Mn}L_{\max}\right)\right)\left(f(x_t)-f(x_\star)\right) \\
& +\tau\widetilde{L}\left(1+2\tau\left(\widetilde{L}+\frac{\omega}{Mn}L_{\max}\right)\right)\frac{1}{n}\sum_{i=0}^{n-1}\|x_t^i-x_t\|^2.
\end{aligned}
$$

Using (22), we obtain

$$
\begin{aligned}
\mathbb{E}_{\mathcal{Q}}\left[\Psi_{t+1}\right] \leq\ & \left(1-\frac{\tau\mu}{2}\right)\|x_t-x_\star\|^2 + \frac{4\omega\tau^2}{M^2n^2}\sum_{i=0}^{n-1}\sum_{m=1}^{M}\|h_{t,m}^{\pi_m^i}-\nabla f_m^{\pi_m^i}(x_\star)\|^2 \\
& -\tau\left(1-8\tau\left(\widetilde{L}+\frac{\omega}{Mn}L_{\max}\right)\right)\left(f(x_t)-f(x_\star)\right) \\
& +\tau\widetilde{L}\left(1+2\tau\left(\widetilde{L}+\frac{\omega}{Mn}L_{\max}\right)\right)\frac{1}{n}\sum_{i=0}^{n-1}\|x_t^i-x_t\|^2 \\
& +\frac{c\tau^2}{Mn}\sum_{m=1}^{M}\sum_{j=0}^{n-1}\mathbb{E}\left[\left\|h_{t+1,m}^{\pi_m^i}-\nabla f_m^{\pi_m^i}(x_\star)\right\|^2\right].
\end{aligned}
$$

To estimate the last term in the above inequality, we apply Lemma C.4:

$$
\begin{aligned}
\mathbb{E}_{\mathcal{Q}}\left[\Psi_{t+1}\right] \leq\ & \left(1-\frac{\tau\mu}{2}\right)\|x_t-x_\star\|^2 + \frac{4\omega\tau^2}{M^2n^2}\sum_{i=0}^{n-1}\sum_{m=1}^{M}\|h_{t,m}^{\pi_m^i}-\nabla f_m^{\pi_m^i}(x_\star)\|^2 \\
& -\tau\left(1-8\tau\left(\widetilde{L}+\frac{\omega}{Mn}L_{\max}\right)\right)\left(f(x_t)-f(x_\star)\right) \\
& +\tau\widetilde{L}\left(1+2\tau\left(\widetilde{L}+\frac{\omega}{Mn}L_{\max}\right)\right)\frac{1}{n}\sum_{i=0}^{n-1}\|x_t^i-x_t\|^2 \\
& +c\tau^2\frac{1-\alpha}{Mn}\sum_{i=0}^{n-1}\sum_{m=1}^{M}\|h_{t,m}^{\pi_m^i}-\nabla f_m^{\pi_m^i}(x_\star)\|^2 \\
& +c\tau^2\frac{2\alpha\widetilde{L}L_{\max}}{n}\sum_{i=0}^{n-1}\|x_t^i-x_t\|^2 + 4c\tau^2\alpha L_{\max}\left(f(x_t)-f(x^\star)\right) \\
\leq\ & \left(1-\frac{\tau\mu}{2}\right)\|x_t-x_\star\|^2 + \left(1-\alpha+\frac{4\omega}{cMn}\right)\frac{c\tau^2}{Mn}\sum_{i=0}^{n-1}\sum_{m=1}^{M}\|h_{t,m}^{\pi_m^i}-\nabla f_m^{\pi_m^i}(x_\star)\|^2 \\
& -\tau\left(1-4c\tau\alpha L_{\max}-8\tau\left(\widetilde{L}+\frac{\omega}{Mn}L_{\max}\right)\right)\left(f(x_t)-f(x_\star)\right) \\
& +\tau\widetilde{L}\left(1+2c\tau\alpha L_{\max}+2\tau\left(\widetilde{L}+\frac{\omega}{Mn}L_{\max}\right)\right)\frac{1}{n}\sum_{i=0}^{n-1}\|x_t^i-x_t\|^2.
\end{aligned}
$$

Let $\mathcal{H}_t = \frac{c\tau^2}{Mn} \sum_{i=0}^{n-1} \sum_{m=1}^{M} \mathbb{E}\left[\|h_{t,m}^{\pi_m^i} - \nabla f_m^{\pi_m^i}(x_\star)\|^2\right]$. Taking the full expectation and using Lemma C.5, we get

$$
\begin{aligned}
\mathbb{E}\left[\Psi_{t+1}\right] \leq\ & \left(1 - \frac{\tau\mu}{2}\right) \mathbb{E}\left[\|x_t - x_\star\|^2\right] + \left(1 - \alpha + \frac{4\omega}{cMn}\right) \mathcal{H}_t \\
& -\tau\left(1 - 4c\tau\alpha L_{\max} - 8\tau\left(\widetilde{L} + \frac{\omega}{Mn}L_{\max}\right)\right) \mathbb{E}\left[f(x_t) - f(x_\star)\right] \\
& + \tau\widetilde{L}\left(1 + 2c\tau\alpha L_{\max} + 2\tau\left(\widetilde{L} + \frac{\omega}{Mn}L_{\max}\right)\right) \frac{1}{n} \sum_{i=0}^{n-1} \mathbb{E}\left[\|x_t^i - x_t\|^2\right] \\
\leq\ & \left(1 - \frac{\tau\mu}{2}\right) \mathbb{E}\left[\|x_t - x_\star\|^2\right] + \left(1 - \alpha + \frac{4\omega}{cMn}\right) \mathcal{H}_t \\
& -\tau\left(1 - 4c\tau\alpha L_{\max} - 8\tau\left(\widetilde{L} + \frac{\omega}{Mn}L_{\max}\right)\right) \mathbb{E}\left[f(x_t) - f(x_\star)\right] \\
& + 24\tau^3\widetilde{L}\left(1 + 2c\tau\alpha L_{\max} + 2\tau\left(\widetilde{L} + \frac{\omega}{Mn}L_{\max}\right)\right)\left(\widetilde{L} + \frac{\omega}{Mn}L_{\max}\right) \mathbb{E}\left[f(x_t) - f(x_\star)\right] \\
& + 8\tau^3\widetilde{L}\left(1 + 2c\tau\alpha L_{\max} + 2\tau\left(\widetilde{L} + \frac{\omega}{Mn}L_{\max}\right)\right) \frac{\sigma_{\star,n}^2}{n} \\
& + \frac{8\tau\widetilde{L}\omega}{cMn}\left(1 + 2c\tau\alpha L_{\max} + 2\tau\left(\widetilde{L} + \frac{\omega}{Mn}L_{\max}\right)\right) \mathcal{H}_t.
\end{aligned}
$$

Selecting $c = \frac{A\omega}{\alpha Mn}$, where $A$ is a positive number to be specified later, we have

$$
1 + 2c\tau\alpha L_{\max} + 2\tau\left(\widetilde{L} + \frac{\omega}{Mn}L_{\max}\right) = 1 + 2\tau\left(\widetilde{L} + \frac{(A+1)\omega}{Mn}L_{\max}\right),
$$

$$
1 - 4c\tau\alpha L_{\max} - 8\tau\left(\widetilde{L} + \frac{\omega}{Mn}L_{\max}\right) \geq 1 - 8\tau\left(\widetilde{L} + \frac{(A+1)\omega}{Mn}L_{\max}\right).
$$

Then, we have

$$
\begin{aligned}
\mathbb{E}\left[\Psi_{t+1}\right] \leq\ & \left(1 - \frac{\tau\mu}{2}\right) \mathbb{E}\left[\|x_t - x_\star\|^2\right] + \left(1 - \alpha + \frac{4\alpha}{A}\right) \mathcal{H}_t \\
& -\tau\left(1 - 8\tau\left(\widetilde{L} + \frac{(A+1)\omega}{Mn}L_{\max}\right)\right) \mathbb{E}\left[f(x_t) - f(x_\star)\right] \\
& + 24\tau^3\widetilde{L}\left(\widetilde{L} + \frac{\omega}{Mn}L_{\max}\right)\left(1 + 2\tau\left(\widetilde{L} + \frac{(A+1)\omega}{Mn}L_{\max}\right)\right) \mathbb{E}\left[f(x_t) - f(x_\star)\right] \\
& + 8\tau^3\widetilde{L}\left(1 + 2\tau\left(\widetilde{L} + \frac{(A+1)\omega}{Mn}L_{\max}\right)\right) \frac{\sigma_{\star,n}^2}{n} \\
& + \frac{8\alpha}{A}\tau\widetilde{L}\left(1 + 2\tau\left(\widetilde{L} + \frac{(A+1)\omega}{Mn}L_{\max}\right)\right) \mathcal{H}_t.
\end{aligned}
$$

Taking $\tau = \frac{1}{B\left(\widetilde{L} + \frac{(A+1)\omega}{Mn}L_{\max}\right)}$, where $B$ is some positive constant, we obtain

$$
\begin{aligned}
\mathbb{E}\left[\Psi_{t+1}\right] \leq\ & \left(1 - \frac{\tau\mu}{2}\right) \mathbb{E}\left[\|x_t - x_\star\|^2\right] + \left(1 - \alpha + \frac{4\alpha}{A} + \frac{8\alpha}{A}\tau\widetilde{L}\left(1 + \frac{2}{B}\right)\right) \mathcal{H}_t \\
& -\tau\left(1 - \frac{8}{B} - \frac{24}{B^2}\left(1 + \frac{2}{B}\right)\right) \mathbb{E}\left[f(x_t) - f(x_\star)\right] \\
& + 8\tau^3\widetilde{L}\left(1 + \frac{2}{B}\right) \frac{\sigma_{\star,n}^2}{n}.
\end{aligned}
$$

Choosing $A = 10$, $B = 12$, $\tau \leq \frac{\alpha}{\mu}$, we have

$$
\begin{aligned}
\mathbb{E}\left[\Psi_{t+1}\right] \leq\ & \left(1 - \min\left\{\frac{\tau\mu}{2}, \frac{\alpha}{2}\right\}\right) \mathbb{E}\left[\Psi_t\right] + 10\tau^3\widetilde{L}\frac{\sigma_{\star,n}^2}{n} \\
\leq\ & \left(1 - \frac{\tau\mu}{2}\right) \mathbb{E}\left[\Psi_t\right] + 10\tau^3\widetilde{L}\frac{\sigma_{\star,n}^2}{n}
\end{aligned}
$$

Recursively unrolling the inequality, substituting $\tau = n\gamma$ and using $\sum\limits_{t=0}^{+\infty} \left(1 - \frac{\tau\mu}{2}\right)^t \leq \frac{2}{\mu\tau}$, we finish proof. $\qquad\square$

**Corollary 8.** *Let the assumptions of Theorem C.2 hold, $\alpha = \frac{1}{1+\omega}$, and*

$$\gamma = \min\left\{\frac{\alpha}{2n\mu}, \frac{1}{12n\left(\widetilde{L} + \frac{11\omega}{Mn}L_{\max}\right)}, \sqrt{\frac{\varepsilon\mu}{40n\widetilde{L}\sigma_{\star,n}^2}}\right\}. \tag{23}$$

*Then, DIANA-RR finds a solution with accuracy $\varepsilon > 0$ after the following number of communication rounds:*

$$\widetilde{\mathcal{O}}\left(n(1+\omega) + \frac{n\widetilde{L}}{\mu} + \frac{\omega}{M}\frac{L_{\max}}{\mu} + \sqrt{\frac{n\widetilde{L}}{\varepsilon\mu^3}}\sigma_{\star,n}\right).$$

*Proof.* Theorem C.2 implies

$$\mathbb{E}\left[\Psi_T\right] \leq (1 - \gamma\mu)^{nT}\Psi_0 + 20\frac{\gamma^2 n\widetilde{L}}{\mu}\sigma_{\star,n}^2.$$

To estimate the number of communication rounds required to find a solution with accuracy $\varepsilon > 0$, we need to upper bound each term from the right-hand side by $\frac{\varepsilon}{2}$. Thus, we get an additional condition on $\gamma$:

$$20\frac{\gamma^2 n\widetilde{L}}{\mu}\sigma_{\star,n}^2 < \frac{\varepsilon}{2},$$

and also the upper bound on the number of communication rounds $nT$

$$nT = \widetilde{\mathcal{O}}\left(\frac{1}{\gamma\mu}\right).$$

Substituting (23) in the previous equation, we obtain the result. $\qquad\square$

# D    MISSING PROOFS FOR Q-NASTYA

We start with deriving a technical lemma along with stating several useful results from (Malinovsky et al., 2022). For convenience, we also introduce the following notation:

$$g_{t,m} = \frac{1}{n} \sum_{i=0}^{n-1} \nabla f_m^{\pi_m^i}(x_{t,m}^i).$$

**Lemma D.1.** *Let Assumptions 1, 2, 3 hold. Then, for all $t \geq 0$ the iterates produced by* Q-NASTYA *satisfy*

$$\mathbb{E}_{\mathcal{Q}}\left[\|g_t\|^2\right] \leq \frac{2L_{\max}^2\left(1+\frac{\omega}{M}\right)}{Mn} \sum_{m=1}^{M} \sum_{i=0}^{n-1} \|x_{t,m}^i - x_t\|^2 + 8L_{\max}\left(1+\frac{\omega}{M}\right)(f(x_t) - f(x_\star)) + \frac{4\omega}{M}\zeta_\star^2,$$

*where $\mathbb{E}_{\mathcal{Q}}$ is expectation w.r.t. $\mathcal{Q}$, and $\zeta_\star^2 = \frac{1}{M}\sum_{m=1}^{M}\|\nabla f_m(x_\star)\|^2$.*

*Proof.* Using the variance decomposition $\mathbb{E}\left[\|\xi\|^2\right] = \mathbb{E}\left[\|\xi - \mathbb{E}\left[\xi\right]\|^2\right] + \|\mathbb{E}\xi\|^2$, we obtain

$$
\begin{aligned}
\mathbb{E}_{\mathcal{Q}}\left[\|g_t\|^2\right] &= \frac{1}{M^2}\sum_{m=1}^{M}\mathbb{E}_{\mathcal{Q}}\left[\left\|\mathcal{Q}\left(\frac{1}{n}\sum_{i=0}^{n-1}\nabla f_m^{\pi_m^i}(x_{t,m}^i)\right) - \frac{1}{n}\sum_{i=0}^{n-1}\nabla f_m^{\pi_m^i}(x_{t,m}^i)\right\|^2\right] \\
&\quad + \left\|\frac{1}{Mn}\sum_{m=1}^{M}\sum_{i=0}^{n-1}\nabla f_m^{\pi_m^i}(x_{t,m}^i)\right\|^2 \\
&\overset{\text{Asm.1}}{\leq} \frac{\omega}{M^2}\sum_{m=1}^{M}\left\|\frac{1}{n}\sum_{i=0}^{n-1}\nabla f_m^{\pi_m^i}(x_{t,m}^i)\right\|^2 + \left\|\frac{1}{Mn}\sum_{m=1}^{M}\sum_{i=0}^{n-1}\nabla f_m^{\pi_m^i}(x_{t,m}^i)\right\|^2.
\end{aligned}
$$

Next, we use $\nabla f_m(x_t) = \frac{1}{n}\sum_{i=0}^{n-1}\nabla f_m^{\pi_m^i}(x_t)$ and $\|a+b\|^2 \leq 2\|a\|^2 + 2\|b\|^2$:

$$
\begin{aligned}
\mathbb{E}_{\mathcal{Q}}\left[\|g_t\|^2\right] &\leq \frac{2\omega}{M^2}\sum_{m=1}^{M}\left\|\frac{1}{n}\sum_{i=0}^{n-1}\left(\nabla f_m^{\pi_m^i}(x_{t,m}^i) - \nabla f_m^{\pi_m^i}(x_t)\right)\right\|^2 + \frac{2\omega}{M^2}\sum_{m=1}^{M}\|\nabla f_m(x_t)\|^2 \\
&\quad + 2\left\|\frac{1}{Mn}\sum_{m=1}^{M}\sum_{i=0}^{n-1}\left(\nabla f_m^{\pi_m^i}(x_{t,m}^i) - \nabla f_m^{\pi_m^i}(x_t)\right)\right\|^2 + 2\left\|\frac{1}{M}\sum_{m=1}^{M}\nabla f_m(x_t)\right\|^2 \\
&\leq \frac{2\left(1+\frac{\omega}{M}\right)}{M}\sum_{m=1}^{M}\left\|\frac{1}{n}\sum_{i=0}^{n-1}\left(\nabla f_m^{\pi_m^i}(x_{t,m}^i) - \nabla f_m^{\pi_m^i}(x_t)\right)\right\|^2 \\
&\quad + \frac{2\omega}{M^2}\sum_{m=1}^{M}\|\nabla f_m(x_t)\|^2 + 2\|\nabla f(x_t)\|^2.
\end{aligned}
$$

Using $L_{i,m}$-smoothness of $f_m^i$ and $f$ and also convexity of $f_m$, we obtain

$$
\begin{aligned}
\mathbb{E}_{\mathcal{Q}}\left[\|g_t\|^2\right] &\leq \frac{2\left(1+\frac{\omega}{M}\right)}{Mn}\sum_{m=1}^{M}\sum_{i=0}^{n-1}\left\|\nabla f_m^{\pi_m^i}(x_{t,m}^i) - \nabla f_m^{\pi_m^i}(x_t)\right\|^2 + \frac{4\omega}{M^2}\sum_{m=1}^{M}\|\nabla f_m(x_t) - \nabla f_m(x_\star)\|^2 \\
&\quad + \frac{4\omega}{M^2}\sum_{m=1}^{M}\|\nabla f_m(x_\star)\|^2 + 2\|\nabla f(x_t) - \nabla f(x_\star)\|^2 \\
&\leq \frac{2L_{\max}^2\left(1+\frac{\omega}{M}\right)}{Mn}\sum_{m=1}^{M}\sum_{i=0}^{n-1}\|x_{t,m}^i - x_t\|^2 + \frac{8L_{\max}\left(1+\frac{\omega}{M}\right)}{M}\sum_{m=1}^{M}D_{f_m}(x_t, x_\star) + \frac{4\omega}{M}\zeta_\star^2.
\end{aligned}
$$

$\square$

**Lemma D.2** (see (Malinovsky et al., 2022)). *Under Assumptions 1, 2, 3, it holds*

$$-\frac{1}{Mn}\sum_{m=1}^{M}\sum_{i=0}^{n-1}\left\langle f_m^{\pi_m^i}(x_{t,m}^i), x_t - x_\star \right\rangle \leq -\frac{\mu}{4}\|x_t - x_\star\|^2 - \frac{1}{2}\left(f(x_t) - f(x_\star)\right) + \frac{L_{\max}}{2Mn}\sum_{m=1}^{M}\sum_{i=0}^{n-1}\left\|x_{t,m}^i - x_t\right\|^2.$$

**Lemma D.3** (see (Malinovsky et al., 2022)). *Under Assumptions 1, 2, 3 and $\gamma \leq \frac{1}{2L_{\max}n}$, it holds*

$$\frac{1}{Mn}\sum_{m=1}^{M}\sum_{i=0}^{n-1}\left\|x_{t,m}^i - x_t\right\|^2 \leq 8\gamma^2 n^2 L_{\max}\left(f(x_t) - f(x_\star)\right) + 2\gamma^2 n\left(\sigma_\star^2 + (n+1)\zeta_\star^2\right).$$

**Theorem D.1.** *Let Assumptions 1, 2, 3 hold and stepsizes $\gamma$, $\eta$ satisfy*

$$0 < \eta \leq \frac{1}{16L_{\max}\left(1 + \frac{\omega}{M}\right)}, \quad 0 < \gamma \leq \frac{1}{5nL_{\max}}. \tag{24}$$

*Then, for all $T \geq 0$ the iterates produced by* Q-NASTYA *satisfy*

$$\mathbb{E}\left[\|x_T - x_\star\|^2\right] \leq \left(1 - \frac{\eta\mu}{2}\right)^T \|x_0 - x_\star\|^2 + \frac{9}{2}\frac{\gamma^2 n L_{\max}}{\mu}\left(\sigma_\star^2 + (n+1)\zeta_\star^2\right) + 8\frac{\eta\omega}{\mu M}\zeta_\star^2.$$

*Proof.* Taking expectation w.r.t. $\mathcal{Q}$ and using Lemma D.1, we get

$$
\begin{aligned}
\mathbb{E}_{\mathcal{Q}}\left[\|x_{t+1} - x_\star\|^2\right] &= \|x_t - x_\star\|^2 - 2\eta\mathbb{E}_{\mathcal{Q}}\left[\langle g_t, x_t - x_\star\rangle\right] + \eta^2\mathbb{E}_{\mathcal{Q}}\left[\|g^t\|^2\right] \\
&\leq \|x_t - x_\star\|^2 - 2\eta\mathbb{E}_{\mathcal{Q}}\left[\left\langle \frac{1}{M}\sum_{m=1}^{M}\mathcal{Q}\left(\frac{1}{n}\sum_{i=0}^{n-1}\nabla f_m^{\pi_m^i}(x_{t,m}^i)\right), x_t - x_\star\right\rangle\right] \\
&\quad + \frac{2\eta^2 L_{\max}^2\left(1 + \frac{\omega}{M}\right)}{Mn}\sum_{m=1}^{M}\sum_{i=0}^{n-1}\left\|x_{t,m}^i - x_t\right\|^2 \\
&\quad + 8\eta^2 L_{\max}\left(1 + \frac{\omega}{M}\right)\left(f(x_t) - f(x_\star)\right) + 4\eta^2\frac{\omega}{M}\zeta_\star^2 \\
&\leq \|x_t - x_\star\|^2 - 2\eta\frac{1}{Mn}\sum_{m=1}^{M}\sum_{i=0}^{n-1}\left\langle \nabla f_m^{\pi_m^i}(x_{t,m}^i), x_t - x_\star\right\rangle \\
&\quad + \frac{2\eta^2 L_{\max}^2\left(1 + \frac{\omega}{M}\right)}{Mn}\sum_{m=1}^{M}\sum_{i=0}^{n-1}\left\|x_{t,m}^i - x_t\right\|^2 \\
&\quad + 8\eta^2 L_{\max}\left(1 + \frac{\omega}{M}\right)\left(f(x_t) - f(x_\star)\right) + 4\eta^2\frac{\omega}{M}\zeta_\star^2.
\end{aligned}
$$

Next, Lemma D.2 implies

$$
\begin{aligned}
\mathbb{E}_{\mathcal{Q}}\left[\|x_{t+1} - x_\star\|^2\right] &\leq \|x_t - x_\star\|^2 - \frac{\eta\mu}{2}\|x_t - x_\star\|^2 - \eta\left(f(x_t) - f(x_\star)\right) \\
&\quad + 8\eta^2 L_{\max}\left(1 + \frac{\omega}{M}\right)\left(f(x_t) - f(x_\star)\right) + \frac{\eta L_{\max}}{Mn}\sum_{m=1}^{M}\sum_{i=0}^{n-1}\left\|x_{t,m}^i - x_t\right\|^2 \\
&\quad + \frac{2\eta^2 L_{\max}^2\left(1 + \frac{\omega}{M}\right)}{Mn}\sum_{m=1}^{M}\sum_{i=0}^{n-1}\left\|x_{t,m}^i - x_t\right\|^2 + 4\eta^2\frac{\omega}{M}\zeta_\star^2 \\
&\leq \left(1 - \frac{\eta\mu}{2}\right)\|x_t - x_\star\|^2 - \eta\left(1 - 8\eta L_{\max}\left(1 + \frac{\omega}{M}\right)\right)\left(f(x_t) - f(x_\star)\right) \\
&\quad + \frac{\eta L_{\max}\left(1 + 2\eta L_{\max}\left(1 + \frac{\omega}{M}\right)\right)}{Mn}\sum_{m=1}^{M}\sum_{i=0}^{n-1}\left\|x_{t,m}^i - x_t\right\|^2 + 4\eta^2\frac{\omega}{M}\zeta_\star^2.
\end{aligned}
$$

Using Lemma D.3, we get

$$
\begin{aligned}
\mathbb{E}_{\mathcal{Q}}\left[\|x_{t+1} - x_\star\|^2\right] \leq\ & \left(1 - \frac{\eta\mu}{2}\right)\|x_t - x_\star\|^2 - \eta\left(1 - 8\eta L\left(1 + \frac{\omega}{M}\right)\right)(f(x_t) - f(x_\star)) \\
& + \eta L_{\max}\left(1 + 2\eta L_{\max}\left(1 + \frac{\omega}{M}\right)\right) \cdot 8\gamma^2 n^2 L_{\max}(f(x_t) - f(x_\star)) \\
& + \eta L_{\max}\left(1 + 2\eta L_{\max}\left(1 + \frac{\omega}{M}\right)\right) \cdot 2\gamma^2 n\left(\sigma_\star^2 + (n+1)\zeta_\star^2\right) \\
& + 4\eta^2\frac{\omega}{M}\zeta_\star^2.
\end{aligned}
$$

In view of (24), we have

$$
\begin{aligned}
\mathbb{E}_{\mathcal{Q}}\left[\|x_{t+1} - x_\star\|^2\right] \leq\ & \left(1 - \frac{\eta\mu}{2}\right)\|x_t - x_\star\|^2 + 4\eta^2\frac{\omega}{M}\zeta_\star^2 \\
& - \eta\left(1 - 8\eta L_{\max}\left(1 + \frac{\omega}{M}\right) - 8\gamma^2 n^2 L_{\max}^2\left(1 + 2L_{\max}\eta\left(1 + \frac{\omega}{M}\right)\right)\right)(f(x_t) - f(x_\star)) \\
& + 2\gamma^2 n\eta L_{\max}\left(1 + 2\eta L\left(1 + \frac{\omega}{M}\right)\right)\left(\sigma_\star^2 + n\zeta_\star^2\right) \\
\leq\ & \left(1 - \frac{\eta\mu}{2}\right)\|x_t - x_\star\|^2 + 4\eta^2\frac{\omega}{M}\zeta_\star^2 + \frac{9}{4}\eta L_{\max}\gamma^2 n\left(\sigma_\star^2 + (n+1)\sigma_\star^2\right).
\end{aligned}
$$

Recursively unrolling the inequality and using $\sum_{t=0}^{+\infty}\left(1 - \frac{\eta\mu}{2}\right)^t \leq \frac{2}{\mu\eta}$, we get the result. $\qquad\square$

**Corollary 9.** *Let the assumptions of Theorem 2.3 hold, $\gamma = \eta/n$, and*

$$
\eta = \min\left\{\frac{1}{16L_{\max}\left(1 + \frac{\omega}{M}\right)}, \sqrt{\frac{\varepsilon\mu n}{9L_{\max}}}\left((n+1)\zeta_\star^2 + \sigma_\star^2\right)^{-1/2}, \frac{\varepsilon\mu M}{24\omega\zeta_\star^2}\right\}. \tag{25}
$$

*Then, Q-NASTYA finds a solution with accuracy $\varepsilon > 0$ after the following number of communication rounds:*

$$
\widetilde{\mathcal{O}}\left(\frac{L_{\max}}{\mu}\left(1 + \frac{\omega}{M}\right) + \frac{\omega}{M}\frac{\zeta_\star^2}{\varepsilon\mu^3} + \sqrt{\frac{L_{\max}}{\varepsilon\mu^3}}\sqrt{\zeta_\star^2 + \sigma_\star^2/n}\right).
$$

*If $\gamma \to 0$, one can choose $\eta = \min\left\{\frac{1}{16L_{\max}\left(1 + \frac{\omega}{M}\right)}, \frac{\varepsilon\mu M}{24\omega\zeta_\star^2}\right\}$ such that the above complexity bound improves to*

$$
\widetilde{\mathcal{O}}\left(\frac{L_{\max}}{\mu}\left(1 + \frac{\omega}{M}\right) + \frac{\omega}{M}\frac{\zeta_\star^2}{\varepsilon\mu^3}\right).
$$

*Proof.* Theorem 2.3 implies

$$
\mathbb{E}\left[\|x_T - x_\star\|^2\right] \leq \left(1 - \frac{\eta\mu}{2}\right)^T\|x_0 - x_\star\|^2 + \frac{9}{2}\frac{\gamma^2 n L_{\max}}{\mu}\left((n+1)\zeta_\star^2 + \sigma_\star^2\right) + 8\frac{\eta\omega}{\mu M}\zeta_\star^2.
$$

To estimate the number of communication rounds required to find a solution with accuracy $\varepsilon > 0$, we need to upper bound each term from the right-hand side by $\varepsilon/3$. Thus, we get additional conditions on $\eta$:

$$
\frac{9}{2}\frac{\eta^2 L_{\max}}{n\mu}\left((n+1)\zeta_\star^2 + \sigma_\star^2\right) < \frac{\varepsilon}{3}, \quad 8\frac{\eta\omega}{\mu M}\zeta_\star^2 < \frac{\varepsilon}{3}
$$

and also the upper bound on the number of communication rounds $T$

$$
T = \widetilde{\mathcal{O}}\left(\frac{1}{\eta\mu}\right).
$$

Substituting (28) in the previous equation, we get the first part of the result. When $\gamma \to 0$, the proof follows similar steps. $\qquad\square$

## E   MISSING PROOFS FOR DIANA-NASTYA

**Lemma E.1.** *Under Assumptions 1, 2, 3, the iterates produced by* DIANA-NASTYA *satisfy*

$$
-\mathbb{E}_{\mathcal{Q}}\left[\frac{1}{M}\sum_{m=1}^{M}\langle \hat{g}_{t,m}-h^{\star}, x_t - x_{\star}\rangle\right] \leq -\frac{\mu}{4}\|x_t - x_{\star}\|^2 - \frac{1}{2}\left(f(x_t) - f(x_{\star})\right)
$$

$$
-\frac{1}{Mn}\sum_{m=1}^{M}\sum_{i=0}^{n-1}D_{f_m^{\pi_m^i}}(x_{\star}, x_{t,m}^i)
$$

$$
+\frac{L_{\max}}{2Mn}\sum_{m=1}^{M}\sum_{i=0}^{n-1}\|x_t - x_{t,m}^i\|^2,
$$

*where $h^{\star} = \nabla f(x_{\star})$.*

*Proof.* Using that $\mathbb{E}_{\mathcal{Q}}[\hat{g}_{t,m}] = g_{t,m}$ and definition of $h^{\star}$, we get

$$
-\mathbb{E}_{\mathcal{Q}}\left[\frac{1}{M}\sum_{m=1}^{M}\langle \hat{g}_{t,m}-h^{\star}, x_t - x_{\star}\rangle\right] = -\frac{1}{M}\sum_{m=1}^{M}\langle g_{t,m}-h^{\star}, x_t - x_{\star}\rangle
$$

$$
= -\frac{1}{Mn}\sum_{m=1}^{M}\sum_{i=0}^{n-1}\left\langle \nabla f_m^{\pi_m^i}(x_{t,m}^i) - \nabla f_m^{\pi_m^i}(x_{\star}), x_t - x_{\star}\right\rangle.
$$

Next, three-point identity and $L_{\max}$-smoothness of each function $f_m^i$ imply

$$
-\mathbb{E}_{\mathcal{Q}}\left[\frac{1}{M}\sum_{m=1}^{M}\langle \hat{g}_{t,m}-h^{\star}, x_t - x_{\star}\rangle\right] = -\frac{1}{Mn}\sum_{m=1}^{M}\sum_{i=0}^{n-1}\left(D_{f_m^{\pi_m^i}}(x_t, x_{\star}) + D_{f_m^{\pi_m^i}}(x_{\star}, x_{t,m}^i) - D_{f_m^{\pi_m^i}}(x_t, x_{t,m}^i)\right)
$$

$$
\leq -D_f(x_t, x_{\star}) - \frac{1}{Mn}\sum_{m=1}^{M}\sum_{i=0}^{n-1}D_{f_m^{\pi_m^i}}(x_{\star}, x_{t,m}^i)
$$

$$
+\frac{L_{\max}}{2Mn}\sum_{m=1}^{M}\sum_{i=0}^{n-1}\|x_t - x_{t,m}^i\|^2
$$

Finally, using $\mu$-strong convexity of $f$, we finish the proof of lemma.   □

**Lemma E.2.** *Under Assumptions 1, 2, 3, the iterates produced by* DIANA-NASTYA *satisfy*

$$
\mathbb{E}_{\mathcal{Q}}\left[\|\hat{g}_t - h^{\star}\|^2\right] \leq \frac{2L_{\max}^2\left(1+\frac{\omega}{M}\right)}{Mn}\sum_{m=1}^{M}\sum_{i=0}^{n-1}\|x_{t,m}^i - x_t\|^2 + 8L_{\max}\left(1+\frac{\omega}{M}\right)\left(f(x_t)-f(x_{\star})\right)
$$

$$
+\frac{4\omega}{M^2}\sum_{m=1}^{M}\|h_{t,m}-h_m^{\star}\|^2.
$$

*Proof.* Since $g_t = \frac{1}{M}\sum_{m=1}^{M}g_{t,m}$ and $\mathbb{E}\|\xi - c\|^2 = \mathbb{E}\|\xi - \mathbb{E}\xi\|^2 + \mathbb{E}\|\mathbb{E}\xi - c\|^2$, we have

$$
\mathbb{E}_{\mathcal{Q}}\left[\|\hat{g}_t - h^{\star}\|^2\right] = \mathbb{E}_{\mathcal{Q}}\left[\left\|\frac{1}{M}\sum_{m=1}^{M}\left(h_{t,m}+\mathcal{Q}\left(g_{t,m}-h_{t,m}\right)-h_m^{\star}\right)\right\|^2\right]
$$

$$
= \mathbb{E}_{\mathcal{Q}}\left[\left\|\frac{1}{M}\sum_{m=1}^{M}\left(h_{t,m}+\mathcal{Q}\left(g_{t,m}-h_{t,m}\right)\right)-g_t\right\|^2\right] + \|g_t - h^{\star}\|^2.
$$

Next, independence of $\mathcal{Q}\left(g_{t,m} - h_{t,m}\right)$, $m \in M$, Assumption 1, and $L_{\max}$-smoothness and convexity of each function $f_m^i$ imply

$$
\begin{aligned}
\mathbb{E}_{\mathcal{Q}}\left[\|\hat{g}_t - h^\star\|^2\right] &\leq \frac{\omega}{M^2}\sum_{m=1}^{M}\|g_{t,m} - h_{t,m}\|^2 + \|g_t - h^\star\|^2 \\
&\leq \frac{2\omega}{M^2}\sum_{m=1}^{M}\left\|\frac{1}{n}\sum_{i=0}^{n-1}\nabla f_m^{\pi_m^i}(x_{t,m}^i) - \nabla f_m(x_t)\right\|^2 + \frac{2\omega}{M^2}\sum_{m=1}^{M}\|\nabla f_m(x_t) - h_{t,m}\|^2 \\
&\quad + 2\|g_t - \nabla f(x_t)\|^2 + 2\|\nabla f(x_t) - h^\star\|^2 \\
&\leq \frac{2\omega}{M^2}\sum_{m=1}^{M}\left\|\frac{1}{n}\sum_{i=0}^{n-1}\nabla f_m^{\pi_m^i}(x_{t,m}^i) - \nabla f_m(x_t)\right\|^2 + \frac{2\omega}{M^2}\sum_{m=1}^{M}\|\nabla f_m(x_t) - h_{t,m}\|^2 \\
&\quad + 2\left\|\frac{1}{M}\sum_{m=1}^{M}\left(\frac{1}{n}\sum_{i=0}^{n-1}\nabla f_m^{\pi_m^i}(x_{t,m}^i) - \nabla f_m(x_t)\right)\right\|^2 + 2\|\nabla f(x_t) - h^\star\|^2 \\
&\leq \frac{2L_{\max}^2\left(1 + \frac{\omega}{M}\right)}{Mn}\sum_{m=1}^{M}\sum_{i=0}^{n-1}\|x_{t,m}^i - x_t\|^2 + \frac{2\omega}{M^2}\sum_{m=1}^{M}\|\nabla f_m(x_t) - h_{t,m}\|^2 \\
&\quad + 2\|\nabla f(x_t) - h^\star\|^2.
\end{aligned}
$$

Using $L_{\max}$-smoothness and convexity of $f_m$, we get

$$
\begin{aligned}
\mathbb{E}_{\mathcal{Q}}\left[\|\hat{g}_t - h^\star\|^2\right] &\leq \frac{2L_{\max}^2\left(1 + \frac{\omega}{M}\right)}{Mn}\sum_{m=1}^{M}\sum_{i=0}^{n-1}\|x_{t,m}^i - x_t\|^2 + \frac{2\omega}{M^2}\sum_{m=1}^{M}\|\nabla f_m(x_t) - h_{t,m}\|^2 \\
&\quad + 4L_{\max}\left(f(x_t) - f(x_\star)\right) \\
&\leq \frac{2L_{\max}^2\left(1 + \frac{\omega}{M}\right)}{Mn}\sum_{m=1}^{M}\sum_{i=0}^{n-1}\|x_{t,m}^i - x_t\|^2 + \frac{4\omega}{M^2}\sum_{m=1}^{M}\|\nabla f_m(x_t) - h_m^\star\|^2 \\
&\quad + \frac{4\omega}{M^2}\sum_{m=1}^{M}\|h_{t,m} - h_m^\star\|^2 + 4L_{\max}\left(f(x_t) - f(x_\star)\right) \\
&\leq \frac{2L_{\max}^2\left(1 + \frac{\omega}{M}\right)}{Mn}\sum_{m=1}^{M}\sum_{i=0}^{n-1}\|x_{t,m}^i - x_t\|^2 + \frac{8L_{\max}\omega}{M^2}\sum_{m=1}^{M}D_{f_m}(x_t, x_\star) \\
&\quad + \frac{4\omega}{M^2}\sum_{m=1}^{M}\|h_{t,m} - h_m^\star\|^2 + 4L_{\max}\left(f(x_t) - f(x_\star)\right).
\end{aligned}
$$

$\square$

**Lemma E.3.** *Under Assumptions 1, 2, 3, and $\alpha \leq \frac{1}{1+\omega}$, the iterates produced by DIANA-NASTYA satisfy*

$$
\begin{aligned}
\frac{1}{M}\sum_{m=1}^{M}\mathbb{E}_{\mathcal{Q}}\left[\|h_{t+1,m} - h_m^\star\|^2\right] &\leq \frac{1-\alpha}{M}\sum_{m=1}^{M}\|h_{t,m} - h_m^\star\|^2 + \frac{2\alpha L_{\max}^2}{Mn}\sum_{m=1}^{M}\sum_{i=0}^{n-1}\|x_{t,m}^i - x_t\|^2 \\
&\quad + 4\alpha L_{\max}\left(f(x_t) - f(x_\star)\right).
\end{aligned}
$$

*Proof.* Taking expectation w.r.t. $\mathcal{Q}$ and using Assumption 1, we obtain

$$
\begin{aligned}
\frac{1}{M}\sum_{m=1}^{M}\mathbb{E}_{\mathcal{Q}}\left[\|h_{t+1,m}-h_m^\star\|^2\right] &= \frac{1}{M}\sum_{m=1}^{M}\mathbb{E}_{\mathcal{Q}}\left[\|h_{t,m}+\alpha\mathcal{Q}(g_{t,m}-h_{t,m})-h_m^\star\|^2\right] \\
&\leq \frac{1}{M}\sum_{m=1}^{M}\left(\|h_{t,m}-h_m^\star\|^2+2\alpha\mathbb{E}_{\mathcal{Q}}\left[\langle\mathcal{Q}(g_{t,m}-h_{t,m}),h_{t,m}-h_m^\star\rangle\right]\right) \\
&\quad +\alpha^2\frac{1}{M}\sum_{m=1}^{M}\mathbb{E}_{\mathcal{Q}}\left[\|\mathcal{Q}(g_{t,m}-h_{t,m})\|^2\right] \\
&\leq \frac{1}{M}\sum_{m=1}^{M}\left(\|h_{t,m}-h_m^\star\|^2+2\alpha\langle g_{t,m}-h_{t,m},h_{t,m}-h_m^\star\rangle\right) \\
&\quad +\alpha^2(1+\omega)\frac{1}{M}\sum_{m=1}^{M}\|g_{t,m}-h_{t,m}\|^2
\end{aligned}
$$

Using $\alpha\leq\frac{1}{1+\omega}$, we get

$$
\begin{aligned}
\frac{1}{M}\sum_{m=1}^{M}\mathbb{E}_{\mathcal{Q}}\left[\|h_{t+1,m}-h_m^\star\|^2\right] &\leq \frac{1}{M}\sum_{m=1}^{M}\left(\|h_{t,m}-h_m^\star\|^2+\alpha\langle g_{t,m}-h_{t,m},h_{t,m}+g_{t,m}-2h_m^\star\rangle\right) \\
&\leq \frac{1}{M}\sum_{m=1}^{M}\left(\|h_{t,m}-h_m^\star\|^2+\alpha\|g_{t,m}-h_m^\star\|^2-\alpha\|h_{t,m}-h_m^\star\|^2\right) \\
&\leq \frac{1-\alpha}{M}\sum_{m=1}^{M}\|h_{t,m}-h_m^\star\|^2+\frac{\alpha}{M}\sum_{m=1}^{M}\|g_{t,m}-h_m^\star\|^2.
\end{aligned}
$$

Finally, $L_{\max}$-smoothness and convexity of $f_m$ imply

$$
\begin{aligned}
\frac{1}{M}\sum_{m=1}^{M}\mathbb{E}_{\mathcal{Q}}\left[\|h_{t+1,m}-h_m^\star\|^2\right] &\leq \frac{1-\alpha}{M}\sum_{m=1}^{M}\|h_{t,m}-h_m^\star\|^2 \\
&\quad +\frac{2\alpha}{M}\sum_{m=1}^{M}\left(\|g_{t,m}-\nabla f_m(x_t)\|^2+\|\nabla f_m(x_t)-h_m^\star\|^2\right) \\
&\leq \frac{1-\alpha}{M}\sum_{m=1}^{M}\|h_{t,m}-h_m^\star\|^2+\frac{4L_{\max}\alpha}{M}\sum_{m=1}^{M}D_{f_m}(x_t,x_\star) \\
&\quad +\frac{2\alpha}{M}\sum_{m=1}^{M}\left\|\frac{1}{n}\sum_{i=0}^{n-1}(\nabla f_m^{\pi_m^i}(x_{t,m}^i)-\nabla f_m^i(x_t))\right\|^2 \\
&\leq \frac{1-\alpha}{M}\sum_{m=1}^{M}\|h_{t,m}-h_m^\star\|^2+4L_{\max}\alpha\left(f(x_t)-f(x_\star)\right) \\
&\quad +\frac{2L_{\max}^2\alpha}{Mn}\sum_{m=1}^{M}\sum_{i=0}^{n-1}\left\|x_{t,m}^i-x_t\right\|^2.
\end{aligned}
$$

$\square$

**Theorem E.1.** *Let Assumptions 1, 2, 3 hold and stepsizes $\gamma$, $\eta$, $\alpha$ satisfy*

$$
0<\gamma\leq\frac{1}{16L_{\max}n},\quad 0<\eta\leq\min\left\{\frac{\alpha}{2\mu},\frac{1}{16L_{\max}\left(1+\frac{9\omega}{M}\right)}\right\},\quad \alpha\leq\frac{1}{1+\omega}. \tag{26}
$$

*Then, for all $T\geq 0$ the iterates produced by* DIANA-NASTYA *satisfy*

$$
\mathbb{E}\left[\Psi_T\right]\leq\left(1-\frac{\eta\mu}{2}\right)^T\Psi_0+\frac{9}{2}\frac{\gamma^2nL}{\mu}\left(\sigma_\star^2+(n+1)\zeta_\star^2\right). \tag{27}
$$

*Proof.* We have

$$
\begin{aligned}
\|x_{t+1} - x_\star\|^2 &= \|x_t - \eta \hat{g}_t - x_\star + \eta h^\star\|^2 \\
&= \|x_t - x_\star\|^2 - 2\eta \langle \hat{g}_t - h^\star, x_t - x_\star \rangle + \eta^2 \|\hat{g}_t - h^\star\|^2.
\end{aligned}
$$

Taking expectation w.r.t. $\mathcal{Q}$ and using Lemma E.1, we obtain

$$
\begin{aligned}
\mathbb{E}_\mathcal{Q}\left[\|x_{t+1} - x_\star\|^2\right] &= \|x_t - x_\star\|^2 - 2\eta \mathbb{E}_\mathcal{Q}\left[\langle \hat{g}_t - h^\star, x_t - x_\star \rangle\right] + \eta^2 \mathbb{E}_\mathcal{Q}\left[\|\hat{g}_t - h^\star\|^2\right] \\
&\leq \left(1 - \frac{\eta\mu}{2}\right)\|x_t - x_\star\|^2 - \eta(f(x_t) - f(x_\star)) - \frac{2\eta}{Mn}\sum_{m=1}^{M}\sum_{i=0}^{n-1} D_{f_m^{\pi_m^i}}(x_\star, x_{t,m}^i) \\
&\quad + \frac{L_{\max}\eta}{Mn}\sum_{m=1}^{M}\sum_{i=0}^{n-1}\|x_{t,m}^i - x_t\|^2 + \eta^2 \mathbb{E}_\mathcal{Q}\left[\|\hat{g}_t - h^\star\|^2\right].
\end{aligned}
$$

Next, Lemma E.2 implies

$$
\begin{aligned}
\mathbb{E}_\mathcal{Q}\left[\|x_{t+1} - x_\star\|^2\right] &\leq \left(1 - \frac{\eta\mu}{2}\right)\|x_t - x_\star\|^2 - \eta(f(x_t) - f(x_\star)) - \frac{2\eta}{Mn}\sum_{m=1}^{M}\sum_{i=0}^{n-1} D_{f_m^{\pi_m^i}}(x_\star, x_{t,m}^i) \\
&\quad + \frac{L_{\max}\eta}{Mn}\sum_{m=1}^{M}\sum_{i=0}^{n-1}\|x_{t,m}^i - x_t\|^2 + \frac{2\eta^2 L_{\max}^2\left(1 + \frac{\omega}{M}\right)}{Mn}\sum_{m=1}^{M}\sum_{i=0}^{n-1}\|x_{t,m}^i - x_t\|^2 \\
&\quad + \eta^2\left(8L_{\max}\left(1 + \frac{\omega}{M}\right)(f(x_t) - f(x_\star)) + \frac{4\omega}{M^2}\sum_{m=1}^{M}\|h_{t,m} - h_m^\star\|^2\right) \\
&\leq \left(1 - \frac{\eta\mu}{2}\right)\|x_t - x_\star\|^2 - \eta\left(1 - 8\eta L_{\max}\left(1 + \frac{\omega}{M}\right)\right)(f(x_t) - f(x_\star)) \\
&\quad + L_{\max}\eta\left(1 + 2\eta L_{\max}\left(1 + \frac{\omega}{M}\right)\right)\frac{1}{Mn}\sum_{m=1}^{M}\sum_{i=0}^{n-1}\|x_{t,m}^i - x_t\|^2 \\
&\quad - \frac{2\eta}{Mn}\sum_{m=1}^{M}\sum_{i=0}^{n-1} D_{f_m^{\pi_m^i}}(x_\star, x_{t,m}^i) + \frac{4\eta^2\omega}{M^2}\sum_{m=1}^{M}\|h_{t,m} - h_m^\star\|^2.
\end{aligned}
$$

Using (6) and Lemma E.3, we get

$$
\begin{aligned}
\mathbb{E}_\mathcal{Q}\left[\Psi_{t+1}\right] &\leq \left(1 - \frac{\eta\mu}{2}\right)\|x_t - x_\star\|^2 - \eta\left(1 - 8\eta L_{\max}\left(1 + \frac{\omega}{M}\right)\right)(f(x_t) - f(x_\star)) \\
&\quad + L_{\max}\eta\left(1 + 2\eta L_{\max}\left(1 + \frac{\omega}{M}\right)\right)\frac{1}{Mn}\sum_{m=1}^{M}\sum_{i=0}^{n-1}\|x_{t,m}^i - x_t\|^2 \\
&\quad - \frac{2\eta}{Mn}\sum_{m=1}^{M}\sum_{i=0}^{n-1} D_{f_m^{\pi_m^i}}(x_\star, x_{t,m}^i) + \frac{4\eta^2\omega}{M^2}\sum_{m=1}^{M}\|h_{t,m} - h_m^\star\|^2 \\
&\quad + c\eta^2\left(\frac{1-\alpha}{M}\sum_{m=1}^{M}\|h_{t,m} - h_m^\star\|^2 + \frac{2\alpha L_{\max}^2}{Mn}\sum_{m=1}^{M}\sum_{i=0}^{n-1}\|x_{t,m}^i - x_t\|^2 + 4\alpha L_{\max}(f(x_t) - f(x_\star))\right) \\
&\leq \left(1 - \frac{\eta\mu}{2}\right)\|x_t - x_\star\|^2 + \eta^2\left(c(1-\alpha) + \frac{4\omega}{M}\right)\frac{1}{M}\sum_{m=1}^{M}\|h_{t,m} - h_m^\star\|^2 \\
&\quad - \eta\left(1 - 8\eta L_{\max}\left(1 + \frac{\omega}{M}\right) - 4\alpha\eta c L_{\max}\right)(f(x_t) - f(x_\star)) \\
&\quad + L\eta\left(1 + 2\eta L_{\max}\left(1 + \frac{\omega}{M}\right) + 2\alpha\eta c L_{\max}\right)\frac{1}{Mn}\sum_{m=1}^{M}\sum_{i=0}^{n-1}\|x_{t,m}^i - x_t\|^2.
\end{aligned}
$$

Taking the full expectation, we derive

$$
\begin{aligned}
\mathbb{E}\left[\Psi_{t+1}\right] \leq\ & \left(1 - \frac{\eta\mu}{2}\right) \mathbb{E}\left[\|x_t - x_\star\|^2\right] + \eta^2\left(c(1-\alpha) + \frac{4\omega}{M}\right)\frac{1}{M}\sum_{m=1}^{M}\mathbb{E}\left[\|h_{t,m} - h_m^\star\|^2\right] \\
& -\eta\left(1 - 8\eta L_{\max}\left(1 + \frac{\omega}{M}\right) - 4\alpha\eta c L\right)\mathbb{E}\left[f(x_t) - f(x_\star)\right] \\
& +L_{\max}\eta\left(1 + 2\eta L_{\max}\left(1 + \frac{\omega}{M}\right) + 2\alpha\eta c L_{\max}\right)\frac{1}{Mn}\sum_{m=1}^{M}\sum_{i=0}^{n-1}\mathbb{E}\left[\|x_{t,m}^i - x_t\|^2\right].
\end{aligned}
$$

Using Lemma D.3, we get

$$
\begin{aligned}
\mathbb{E}\left[\Psi_{t+1}\right] \leq\ & \left(1 - \frac{\eta\mu}{2}\right)\mathbb{E}\left[\|x_t - x_\star\|^2\right] + \eta^2\left(c(1-\alpha) + \frac{4\omega}{M}\right)\frac{1}{M}\sum_{m=1}^{M}\mathbb{E}\left[\|h_{t,m} - h_m^\star\|^2\right] \\
& -\eta\left(1 - 8\eta L_{\max}\left(1 + \frac{\omega}{M}\right) - 4\alpha\eta c L_{\max}\right)\mathbb{E}\left[f(x_t) - f(x_\star)\right] \\
& +8\gamma^2 n^2 L_{\max}^2\eta\left(1 + 2\eta L_{\max}\left(1 + \frac{\omega}{M}\right) + 2\alpha\eta c L_{\max}\right)\mathbb{E}\left[f(x_t) - f(x_\star)\right] \\
& +2\gamma^2 n L_{\max}\eta\left(1 + 2\eta L_{\max}\left(1 + \frac{\omega}{M}\right) + 2\alpha\eta c L_{\max}\right)\left(\sigma_\star^2 + (n+1)\zeta_\star^2\right).
\end{aligned}
$$

In view of (26), we have

$$
\begin{aligned}
\mathbb{E}\left[\Psi_{t+1}\right] \leq\ & \left(1 - \frac{\eta\mu}{2}\right)\mathbb{E}\left[\|x_t - x_\star\|^2\right] + \left(1 - \frac{\alpha}{2}\right)\frac{c\eta^2}{M}\sum_{m=1}^{M}\mathbb{E}\left[\|h_{t,m} - h_m^\star\|^2\right] \\
& +\frac{9}{4}\gamma^2 n L_{\max}\eta\left(\sigma_\star^2 + (n+1)\zeta_\star^2\right)
\end{aligned}
$$

Using definition of Lyapunov function and using $\sum_{t=0}^{+\infty}\left(1 - \frac{\eta\mu}{2}\right)^t \leq \frac{2}{\mu\eta}$, we get the result. $\qquad\square$

**Corollary 10.** *Let the assumptions of Theorem 2.4 hold, $\gamma = \eta/n$, $\alpha = \frac{1}{1+\omega}$, and*

$$
\eta = \min\left\{\frac{\alpha}{2\mu}, \frac{1}{16 L_{\max}\left(1 + \frac{9\omega}{M}\right)}, \sqrt{\frac{\varepsilon\mu n}{9 L_{\max}}}\left((n+1)\zeta_\star^2 + \sigma_\star^2\right)^{-1/2}\right\}. \tag{28}
$$

*Then,* DIANA-NASTYA *finds a solution with accuracy $\varepsilon > 0$ after the following number of communication rounds:*

$$
\widetilde{\mathcal{O}}\left(\omega + \frac{L_{\max}}{\mu}\left(1 + \frac{\omega}{M}\right) + \sqrt{\frac{L_{\max}}{\varepsilon\mu^3}}\sqrt{\zeta_\star^2 + \sigma_\star^2/n}\right).
$$

*If $\gamma \to 0$, one can choose $\eta = \min\left\{\frac{\alpha}{2\mu}, \frac{1}{16 L_{\max}\left(1 + \frac{9\omega}{M}\right)}\right\}$ such that the number of communication rounds $T$ to find solution with accuracy $\varepsilon > 0$ is*

$$
\widetilde{\mathcal{O}}\left(\omega + \frac{L_{\max}}{\mu}\left(1 + \frac{\omega}{M}\right)\right).
$$

*Proof.* Theorem 2.4 implies

$$
\mathbb{E}\left[\Psi_T\right] \leq \left(1 - \frac{\eta\mu}{2}\right)^T \Psi_0 + \frac{9}{2}\frac{\gamma^2 n L_{\max}}{\mu}\left((n+1)\zeta_\star^2 + \sigma_\star^2\right).
$$

To estimate the number of communication rounds required to find a solution with accuracy $\varepsilon > 0$, we need to upper bound each term from the right-hand side by $\frac{\varepsilon}{2}$. Thus, we get an additional restriction on $\eta$:

$$
\frac{9}{2}\frac{\eta^2 L_{\max}}{n\mu}\left((n+1)\zeta_\star^2 + \sigma_\star^2\right) < \frac{\varepsilon}{2},
$$

and also the upper bound on the number of communication rounds $T$

$$
T = \widetilde{\mathcal{O}}\left(\frac{1}{\eta\mu}\right).
$$

Substituting (28) in the previous equation, we get the first part of the result. When $\gamma \to 0$, the proof follows similar steps. $\qquad\square$

# F    ALTERNATIVE ANALYSIS OF Q-NASTYA

In this analysis, we will use additional sequence:

$$x_{\star,m}^i = x_\star - \gamma \sum_{j=0}^{i-1} \nabla f_m(x_\star). \tag{29}$$

**Theorem F.1.** *Let Assumptions 1, 3, 4 hold. Moreover, we assume that* $(1-\gamma\mu)^n \leq \frac{9/10 - 1/C}{1+1/C} = \widehat{C} < 1$ *for some numerical constant* $C > 1$. *Also let* $\beta = \frac{\eta}{\gamma n} \leq \frac{1}{3C\frac{\omega}{M}+1}$ *and* $\gamma \leq \frac{1}{L_{\max}}$. *Then, for all* $T \geq 0$ *the iterates produced by* Q-NASTYA *satisfy*

$$\leq \max\left(1 - \frac{\beta}{10}, 1 - \frac{\alpha}{2}\right)\Psi_t + \frac{2}{\mu}\beta\gamma^2\hat{\sigma}_{rad}^2 \tag{30}$$

$$\mathbb{E}\left[\|x_T - x_\star\|^2\right] \leq \left(1 - \frac{\beta}{10}\right)\|x_t - x_*\|^2 + \frac{4}{\mu}\beta\gamma^2\hat{\sigma}_{rad}^2 + 3\beta^2\frac{\omega}{M}\frac{1}{M}\hat{\Delta}_\star,$$

*where* $\hat{\Delta}_\star = \frac{1}{M}\sum_{m=1}^M \|x_{\star,m}^n - x_\star\|^2$ *and* $\hat{\sigma}_{rad}^2 \leq L_{\max}\left(\zeta_\star^2 + n\sigma_\star^2/4\right)$.

*Proof.* The update rule for one epoch can be rewritten as

$$x_{t+1} = x_t - \eta\frac{1}{M}\sum_{M}^{m=1} Q\left(\frac{x_t - x_{t,m}^n}{\gamma n}\right).$$

Using this, we derive

$$\|x_{t+1} - x_*\|^2 = \left\|x_t - \eta\frac{1}{M}\sum_{m=1}^M Q\left(\frac{x_t - x_{t,m}^n}{\gamma n}\right) - x_*\right\|^2$$

$$= \|x_t - x_*\|^2 - 2\eta\left\langle x_t - x_*, \frac{1}{M}\sum_{m=1}^M Q\left(\frac{x_t - x_{t,m}^n}{\gamma n}\right)\right\rangle$$

$$+ \eta^2\left\|\frac{1}{M}\sum_{m=1}^M Q\left(\frac{x_t - x_{t,m}^n}{\gamma n}\right)\right\|^2.$$

Taking conditional expectation w.r.t. the randomness comming from compression, we get

$$\mathbb{E}_Q\|x_{t+1} - x_*\|^2 = \|x_t - x_*\|^2 - 2\eta\left\langle x_t - x_*, \frac{1}{M}\sum_{m=1}^M\left(\frac{x_t - x_{t,m}^n}{\gamma n}\right)\right\rangle$$

$$+ \eta^2\mathbb{E}_Q\left\|\frac{1}{M}\sum_{m=1}^M Q\left(\frac{x_t - x_{t,m}^n}{\gamma n}\right)\right\|^2.$$

Next, we use the definition of quantization operator and independence of $Q\left(\frac{x_t - x_{t,m}^n}{\gamma n}\right)$, $m \in [M]$:

$$\mathbb{E}_Q\|x_{t+1} - x_*\|^2 \leq \|x_t - x_*\|^2 - 2\eta\left\langle x_t - x_*, \frac{1}{M}\sum_{m=1}^M\left(\frac{x_t - x_{t,m}^n}{\gamma n}\right)\right\rangle$$

$$+ \eta^2\left(\frac{\omega}{M}\frac{1}{M}\sum_{m=1}^M\left\|\frac{x_t - x_{t,m}^n}{\gamma n}\right\|^2 + \left\|\frac{1}{M}\sum_{m=1}^M\frac{x_t - x_{t,m}^n}{\gamma n}\right\|^2\right).$$

Since $\beta = \frac{\eta}{\gamma n}$, we obtain

$$
\begin{aligned}
\mathbb{E}_Q \|x_{t+1} - x_*\|^2 &\leq \|x_t - x_*\|^2 - 2\beta \left\langle x_t - x_*, \frac{1}{M} \sum_{m=1}^M \left(x_t - x_{t,m}^n\right) \right\rangle \\
&\quad + \beta^2 \frac{\omega}{M} \frac{1}{M} \sum_{m=1}^M \left\|x_t - x_{t,m}^n\right\|^2 + \beta^2 \left\|\frac{1}{M} \sum_{m=1}^M \left(x_t - x_{t,m}^n\right)\right\|^2 \\
&= \|x_t - x_*\|^2 + 2\beta \left\langle x_t - x_*, \frac{1}{M} \sum_{m=1}^M \left(x_{t,m}^n - x_t\right) \right\rangle \\
&\quad + \beta^2 \frac{\omega}{M} \frac{1}{M} \sum_{m=1}^M \left\|x_t - x_{t,m}^n\right\|^2 + \beta^2 \left\|\frac{1}{M} \sum_{m=1}^M \left(x_{t,m}^n - x_t\right)\right\|^2 \\
&= \left\|x_t - x_* + \beta\left(\frac{1}{M} \sum_{m=1}^M \left(x_{t,m}^n - x_t\right)\right)\right\|^2 + \beta^2 \frac{\omega}{M} \frac{1}{M} \sum_{m=1}^M \left\|x_t - x_{t,m}^n\right\|^2 \\
&= \left\|(1-\beta)(x_t - x_*) + \beta\left(\frac{1}{M} \sum_{m=1}^M \left(x_{t,m}^n\right) - x_*\right)\right\|^2 \\
&\quad + \beta^2 \frac{\omega}{M} \frac{1}{M} \sum_{m=1}^M \left\|x_t - x_{t,m}^n\right\|^2.
\end{aligned}
$$

Using the condition that $x_* = \frac{1}{M} \sum_{m=1}^M x_{*,m}^n$ we have:

$$
\mathbb{E}_Q \|x_{t+1} - x_*\|^2 \leq \left\|(1-\beta)(x_t - x_*) + \beta\left(\frac{1}{M} \sum_{m=1}^M \left(x_{t,m}^n - x_{*,m}^n\right)\right)\right\|^2 + \beta^2 \frac{\omega}{M} \frac{1}{M} \sum_{m=1}^M \left\|x_t - x_{t,m}^n\right\|^2.
$$

Convexity of squared norm and Jensen's inequality imply

$$
\mathbb{E}_Q \|x_{t+1} - x_*\|^2 \leq (1-\beta)\|x_t - x_*\|^2 + \beta\left\|\frac{1}{M} \sum_{m=1}^M \left(x_{t,m}^n - x_{*,m}^n\right)\right\|^2 + \beta^2 \frac{\omega}{M} \frac{1}{M} \sum_{m=1}^M \left\|x_t - x_{t,m}^n\right\|^2.
$$

Next, from Young's inequality we get

$$
\begin{aligned}
\mathbb{E}_Q \|x_{t+1} - x_*\|^2 &\leq (1-\beta)\|x_t - x_*\|^2 + \beta\left\|\frac{1}{M} \sum_{m=1}^M \left(x_{t,m}^n - x_{*,m}^n\right)\right\|^2 + 3\beta^2 \frac{\omega}{M}\|x_t - x_*\|^2 \\
&\quad + 3\beta^2 \frac{\omega}{M} \frac{1}{M} \sum_{m=1}^M \|x_{t,m}^n - x_{*,m}^n\|^2 + 3\beta^2 \frac{\omega}{M} \frac{1}{M} \sum_{m=1}^M \|x_{*,m}^n - x_*\|^2.
\end{aligned}
$$

Theorem 4 from (Mishchenko et al., 2021) gives

$$
\begin{aligned}
\mathbb{E}\left[\frac{1}{M} \sum_{m=1}^M \|x_{t,m}^n - x_{*,m}^n\|^2\right] &\leq (1-\gamma\mu)^n \left[\|x_t - x_*\|^2\right] + 2\gamma^3 \hat{\sigma}_{\text{rad}}^2 \left(\sum_{j=0}^{n-1}(1-\gamma\mu)^j\right) \\
&= (1-\gamma\mu)^n \left[\|x_t - x_*\|^2\right] + 2\gamma^2 \hat{\sigma}_{\text{rad}}^2 \frac{1}{\gamma\mu}.
\end{aligned}
$$

It leads to

$$
\begin{aligned}
\mathbb{E}\|x_{t+1} - x_*\|^2 &\le (1-\beta)\|x_t - x_*\|^2 + \beta\left((1-\gamma\mu)^n\left[\|x_t - x_*\|^2\right] + 2\gamma^3\hat{\sigma}_{\text{rad}}^2\frac{1}{\gamma\mu}\right) \\
&\quad + 3\beta^2\frac{\omega}{M}\|x_t - x_*\|^2 + 3\beta^2\frac{\omega}{M}\left((1-\gamma\mu)^n\left[\|x_t - x_*\|^2\right] + 2\gamma^3\hat{\sigma}_{\text{rad}}^2\frac{1}{\gamma\mu}\right) \\
&\quad + 3\beta^2\frac{\omega}{M}\frac{1}{M}\sum_{m=1}^{M}\|x_{*,m}^n - x_*\|^2 \\
&\le \left(1 - \beta + \beta(1-\gamma\mu)^n + 3\beta^2\frac{\omega}{M} + 3\beta^2\frac{\omega}{M}(1-\gamma\mu)^n\right)\|x_t - x_*\|^2 \\
&\quad + 2\beta\gamma^3\hat{\sigma}_{\text{rad}}^2\frac{1}{\gamma\mu}\left(1 + 3\beta\frac{\omega}{M}\right) + 3\beta^2\frac{\omega}{M}\frac{1}{M}\sum_{m=1}^{M}\|x_{*,m}^n - x_*\|^2.
\end{aligned}
$$

Using $(1-\gamma\mu)^n \le \frac{9/10 - 1/C}{1 + 1/C}$, we have

$$
\begin{aligned}
(1-\gamma\mu)^n &\le \frac{9/10 - 1/C}{1 + 1/C} \\
(1-\gamma\mu)^n\left(1 + \frac{1}{C}\right) &\le \frac{9}{10} - \frac{1}{C} \\
-\frac{9}{10}\beta + \beta(1-\gamma\mu)^n + \frac{\beta}{C} + \frac{\beta}{C}(1-\gamma\mu)^n &\le 0 \\
1 - \beta + \beta(1-\gamma\mu)^n + \frac{\beta}{C} + \frac{\beta}{C}(1-\gamma\mu)^n &\le 1 - \frac{\beta}{10}.
\end{aligned}
$$

Next, applying $\beta \le \frac{1}{1 + 3C\frac{\omega}{M}}$, we derive

$$
1 - \beta + \beta(1-\gamma\mu)^n + 3\beta^2\frac{\omega}{M} + 3\beta^2\frac{\omega}{M}(1-\gamma\mu)^n \le 1 - \frac{\beta}{10}.
$$

Finally, we have

$$
\begin{aligned}
\mathbb{E}\|x_{t+1} - x_\star\|^2 &\le \left(1 - \frac{\beta}{10}\right)\|x_t - x_*\|^2 + 2\beta\gamma^2\hat{\sigma}_{\text{rad}}^2\frac{1}{\mu}\left(1 + \frac{1}{C}\right) \\
&\quad + 3\beta^2\frac{\omega}{M}\frac{1}{M}\sum_{m=1}^{M}\|x_{*,m}^n - x_*\|^2 \\
&\le \left(1 - \frac{\beta}{10}\right)\|x_t - x_*\|^2 + \frac{4}{\mu}\beta\gamma^2\hat{\sigma}_{\text{rad}}^2 \\
&\quad + 3\beta^2\frac{\omega}{M}\frac{1}{M}\sum_{m=1}^{M}\|x_{*,m}^n - x_*\|^2.
\end{aligned}
$$

$\square$

# G  ALTERNATIVE ANALYSIS OF DIANA-NASTYA

**Theorem G.1.** *Let Assumptions 1, 3, 4 hold. Moreover, we assume that $(1 - \gamma\mu)^n \leq \frac{9/10 - 1/B}{1 + 1/B} = \widehat{B} < 1$ for some numerical constant $B > 1$. Also let $\beta = \frac{\eta}{\gamma n} \leq \frac{1}{12B\frac{\omega}{M} + 1}$ and $\gamma \leq \frac{1}{L_{\max}}$ and also $\alpha \leq \frac{1}{\omega + 1}$. Then, for all $T \geq 0$ the iterates produced by DIANA-NASTYA satisfy*

$$\mathbb{E}\Psi_T \leq \max\left(1 - \frac{\beta}{10}, 1 - \frac{\alpha}{2}\right)^T \Psi_0 + \frac{2}{\mu\min(\frac{\beta}{10}, \frac{\alpha}{2})}\beta\gamma^2\hat{\sigma}_{rad}^2. \tag{31}$$

*Proof.* We start with expanding the square:

$$\|x_{t+1} - x_*\|^2 = \|x_t - \eta\hat{g}_t - x_*\|^2$$

$$= \left\|x_t - \eta\frac{1}{M}\sum_{m=1}^{M}(h_{t,m} + Q(g_{t,m} - h_{t,m})) - x_*\right\|^2$$

$$= \|x_t - x_*\|^2 - 2\eta\left\langle\frac{1}{M}\sum_{m=1}^{M}(h_{t,m} + Q(g_{t,m} - h_{t,m})), x_t - x_*\right\rangle$$

$$+ \eta^2\left\|\frac{1}{M}\sum_{m=1}^{M}(h_{t,m} + Q(g_{t,m} - h_{t,m}))\right\|^2.$$

Taking the expectation w.r.t. $\mathcal{Q}$, we get

$$\mathbb{E}_Q\|x_{t+1} - x_*\|^2 = \|x_t - x_*\|^2 - 2\eta\left\langle\frac{1}{M}\sum_{m=1}^{M}g_{t,m}, x_t - x_*\right\rangle$$

$$+ \eta^2\mathbb{E}_Q\left\|\frac{1}{M}\sum_{m=1}^{M}(h_{t,m} + Q(g_{t,m} - h_{t,m}))\right\|^2$$

$$= \|x_t - x_*\|^2 - 2\eta\left\langle\frac{1}{M}\sum_{m=1}^{M}g_{t,m}, x_t - x_*\right\rangle$$

$$+ \eta^2\mathbb{E}_Q\left\|\frac{1}{M}\sum_{m=1}^{M}(h_{t,m} + Q(g_{t,m} - h_{t,m}) - g_{t,m})\right\|^2 + \eta^2\left\|\frac{1}{M}\sum_{m=1}^{M}g_{t,m}\right\|^2$$

$$\leq \|x_t - x_*\|^2 - 2\eta\left\langle\frac{1}{M}\sum_{m=1}^{M}g_{t,m}, x_t - x_*\right\rangle$$

$$+ \eta^2\frac{\omega}{M^2}\sum_{m=1}^{M}\|g_{t,m} - h_{t,m}\|^2 + \eta^2\left\|\frac{1}{M}\sum_{m=1}^{M}g_{t,m}\right\|^2$$

$$\leq \|x_t - x_*\|^2 - 2\eta\left\langle\frac{1}{M}\sum_{m=1}^{M}g_{t,m}, x_t - x_*\right\rangle$$

$$+ \eta^2\frac{2\omega}{M^2}\sum_{m=1}^{M}\|g_{t,m} - h_{*,m}\|^2 + \eta^2\frac{2\omega}{M^2}\sum_{m=1}^{M}\|h_{t,m} - h_{*,m}\|^2 + \eta^2\left\|\frac{1}{M}\sum_{m=1}^{M}g_{t,m}\right\|^2.$$

Next, using definition of $g_{t,m}$, we obtain

$$
\mathbb{E}\|x_{t+1} - x_*\|^2 \leq \|x_t - x_*\|^2 - 2\eta \left\langle \frac{1}{M} \sum_{m=1}^{M} \frac{x_t - x_{t,m}^n}{\gamma n}, x_t - x_* \right\rangle + \eta^2 \left\| \frac{1}{M} \sum_{m=1}^{M} \frac{x_t - x_{t,m}^n}{\gamma n} \right\|^2
$$

$$
+ \eta^2 \frac{2\omega}{M^2} \sum_{m=1}^{M} \|g_{t,m} - h_{*,m}\|^2 + \eta^2 \frac{2\omega}{M^2} \sum_{m=1}^{M} \|h_{t,m} - h_{*,m}\|^2
$$

$$
= \|x_t - x_*\|^2 + 2\alpha \left\langle \frac{1}{M} \sum_{m=1}^{M} \left( x_{t,m}^n - x_t \right), x_t - x_* \right\rangle + \alpha^2 \left\| \frac{1}{M} \sum_{m=1}^{M} \left( x_{t,m}^n - x_t \right) \right\|^2
$$

$$
+ \eta^2 \frac{2\omega}{M^2} \sum_{m=1}^{M} \|g_{t,m} - h_{*,m}\|^2 + \eta^2 \frac{2\omega}{M^2} \sum_{m=1}^{M} \|h_{t,m} - h_{*,m}\|^2
$$

$$
= \left\| x_t - x_* + \alpha \frac{1}{M} \sum_{m=1}^{M} \left( x_{t,m}^n - x_t \right) \right\|^2
$$

$$
+ \eta^2 \frac{2\omega}{M^2} \sum_{m=1}^{M} \|g_{t,m} - h_{*,m}\|^2 + \eta^2 \frac{2\omega}{M^2} \sum_{m=1}^{M} \|h_{t,m} - h_{*,m}\|^2
$$

$$
= \left\| (1-\beta)(x_t - x_*) + \beta \left( \frac{1}{M} \sum_{m=1}^{M} \left( x_{t,m}^n - x_{*,m}^n \right) \right) \right\|^2
$$

$$
\leq (1-\beta)\|x_t - x_*\|^2 + \beta \frac{1}{M} \sum_{m=1}^{M} \|x_{t,m}^n - x_{*,m}^n\|^2
$$

$$
+ \eta^2 \frac{2\omega}{M^2} \sum_{m=1}^{M} \|g_{t,m} - h_{*,m}\|^2 + \eta^2 \frac{2\omega}{M^2} \sum_{m=1}^{M} \|h_{t,m} - h_{*,m}\|^2.
$$

Let us consider recursion for control variable:

$$
\|h_{t+1,m} - h_{*,m}\|^2 = \|h_{t,m} + \alpha Q(g_{t,m} - h_{t,m}) - h_{*,m}\|^2
$$
$$
= \|h_{t,m} - h_{*,m}\|^2 + \alpha \left\langle Q(g_{t,m} - h_{t,m}), h_{t,m} - h_{*,m} \right\rangle + \alpha^2 \|Q(g_{t,m} - h_{t,m})\|^2.
$$

Taking the expectation w.r.t. $\mathcal{Q}$, we have

$$
\mathbb{E}_{\mathcal{Q}}\|h_{t+1,m} - h_{*,m}\|^2 \leq \|h_{t,m} - h_{*,m}\|^2 + 2\alpha \left\langle g_{t,m} - h_{t,m}, h_{t,m} - h_{*,m} \right\rangle + \alpha^2 (\omega + 1) \|g_{t,m} - h_{t,m}\|^2.
$$

Using $\alpha \leq \frac{1}{\omega+1}$ we have

$$
\begin{aligned}
\mathbb{E}\|h_{t+1,m} - h_{*,m}\|^2 &\leq \|h_{t,m} - h_{*,m}\|^2 \\
&\quad + 2\alpha \left\langle g_{t,m} - h_{t,m}, h_{t,m} - h_{*,m} \right\rangle + \alpha \|g_{t,m} - h_{t,m}\|^2 \\
&= \|h_{t,m} - h_{*,m}\|^2 \\
&\quad + 2\alpha \left\langle g_{t,m} - h_{t,m}, h_{t,m} - h_{*,m} \right\rangle + \alpha \left\langle g_{t,m} - h_{t,m}, g_{t,m} - h_{t,m} \right\rangle \\
&= \|h_{t,m} - h_{*,m}\|^2 \\
&\quad + \alpha \left\langle g_{t,m} - h_{t,m}, g_{t,m} - h_{t,m} + 2h_{t,m} - 2h_{*,m} \right\rangle \\
&= \|h_{t,m} - h_{*,m}\|^2 \\
&\quad + \alpha \left\langle g_{t,m} - h_{t,m}, g_{t,m} + h_{t,m} - 2h_{*,m} \right\rangle \\
&= \|h_{t,m} - h_{*,m}\|^2 \\
&\quad + \alpha \left\langle g_{t,m} - h_{t,m} - h_{*,m} + h_{*,m}, g_{t,m} + h_{t,m} - 2h_{*,m} \right\rangle \\
&= \|h_{t,m} - h_{*,m}\|^2 \\
&\quad + \alpha \left\langle g_{t,m} - h_{*,m} - (h_{t,m} - h_{*,m}), (g_{t,m} - h_{*,m}) + (h_{t,m} - h_{*,m}) \right\rangle \\
&= \|h_{t,m} - h_{*,m}\|^2 + \alpha \|g_{t,m} - h_{*,m}\|^2 - \alpha \|h_{t,m} - h_{*,m}\|^2 \\
&= (1-\alpha)\|h_{t,m} - h_{*,m}\|^2 + \alpha \|g_{t,m} - h_{*,m}\|^2.
\end{aligned}
$$

Using this bound we get that

$$\frac{1}{M}\sum_{m=1}^{M}\mathbb{E}_{\mathcal{Q}}\|h_{t+1,m} - h_{*,m}\|^2 \le (1-\alpha)\frac{1}{M}\sum_{m=1}^{M}\|h_{t,m} - h_{*,m}\|^2 + \alpha\frac{1}{M}\sum_{m=1}^{M}\|g_{t,m} - h_{*,m}\|^2.$$

Let us consider Lyapunov function:

$$\Psi_t = \|x_t - x_*\|^2 + \frac{4\omega\eta^2}{\alpha M}\frac{1}{M}\sum_{m=1}^{M}\|h_{t,m} - h_{*,m}\|^2.$$

Using previous bounds and Theorem 4 from (Mishchenko et al., 2021) we have

$$\mathbb{E}\Psi_{t+1} \le (1-\beta)\|x_t - x_*\|^2 + \beta\left((1-\gamma\mu)^n\mathbb{E}\|x_t - x_*\|^2 + \gamma^3\frac{1}{\gamma\mu}\hat{\sigma}_{rad}^2\right)$$

$$+ \eta^2\frac{2\omega}{M}\frac{1}{M}\sum_{m=1}^{M}\mathbb{E}\|g_{t,m} - h_{*,m}\|^2 + \eta^2\frac{2\omega}{M}\frac{1}{M}\sum_{m=1}^{M}\mathbb{E}\|h_{t,m} - h_{*,m}\|^2$$

$$+ (1-\alpha)\frac{4\omega\eta^2}{\alpha M}\frac{1}{M}\sum_{m=1}^{M}\mathbb{E}\|h_{t,m} - h_{*,m}\|^2 + \alpha\frac{4\omega\eta^2}{\alpha M}\frac{1}{M}\sum_{m=1}^{M}\mathbb{E}\|g_{t,m} - h_{*,m}\|^2$$

$$\le \left(1-\frac{\alpha}{2}\right)\frac{4\omega\eta^2}{\alpha M}\frac{1}{M}\sum_{m=1}^{M}\mathbb{E}\|h_{t,m} - h_{*,m}\|^2 + \eta^2\frac{6\omega}{M}\frac{1}{M}\sum_{m=1}^{M}\mathbb{E}\|g_{t,m} - h_{*,m}\|^2$$

$$+ (1-\beta)\mathbb{E}\|x_t - x_*\|^2 + \beta\left((1-\gamma\mu)^n\mathbb{E}\|x_t - x_*\|^2 + \gamma^3\frac{1}{\gamma\mu}\hat{\sigma}_{rad}^2.\right)$$

Let us consider

$$\eta^2\frac{1}{M}\sum_{m=1}^{M}\mathbb{E}\|g_{t,m} - h_{*,m}\|^2 = \eta^2\frac{1}{M}\sum_{m=1}^{M}\mathbb{E}\left\|\frac{x_t - x_{t,m}^n}{\gamma n} - \frac{x_* - x_{*,m}^n}{\gamma n}\right\|^2$$

$$\le 2\eta^2\frac{1}{M}\sum_{m=1}^{M}\mathbb{E}\left\|\frac{x_t - x_*}{\gamma n}\right\|^2 + 2\eta^2\frac{1}{M}\sum_{m=1}^{M}\mathbb{E}\left\|\frac{x_{t,m}^n - x_{*,m}^n}{\gamma n}\right\|^2$$

$$\le 2\beta^2\frac{1}{M}\sum_{m=1}^{M}\mathbb{E}\|x_t - x_*\|^2 + 2\beta^2\frac{1}{M}\sum_{m=1}^{M}\mathbb{E}\|x_{t,m}^n - x_{*,m}^n\|^2$$

$$\le 2\beta^2\mathbb{E}\|x_t - x_*\|^2 + 2\beta^2\frac{1}{M}\sum_{m=1}^{M}\mathbb{E}\|x_{t,m}^n - x_{*,m}^n\|^2.$$

Putting all the terms together and using $(1-\gamma\mu)^n \le \frac{9/10 - 1/B}{1+1/B} = \widehat{B} < 1$, $\beta \le \frac{1}{12B\frac{\omega}{M}+1}$ we have

$$\mathbb{E}\Psi_{t+1} \le \left(1 - \beta + 12\frac{\omega}{M}\beta^2 + 12\frac{\omega}{M}\beta^2(1-\gamma\mu)^n + \beta(1-\gamma\mu)^n\right)\mathbb{E}\|x_t - x_*\|^2 + \beta\gamma^3\frac{1}{\gamma\mu}\hat{\sigma}_{rad}^2$$

$$+ 2\beta^2\frac{6\omega}{M}\gamma^3\frac{1}{\gamma\mu}\hat{\sigma}_{rad}^2 + \left(1-\frac{\alpha}{2}\right)\frac{4\omega\eta^2}{\alpha M}\frac{1}{M}\sum_{m=1}^{M}\mathbb{E}\|h_{t,m} - h_{*,m}\|^2$$

$$\le \left(1 - \frac{\beta}{10}\right)\mathbb{E}\|x_t - x_\star\|^2 + \frac{2}{\mu}\beta\gamma^2\hat{\sigma}_{rad}^2 + \left(1-\frac{\alpha}{2}\right)\frac{4\omega\eta^2}{\alpha M}\frac{1}{M}\sum_{m=1}^{M}\mathbb{E}\|h_{t,m} - h_{*,m}\|^2$$

$$\le \max\left(1 - \frac{\beta}{10}, 1 - \frac{\alpha}{2}\right)\Psi_t + \frac{2}{\mu}\beta\gamma^2\hat{\sigma}_{rad}^2.$$

Unrolling this recursion we get the final result. □

