# OpenReview forum: "Federated Optimization Algorithms with Random Reshuffling and Gradient Compression"
_ICLR.cc/2024/Conference — Submitted to ICLR 2024_

### Official Review · Reviewer_mMnw · 2023-11-01

**Soundness:** 4 excellent
**Presentation:** 3 good
**Contribution:** 2 fair
**Rating:** 6
**Confidence:** 3

**Summary:**

This paper introduces an innovative method to enhance communication efficiency in distributed machine learning model training. The proposed method incorporates without-replacement sampling and gradient compression, leading to improved performance in comparison to existing algorithms. The paper provides theoretical analysis and experimental results to support the effectiveness of the proposed approach.

**Strengths:**

1.The paper introduces an innovative method to enhance communication efficiency in distributed machine learning model training. This approach incorporates without-replacement sampling and gradient compression, leading to improved performance compared to existing algorithms.

2.The paper offers a comprehensive validation of the proposed approach by providing both theoretical analysis and empirical results. These findings illustrate the superiority of the proposed method over existing algorithms in terms of convergence rate and communication efficiency.

3.In addition to its contributions, the paper conscientiously addresses the limitations and challenges associated with the proposed approach. It also suggests potential avenues for future research in this area.

**Weaknesses:**

The DIANA-NASTYA algorithm's theoretical analysis is conducted without the need for a strongly convex assumption. A strongly convex assumption often allows for more precise and efficient convergence guarantees. To investigate the impact of such an assumption, further analysis would be necessary to determine whether the algorithm's performance improves, and if so, to what extent.

The experiments primarily revolve around deep learning applications, which are typically considered non-convex problems. However, it's important to note that the paper lacks theoretical analysis specifically tailored to these non-convex problem settings.

This work appears to be an incremental extension of DIANA. While it introduces some additional techniques, the improvements achieved may not be substantial or readily discernible.

**Questions:**

see the weaknesses.

---

> ### Author Response · Authors · 2023-11-19
> **Answer to Review**
>
> **Strengths:**
>
> >1.The paper introduces an innovative method to enhance communication efficiency in distributed machine learning model training. This approach incorporates without-replacement sampling and gradient compression, leading to improved performance compared to existing algorithms.
>
> >2.The paper offers a comprehensive validation of the proposed approach by providing both theoretical analysis and empirical results. These findings illustrate the superiority of the proposed method over existing algorithms in terms of convergence rate and communication efficiency.
>
> >3.In addition to its contributions, the paper conscientiously addresses the limitations and challenges associated with the proposed approach. It also suggests potential avenues for future research in this area.
>
> Thank you for providing positive feedback! We truly appreciate your encouraging words. Your support motivates us to continue our efforts and strive for excellence. If you have any further comments, suggestions, or questions, please feel free to share them. Once again, thank you for taking the time to express your positive impressions.
>
>
> **Weaknesses:**
>
> >The DIANA-NASTYA algorithm's theoretical analysis is conducted without the need for a strongly convex assumption. A strongly convex assumption often allows for more precise and efficient convergence guarantees. To investigate the impact of such an assumption, further analysis would be necessary to determine whether the algorithm's performance improves, and if so, to what extent.
>
> We express our respectful disagreement with the presented assertion. Our theoretical analysis is conducted with the prerequisite of a strongly convex assumption. This specific assumption is instrumental in deriving a linear convergence guarantee, ensuring an exponentially swift convergence towards the neighborhood of the solution. We would appreciate further clarification if your question pertains to the strong convexity assumption. In the context of individual strong convexity, we regretfully acknowledge that we did not include this case in the paper due to space limitations. However, we are open to exploring this aspect in future work, should it be deemed necessary or beneficial for a more comprehensive analysis.
>
> >The experiments primarily revolve around deep learning applications, which are typically considered non-convex problems. However, it's important to note that the paper lacks theoretical analysis specifically tailored to these non-convex problem settings.
>
> We recognize the absence of convergence analysis in non-convex cases within our paper. To fortify the foundation of our theoretical findings, we conducted experiments with a specific emphasis on logistic regression. The exploration of deep learning experiments holds significant interest given their widespread applications in various domains. As articulated in our response to reviewer TutK, our intentional omission of insights into non-convex cases is a deliberate choice. We believe that a dedicated exploration of non-convex scenarios warrants separate consideration and could be a promising avenue for future research.

---

> > ### Author Response · Authors · 2023-11-19
> > **Reply to Answer 2**
> >
> > >This work appears to be an incremental extension of DIANA. While it introduces some additional techniques, the improvements achieved may not be substantial or readily discernible.
> >
> > We respectfully dissent from this viewpoint. We maintain that every outcome presented in this paper exhibits a novel character, and the introduced combinations possess a depth that transcends simplicity. It is pertinent to note that had the problem we address been inherently straightforward, the results would have undoubtedly surfaced earlier within the literature, given the inherent significance of the issues at hand.
> >
> > It is worth highlighting that without-replacement sampling stands as a conventional choice among practitioners, especially within the domain of distributed learning applications. While the enhanced analysis of random reshuffling was recognized in 2020, advanced communication compression techniques had been extant since at least 2019. Nonetheless, the specific problem that we tackle remained unexplored prior to our contribution, thereby substantiating the non-trivial nature of the amalgamated components. Furthermore, as expounded within the paper, the ostensibly intuitive amalgamation, Q-RR, demonstrates no enhancements beyond the efficacy of QSGD. This realization holds significant non-triviality in its own right. In response, we introduce a distinct methodology, DIANA-RR. A noteworthy observation is that unlike DIANA, achieving our results necessitated the incorporation of multiple shifting vectors within DIANA-RR. This intricacy was not immediately apparent, further underscoring the profound departure of DIANA-RR from its rudimentary combination.
> >
> > Lastly, we emphasize that prevailing methods, which incorporate both local steps and compression, undergo analysis under more constrictive assumptions such as homogeneity or bounded variance (FedCOM, FedPAQ), resulting in inferior convergence guarantees compared to Q-NASTYA and DIANA-NASTYA (FedCOM, FedPAQ, FedCRR). This observation accentuates the non-trivial nature of the extensions incorporating local steps.
> >
> > Should the esteemed reviewers seek additional elucidation, raise concerns, or express the need for discourse, we extend a warm invitation to engage in further discussion. Your inquiries will be met with enthusiasm and a willingness to contribute.

---

### Official Review · Reviewer_TutK · 2023-11-01

**Soundness:** 4 excellent
**Presentation:** 4 excellent
**Contribution:** 2 fair
**Rating:** 6
**Confidence:** 4

**Summary:**

This paper combines several existing techniques to accelerate the communication complexity of distributed stochastic optimization. It is known that random reshuffling gives better convergence rate of stochastic gradient descent, and gradient compression can save communication bandwidth by sending fewer bits over the network. It is quite natural to consider a distributed learning algorithm with both random reshuffling and gradient compression. However, the noise introduced by the gradient compression might cancel out the improvements of convergence of random reshuffling, thus it is not a priori clear if the combination is actually useful. This paper proves several theoretical guarantees and that random reshuffling can indeed improve upon some existing algorithms with gradient compression. This paper also provide experiments that demonstrate the results.

**Strengths:**

The theorems and proofs are stated very clearly. I don't see any major flaws in the proofs. The paper discusses the improvements over previous results thoroughly.

**Weaknesses:**

The results in this paper are not suprising. The proofs only utilize existing methods and techniques and are more or less routine.

**Questions:**

I only have one major question. I notice that the authors prove some non-strongly convex results in the appendix B.2 and C.2. Can the authors provide some discussion on this matter? How does the non-strongly convex setting change the results and the improvements? Have the authors considered nonconvex setting? I really like to see more discussion for alternative assumptions.

---

> ### Author Response · Authors · 2023-11-19
> **Reply to Review**
>
> **Strengths:**
>
> >The theorems and proofs are stated very clearly. I don't see any major flaws in the proofs. The paper discusses the improvements over previous results thoroughly.
>
> Thank you for providing such encouraging feedback! Your positive remarks are truly appreciated and serve as a source of motivation for our ongoing work. If you have any additional thoughts, suggestions, or questions, we would be more than happy to hear them. Once again, thank you for taking the time to share your positive impressions with us.
>
> **Weaknesses:**
>
> >The results in this paper are not surprising. The proofs only utilize existing methods and techniques and are more or less routine.
>
> We respectfully dissent from this viewpoint. We maintain that every outcome presented in this paper exhibits a novel character, and the introduced combinations possess a depth that transcends simplicity. It is pertinent to note that had the problem we address been inherently straightforward, the results would have undoubtedly surfaced earlier within the literature, given the inherent significance of the issues at hand.
>
> It is worth highlighting that without-replacement sampling stands as a conventional choice among practitioners, especially within the domain of distributed learning applications. While the enhanced analysis of random reshuffling was recognized in 2020, advanced communication compression techniques had been extant since at least 2019. Nonetheless, the specific problem that we tackle remained unexplored prior to our contribution, thereby substantiating the non-trivial nature of the amalgamated components.
> Furthermore, as expounded within the paper, the ostensibly intuitive amalgamation, Q-RR, demonstrates no enhancements beyond the efficacy of QSGD. This realization holds significant non-triviality in its own right. In response, we introduce a distinct methodology, DIANA-RR. A noteworthy observation is that unlike DIANA, achieving our results necessitated the incorporation of multiple shifting vectors within DIANA-RR. This intricacy was not immediately apparent, further underscoring the profound departure of DIANA-RR from its rudimentary combination.
>
> Lastly, we emphasize that prevailing methods, which incorporate both local steps and compression, undergo analysis under more constrictive assumptions such as homogeneity or bounded variance (FedCOM, FedPAQ), resulting in inferior convergence guarantees compared to Q-NASTYA and DIANA-NASTYA (FedCOM, FedPAQ, FedCRR). This observation accentuates the non-trivial nature of the extensions incorporating local steps.
>
> Should the esteemed reviewers seek additional elucidation, raise concerns, or express the need for discourse, we extend a warm invitation to engage in further discussion. Your inquiries will be met with enthusiasm and a willingness to contribute.
>
> **Questions:**
>
> >I only have one major question. I notice that the authors prove some non-strongly convex results in the appendix B.2 and C.2. Can the authors provide some discussion on this matter? How does the non-strongly convex setting change the results and the improvements? Have the authors considered nonconvex setting? I really like to see more discussion for alternative assumptions.
>
> Thank you for your questions! Firstly, we aim to compare the complexities of Q-RR and DIANA-RR. The DIANA technique proves instrumental in eliminating terms associated with compression variance. Consequently, our focus shifts to comparing the complexities of Q-RR under different assumptions. In the main section, we present the convergence rate of Q-RR when each summand is strongly convex. This results in the term related to random reshuffling being $\frac{\sigma_{rad}}{\sqrt{\tilde{\mu}\varepsilon}}$, as determined by the analysis for RR in this particular case.
>
> However, for Q-RR under the assumption that each $f_i$ is merely convex, with only the objective function being strongly convex, a larger term related to shuffling emerges: $\sqrt{\frac{n\tilde{L}}{\varepsilon\mu^3}}\sigma_{*, n}$. Once again, this difference arises due to our analytical approach. The same observation holds for DIANA-RR.
>
> Regarding the non-convex case, unfortunately, our study does not delve into non-convex setups. This decision is motivated by two primary reasons. Firstly, the rigorous analysis demonstrating the advantages of RR over standard SGD remains an open question. Even in recent work (https://arxiv.org/pdf/2305.19259v3.pdf), there is an identified mistake, leading us to withhold insights in this direction. Secondly, our choice of algorithms with compression, such as MARINA/DASHA/EF21, would be contingent on the availability of a robust analysis for RR.
>
> While recognizing the potential importance of non-convex cases, we believe that thorough exploration and contributions in this realm merit a dedicated effort, possibly in a separate paper.

---

> > ### Comment · Reviewer_TutK · 2023-12-05
> > **Updated review**
> >
> > I will keep my previous score. The paper would be better if there are more discussions on non strongly convex setting.

---

### Official Review · Reviewer_VtDw · 2023-11-03

**Soundness:** 3 good
**Presentation:** 2 fair
**Contribution:** 2 fair
**Rating:** 5
**Confidence:** 5

**Summary:**

In this work, authors study the behavior of Federated learning with gradient compression and without-replacement sampling. Authors first develop a distributed variant of random reshuffling with gradient compression (Q-RR) and show that the compression variance will overwhelm the variance of the gradient. Next, authors propose a variant
of Q-RR called Q-NASTYA, which uses local gradient steps and different local and global stepsizes.

Thanks for the authors' feedback, I have changed my score accordingly.

**Strengths:**

The idea sounds interesting to me. In federated learning, the communication cost can be the bottleneck for the scalability of the training system, especially on edge devices. Moreover, without-replacement has attracted lots of interest in recent studies. The motivation of this work is solid.

**Weaknesses:**

There are several questions that need to be addressed:
1) I see no experiment results in the main draft, although authors put those details in the supplementary, I still believe it shall be put in the main part.
2) The communication compression is at most 50% since only half of the rounds are compressed. This is not the optimal design since there should be a better way for compressing the second round of communication, since you have already assumed that all workers participate the updating of the global x_t at each round.
3) About the compression, why not use the error-compensated compression (e.g. DoubleSqueeze and PowerSGD) since it is the SOTA method for incorporating compression in optimization? With error compensation, the variance of the compression introduced in the training will be greatly reduced and you even do not need the compression to be unbiased.
4) The paper is hard to follow, I still cannot get a high level comparison of your algorithms with other works and fail to find a clue about why your design works.

**Questions:**

Please refer to my question about the weakness part.

---

> ### Author Response · Authors · 2023-11-19
> **Reply to Review**
>
> **Strengths:**
> >The idea sounds interesting to me. In federated learning, the communication cost can be the bottleneck for the scalability of the training system, especially on edge devices. Moreover, without-replacement has attracted lots of interest in recent studies. The motivation of this work is solid.
>
> Thank you for providing positive feedback! We greatly appreciate your encouraging remarks, and they serve as a motivating factor for our ongoing efforts. If you have any further thoughts or suggestions, we would be delighted to hear them. Once again, thank you for taking the time to share your positive impressions with us.
>
> **Weaknesses:**
>
> >I see no experiment results in the main draft, although authors put those details in the supplementary, I still believe it shall be put in the main part.
>
> Thank you for your suggestion! While the primary contribution of our work is centered around theory, we are considering the incorporation of experimental results into the main section. This approach aims to provide a more well-rounded presentation, combining theoretical insights with practical findings. Your input is appreciated, and we're open to ensuring a comprehensive and balanced representation of our work.
>
> >The communication compression is at most 50% since only half of the rounds are compressed. This is not the optimal design since there should be a better way for compressing the second round of communication, since you have already assumed that all workers participate in the updating of the global x_t at each round.
>
> In the context of our research, we operate under the assumption that the transmission of data from the workers to the server is a time-consuming process, a characteristic observed in numerous related studies.
>
> Alistarh, D., Grubic, D., Li, J., Tomioka, R., & Vojnovic, M. (2017). QSGD: Communication-efficient SGD via gradient quantization and encoding. Advances in neural information processing systems, 30.
>
> Karimireddy, S. P., Rebjock, Q., Stich, S., & Jaggi, M. (2019, May). Error feedback fixes signsgd and other gradient compression schemes. In International Conference on Machine Learning (pp. 3252-3261). PMLR.
>
> Recognizing the significance of this aspect, we are aware of existing literature that delves into bidirectional compression techniques. We find the exploration of bidirectional compression to be a compelling avenue for potential extension, an area where further investigation could yield valuable insights and advancements in our understanding of the communication dynamics between workers and servers.
>
> >About the compression, why not use the error-compensated compression (e.g. DoubleSqueeze and PowerSGD) since it is the SOTA method for incorporating compression in optimization? With error compensation, the variance of the compression introduced in the training will be greatly reduced and you do not even need the compression to be unbiased.
> Thank you for your recommendation!
>
> We respectfully disagree with the assertion made. Error-compensated compression finds application in biased compression schemes. Although error-compensated methods demonstrate effectiveness in practical scenarios, the theoretical analysis falls short of providing strong convergence results, particularly in convex and strongly convex regimes.
>
> Theoretically, unbiased compression stands out as a more preferable choice across all regimes (strongly convex, convex, and non-convex). In the case of unbiased compression, the DIANA method serves as an error compensation mechanism for unbiased compressors.
>
> Furthermore, the combination of biased compression with sampling without replacement poses an open problem, one that we defer to future research endeavors.
>
> The DIANA method, by completely eliminating the variance induced by unbiased compression, coupled with sampling without replacement, offers improved guarantees and practical performance. This represents our notable contribution to the field.

---

> ### Author Response · Authors · 2023-11-19
> **Reply to Review 2**
>
> >The paper is hard to follow, I still cannot get a high level comparison of your algorithms with other works and fail to find a clue about why your design works.
>
> Thank you for your feedback on the paper. Allow us to provide a high-level overview of our algorithms and the contributions of our work.
>
> Our approach in distributed optimization initially leverages the Distributed SGD method, incorporating Random Reshuffling (Sampling without replacement) to mitigate variance. To further reduce the volume of communicated bits, we introduced Compressed SGD (Q-SGD). Extending the idea of sampling without replacement, we present the Compressed Random Reshuffling method (Q-RR). However, the straightforward fusion of Random Reshuffling and Compression doesn't yield improved convergence guarantees, as the variance introduced by Compression outweighs the sampling variance. To address this, we integrate a variance reduction mechanism for compression, specifically the DIANA technique, into the Compressed Random Reshuffling method, resulting in the DIANA-RR method. Notably, this method exhibits superior convergence complexity compared to the original DIANA method with sampling with replacement. Our approach necessitated significant modifications and the development of a novel proof technique to achieve these results.
>
> In the context of federated learning scenarios, where minimizing communication rounds is crucial, local methods are commonly employed. The well-established framework for Federated Learning is Federated Averaging (FedAvg). To address variance concerns, Federated Random Reshuffling is introduced. We also explore an advanced version, Nasty Method, which incorporates several step sizes. Adding a compression mechanism to Nasty Method and, akin to Q-RR, a naive combination doesn't yield optimal results. We further enhance the Nasty algorithm by introducing a DIANA-type mechanism, which significantly improves convergence results. This holistic approach aims to offer a comprehensive understanding of our algorithms and their contributions in the context of distributed optimization and federated learning scenarios.

---

### Official Review · Reviewer_iwJs · 2023-11-08

**Soundness:** 3 good
**Presentation:** 3 good
**Contribution:** 3 good
**Rating:** 6
**Confidence:** 4

**Summary:**

The paper studies the convergence of federated learning algorithms with gradient compression and random reshuffling.

**Strengths:**

The authors show that the naive combination of compression and reshuffling (called Q-RR) doesn’t outperform compressed SGD without reshuffling. To alleviate this issue, they develop an algorithm combining Q-RR with DIANA, hence reducing the compression variance. They then introduce a version of the algorithm supporting the local steps.

**Weaknesses:**

Also questions included below:

-- The notation for $\pi$ in section 1.3 doesn’t match the rest of the paper.

-- Algorithm 2, line 3: is $\pi_m$ sampled for each machine $m$? Do machines have different permutations? Same for other algorithms.

-- Algorithm 3: Are lines 6 and 7 preformed on the server? Then they are not performed in parallel

-- Definition 2.1 - What is the meaning of the sequence? What is the “scope” of the sequence (you have different $\pi_m^i$ for different
$t$)? If this sequence is different for each $t$, how do you aggregate $\sigma_{rad}^2$ over different $t$ (do you take a maximum)? Also, $x_\star$ is undefined

-- I think that derivations (e.g. on page 25) are rather hard to parse. I would introduce auxiliary notation for $f_m^{\pi_m^i}$ and for $h \cdots$.

**Questions:**

-- Page 4 - “Finally, to illustrate our theoretical findings we conduct experiments on federated linear regression tasks.” - Did you mean logistic regression? Also, I believe that this line makes your experimental results look weaker than what they actually are.

-- Since you use $\zeta_\star$ and $\sigma_\star$ in multiple results, they should be defined in an Assumption, instead of being defined in Theorem 2.1

-- Page 7 - “we can do enable…”

-- Page 15, after Equation (9) - broken math

-- Page 24: $f^{i, \pi}$

-- “mygreen” in bookmark names, e.g. “Algorithm mygreen Q-RR”

---

> ### Author Response · Authors · 2023-11-19
> **Reply to the Review**
>
> **Strengths:**
>
> >The authors show that the naive combination of compression and reshuffling (called Q-RR) doesn’t outperform compressed SGD without reshuffling. To alleviate this issue, they develop an algorithm combining Q-RR with DIANA, hence reducing the compression variance. They then introduce a version of the algorithm supporting the local steps.
>
> Thank you for taking the time to review and provide positive feedback. Your input is highly valuable to us. If you have any further comments or suggestions, we would greatly appreciate hearing them. Once again, thank you for your thoughtful review.
>
>
> **Weaknesses:**
>
> > The notation for $\pi_i$  in section 1.3 doesn’t match the rest of the paper.
>
> We intend to rectify this issue within the introduction, implementing consistent notation that will be maintained uniformly across the entirety of the paper. This ensures a cohesive and standardized presentation of our content, contributing to the overall clarity and coherence of our work.
>
> >Algorithm 2, line 3: is $\pi_m$ sampled for each machine $m$? Do machines have different permutations? Same for other algorithms.
>
> In Algorithm 2, we generate samples by obtaining distinct permutations for each machine. In Algorithm 3 (DIANA-RR), our analysis involves employing a shuffle once option for each machine.
>
> > Algorithm 3: Are lines 6 and 7 preformed on the server? Then they are not performed in parallel
>
> Line 6 delineates the communication protocol between workers and a server. Operations specified in Lines 7 and 8 are executed on both the server and the workers. Notably, we work with a matrix of shifts, but each worker retains only $n$ of them. Given that the algorithm necessitates transmitting compressed differences exclusively from each node to the server, we maintain the average of shifts on the server and update this value exclusively. Consequently, we perform the computation of shifts in parallel.
>
> > Definition 2.1 - What is the meaning of the sequence? What is the “scope” of the sequence (you have different $\pi^i_m$ for different $t$ )? If this sequence is different for each $t$ , how do you aggregate $\sigma^2_{rad}$ over different $t$ (do you take a maximum)? Also, $x_{\star}$ is undefined
>
> The rationale behind this sequence is elucidated in detail in the following document: https://arxiv.org/pdf/2102.06704.pdf. This sequence constitutes a supportive set of points, facilitating enhanced analysis with superior rates. Consequently, the incorporation of a shuffling radius leads to improved rates. Notably, according to Definition 2.1, $\sigma^2_{rad}$ remains independent of the number of iterations. This could occur if the stepsize $\gamma$ varied across iterations.
>
> Following Assumption 2, we posit that the function $f$ possesses a unique minimizer denoted as $x_{\star}$.
>
> > I think that derivations (e.g. on page 25) are rather hard to parse. I would introduce auxiliary notation for $f_m^{\pi^i_m}$ and for h.
>
> Thank you for your recommendation! We will certainly consider how to make adjustments. Nevertheless, it currently appears readable to us.
>
> **Questions:**
>
> > Page 4 - “Finally, to illustrate our theoretical findings we conduct experiments on federated linear regression tasks.” - Did you mean logistic regression? Also, I believe that this line makes your experimental results look weaker than what they actually are.
>
> Yes, we indeed made a small typo. Thank you for bringing it to our attention! We appreciate your recommendation, and we intend to address and correct it in the final version!
>
> > Since you use $\zeta_{\star}$ and $\sigma_{\star}$ in multiple results, they should be defined in an Assumption, instead of being defined in Theorem 2.1
>
> Thank you for your suggestion! We will consider how we can enhance it.
>
> > Page 7 - “we can do enable…”
>
>
>
> We will address and rectify it.
>
> > Page 15, after Equation (9) - broken math
>
> Thank you for identifying the typos! We will correct them in the final version.
>
> > Page 24: $f^{i,\pi}$
>
> Thank you for identifying the typos! We will correct them in the final version.
>
> > “mygreen” in bookmark names, e.g. “Algorithm mygreen Q-RR”
>
> Thank you for bringing this to our attention! We will correct it in the final version.

---

### Meta-Review · Area_Chair_SEpU · 2023-12-06

**Metareview:**

The paper considers federated learning settings, focusing on stochastic smooth strongly convex optimization settings with without-replacement sampling (shuffled SGD with random reshuffling-RR). The paper's primary contributions are in combining gradient compression (which is commonly used in these settings to improve communication efficiency) with RR. The first insight provided in the paper is that gradient compression when employed jointly with RR does not offer any benefits compared to gradient compression with standard with-replacement sampling, due to the limitations imposed by the additional compression variance. To address this issue, the paper shows that the compression variance can be effectively controlled using variants of existing variance reduction techniques, in which case RR does have benefits over standard with-replacement sampling.

The literature on theoretical guarantees for with-replacement sampling is fairly recent and there is still much missing from our current understanding, thus the paper does contribute to this area. However, as pointed out in the reviews, strong convexity is a very strong assumption and, further, it is not well explained or clear what the main technical ideas in the paper are, as the paper appears to be a combination of existing ideas and techniques (this is not a bad thing on its own, but we do want to see technical depth). Even after this issue was raised in almost all the reviews, the authors provided a vague response asserting novelty and arguments of the form "no one has done it before so it must be challenging." While this may be true, the answers did not provide evidence/clear arguments for main technical ideas that were needed, which is what would have helped bring the paper over the threshold.

**Justification For Why Not Higher Score:**

Neither I nor the reviewers could understand what the main new technical ideas were needed to get the result, giving us all the impression that the paper is at the level of a grad student exercise in combining existing techniques.

**Justification For Why Not Lower Score:**

N/A

---

### Decision · Program_Chairs · 2024-01-16

Reject